# Vegetation and fire regimes in the Neotropics over the last 21,000 years

Thomas Kenji Akabane<sup>1,2</sup>; Cristiano Mazur Chiessi<sup>3</sup>; Paulo Eduardo De Oliveira<sup>1,4</sup>; Jennifer Watling<sup>5</sup>; Ana Carolina Carnaval<sup>6</sup>; Vincent Hanquiez<sup>2</sup>; Dailson José Bertassoli Jr.<sup>1</sup>; Thaís Aparecida Silva<sup>1</sup>,

Marília H Shimizu<sup>7</sup>, Anne-Laure Daniau<sup>2</sup>






- 1 Institute of Geosciences, University of São Paulo, Rua do Lago 562, 05508-080 São Paulo SP, Brazil
- 2 University of Bordeaux, CNRS, Bordeaux INP, EPOC, UMR 5805, F-33600 Pessac, France
- 3 School of Arts, Sciences and Humanities, University of São Paulo, Av. Arlindo Bettio 1000, 03828-000 São Paulo SP, Brazil
- 4 Keller Science Action Center, The Field Museum of Natural History, 1400 S. Lake Shore Drive, 60605, Chicago IL, USA
  - 5 Museum of Archaeology and Ethnography, University of São Paulo, São Paulo 05508-070, Brazil
  - 6 Biology Department, City College of New York, New York, USA
  - 7 General Coordination of Earth Science, National Institute for Space Research, São José dos Campos, Brazil
- 15 Correspondence to: Thomas Kenji Akabane (thomask.akabane@gmail.com)

Abstract. Vegetation and fire activity have dynamically changed in response to past variations in global and regional climate. Here we investigate these responses across the Neotropics based on the analysis of modern vegetation distribution and fire activity in relation to modern climate patterns, and a compilation of 255 vegetation records and 131 charcoal records encompassing the last 21,000 years before present (ka) in relation to past climate changes. Our analyses on the dynamics of past tree cover and fire activity focus on seven subregions: (1) northern Neotropics (NNeo); (2) tropical Andes (TrAn); (3) Amazonia; (4) northeastern Brazil (NEB); (5) central-eastern Brazil (CEB); (6) southeastern South America (SESA); and (7) extratropical Andes (ExTrAn). The regionalized assessment unveils spatial heterogeneity in the timing and controls of vegetation and fire dynamics. Temperature, atmospheric CO<sub>2</sub> concentrations, and precipitation exhibit distinct and alternating roles as primary drivers of tree cover and fire regime changes with additional impacts from human activity. During the Last Glacial Maximum (LGM, here covering 21-19 ka), arboreal growth in high elevation sites (TrAn) and in sub- and extra-tropical latitudes (SESA and ExTrAn) was mainly limited by low temperatures and atmospheric CO2 concentrations, while fuel-limited conditions restrained fire activity. In warmer tropical regions (NNeo, Amazonia, CEB), moisture availability was likely the main controlling factor of both vegetation and fire, with the effects of low CO<sub>2</sub> amplifying these constraints. Throughout the deglacial phase (19-11.7 ka), progressive warming and increasing atmospheric CO2 concentrations fostered a gradual biomass expansion, and together these changes led to intensified fire activity in the sub- and extra-tropical temperature-limited regions. Meanwhile, increased (decreased) precipitation associated with millennial-scale events favored increases (decreases) in tree cover in regions such as CEB and NEB (NNeo). Between 14-13 ka, most southern latitude subregions (Amazonia, CEB, SESA, ExTrAn) saw a rise in fire activity coeval with a second rapid warming, contrary to decreased fire activity in NNeo amid relatively wetter conditions. Throughout the Holocene, when temperature and atmospheric CO<sub>2</sub> fluctuations were lower, shifts in precipitation became the primary driver of vegetation and fire dynamics across all the Neotropics. Changes in the Intertropical Convergence Zone and gradual intensification of the South American Summer Monsoon throughout the Holocene favored a continuous increase in tree cover over Amazonia, CEB, and SESA, but led to a forest cover decrease in NNeo and NEB. From the early- to the mid-Holocene, the strengthening of the Southern Westerly Winds promoted vegetation expansion and fire regime weakening in ExTrAn. In the late Holocene, human impacts became more pronounced, with a clearer effect on regional tree cover and fire activity, particularly in NNeo and TrAn.

#### 1 Introduction








The Neotropics is the most species-rich biogeographical domain and home of at least one third of global biodiversity (Raven et al., 2020). It extends from the southern parts of North America to southernmost South America and encompasses a wide range of environments, from the wettest rainforests to the driest deserts on Earth. The distinctly high diversity in the region is attributed to a combination of biotic processes, such as *in situ* adaptation (Simon et al., 2009), species interchange (Antonelli et al., 2018) and ecological interactions (Fine et al., 2004), together with abiotic processes including landscape evolution (Hoorn et al., 2010; Richardson et al., 2001) and climate fluctuations throughout the Cenozoic (Cracraft et al., 2020; Jaramillo et al., 2006; Rull, 2011; Sawakuchi et al., 2022). Largely influenced by changes in the tectonic regime, the long-term climate cooling throughout the Cenozoic induced a retraction of warm tropical forests and culminated in the onset of pronounced glacial-interglacial cycles in the Quaternary (Morley, 2011; Westerhold et al., 2020).

Quaternary glacial-interglacial cycles together with millennial-scale climate changes played a significant role in shaping species distribution over time across the Neotropics. For instance, the distinct hydroclimate (Baker and Fritz, 2015; Wang et al., 2017), 3–8 °C colder temperatures (Bush et al., 2001; Chiessi et al., 2015; Colinvaux et al., 1996; Wille et al., 2001), and the ca. 100 ppm lower atmospheric CO<sub>2</sub> concentrations (CO<sub>2atm</sub>) (Bereiter et al., 2015; Petit et al., 1999) of the Last Glacial Maximum (LGM, typically between 23,000–19,000 yr ago (23 – 19 ka)) induced substantial and widespread changes in vegetation composition and structure. These changes predominantly, but not exclusively, led to a lower biomass state with a weaker fire regime (e.g., Behling, 2002a; Bush et al., 2009; Bush and Flenley, 2007; Haas et al., 2023; Ledru, 2002; Nanavati et al., 2019; Power et al., 2010b). However, the magnitude and timing of these changes in vegetation and fire were hetereogeneous, resulting in diverse regional patterns. This variability highlights the complexity of environmental dynamics since the LGM and the need for region-specific analyses to understand ecosystem responses to past climatic shifts.

Investigating the primary mechanisms controlling ecosystem dynamics is crucial for anticipating the impacts of ongoing climate changes. The projected large-scale changes in specific climate system components, unprecedented in the instrumental record, may only find parallels in the geological past (Wunderling et al., 2024). In this sense, sedimentary pollen and charcoal records offer the oportunity to obtain valuable insights into long-term responses and a deeper understanding of the linkages between climate changes and environmental shifts (Daniau et al., 2012; Flantua et al., 2016; Marlon et al., 2013; Nanavati et al., 2019; Power et al., 2010a).

Here we use 255 pollen records and 131 charcoal records to reconstruct tree cover and fire regime changes in the Neotropics spanning the last 21 ka. We also use compiled archeological radiocarbon data available in literature and databases to discuss vegetation and fire changes in relation to potential anthropogenic impacts (Fig. A1). We focus on subregions including the (1) northern Neotropics (NNeo); (2) tropical Andes (TrAn); (3) Amazonia; (4) central-eastern Brazil (CEB); (5) northeastern Brazil (NEB); (6) southeastern South America (SESA); and (7) extratropical Andes (ExTrAn) (Fig. 1 and 2). These subregions were chosen based on data availability and are delimited by their present-day climatic and vegetation features. Thus, our scope focuses on the underlying controls of long-term and broad-scale patterns. Furthermore, we analyze modern fire patterns using satellite data to compare with climate and vegetation parameters to contribute to our interpretations. These analyses allow us to assess distinct vegetation and fire dynamics from each subregion and the competing drivers influencing their responses.

#### 2 Vegetation and climate settings








- (1) The <u>northern Neotropics (NNeo)</u>, mainly comprising Central America and northernmost South America, is mostly covered by moist tropical forests, pine-oak forests, dry forests, and shrublands (Fig. 1a). The region exhibits a seasonal climate, with lowland mean annual temperatures ranging from 25 to 28 °C (Fig. 1b). Precipitation patterns are strongly influenced by the latitudinal shifts of the Intertropical Convergence Zone (ITCZ) with a regional mean of 1600 mm yr<sup>-1</sup>. During boreal summer, when the ITCZ shifts further north, most of the rainfall is transported into the Central American region by the Caribbean Low Level Jet (Cook and Vizy, 2010) (Fig. 1d). Large wildfires in the region are associated with humid periods succeeded by extreme dry periods, often linked with El Niño Southern Oscillation (ENSO) variability with La Niña-driven wet events followed by El Niño-driven droughts (Ponce-Calderón et al., 2021).
- (2) <u>Amazonia</u> is the most extensive tropical rainforest on Earth, marked by weak seasonality, mean monthly temperatures ranging between 25 and 27 °C, and mean annual precipitation of 2300 mm yr¹ (Fisch et al., 1998; Marengo, 1992). Precipitation seasonality increases south- and eastwards, while parts of northwestern Amazonia remain wet throughout the year (Fisch et al., 1998; Marengo, 1992) (Fig. 1c,d). During austral summer, increased land-ocean thermal contrast enhances the atmospheric transport of humidity towards the continent promoting the South American Summer Monsoon (SASM), responsible for most of the precipitation over Amazonia (Garreaud et al., 2009; Vera et al., 2006) (Fig. 1c). Persistent moist conditions of the rainforest naturally inhibit wildfires. However, initial fire events whether triggered by severe droughts (e.g., related to El Niño), ongoing climate changes, and/or human impacts further increase forest flammability through canopy degradation and fuel accumulation. This favors subsequent fire events, thereby fostering a positive feedback loop (Brando et al., 2020; Bush et al., 2008; Cochrane et al., 1999; Nepstad et al., 1999).
- (3) The herein defined *Tropical Andes (TrAn)* comprises areas above 2200 m altitude, which include (i) upper montane forests from 2300 to 3300 m, (ii) the páramo, from ca. 3200, and (iii) the puna, from ca. 3700 m, to the snow line (Troll, 1968) (Fig. 1a). Precipitation patterns in the region are heterogeneous and partially linked to the SASM (Espinoza et al., 2020; Segura et al., 2019) (Fig. 1c). Andean montane forests feature the wettest parts of Amazonia recording more than 5000 mm yr<sup>-1</sup> in some areas (Espinoza et al., 2020) (Fig. 1c). The páramo and the puna, the alpine vegetation of the Andes, are biogeographically separated by the Huancabamba depression at ca. 6°S (Cuesta et al., 2017; Troll, 1968). Most of the páramo, located north of the Huancabamba depression, receives high precipitation between 1000 and 2000 mm yr<sup>-1</sup> (Cuatrecasas, 1968). The puna, located south of the Huancabamba depression, covers the Altiplano under precipitation from 200 to 500 mm yr<sup>-1</sup> (Vuille and Keimig, 2004) (Fig. 1c,d). Most of the annual precipitation in the region is related to moisture-laden easterly winds from Amazonia and associated with the onset of the Bolivian High, an upper-level highpressure cell linked to the SASM. The interannual precipitation variability is influenced by zonal winds modulated by sea surface temperatures (SST) across the tropical Pacific Ocean (Garreaud et al., 2003; Vuille, 1999). During weakened easterlies, such as during El Niño events, the incursion of dry-warm upper-level westerly winds inhibit precipitation in the Altiplano and weakens the Bolivian High. Contrarily, during La Niña events, the incursion of Amazon moisture conveyed by easterly winds is facilitated (Garreaud et al., 2003; Vuille, 1999). While natural fires in the forests of the TrAn are rare, wildfires in grasslands are frequent, generally of low-intensity, and predominantly driven by human activity (Bush et al., 2015; Gutierrez-Flores et al., 2024).
- (4) <u>Northeastern Brazil (NEB)</u> includes mostly the xeric vegetation of Caatinga (Fig. 1a). The region is characterized by mean annual temperatures of 24 to 28 °C and semiarid conditions with precipitation below 900 mm yr<sup>-1</sup>, concentrated between February and May, while potential annual evapotranspiration exceeds 2200 mm yr<sup>-1</sup> (Pinheiro et al., 2016). Low precipitation in the region is driven by the Nordeste Low, a high-pressure cell dynamically linked to the Bolivian High

- (Lenters and Cook, 1997). In Caatinga, scarce fuel availability related to low biomass production limits regular fire events (Alvarado et al., 2020; Argibay et al., 2020).
  - (5) <u>Central-eastern Brazil (CEB)</u> is primarily associated with Cerrado vegetation, characterized by a mosaic of physiognomies, ranging from open grasslands and savannas to closed shrublands and woodlands, typically covered by a continuous herbaceous layer (Eiten, 1972). The climate of CEB is characterized by the austral summer establishment of the SASM, marked by the occurrence of the South Atlantic Convergence Zone (SACZ), followed by a dry season lasting four to six months, and has an annual precipitation of about 750 to 2000 mm yr<sup>-1</sup> (Eiten, 1972; Goodland, 1971; Vera et al., 2006) (Fig. 1c). In this ecosystem, wildfires are regular and play a key role in controlling vegetation physiognomy and biodiversity (Durigan and Ratter, 2016; Mistry, 1998; Moreira, 2000).



- (6) <u>Southeastern South America (SESA)</u> is mostly occupied by the Atlantic Forest, composed of both evergreen and semideciduous forests, as well as mixed forests in the southern and mountainous regions (Fig. 1a). SESA mean annual temperatures range from ca. 25 °C in the tropical forests to 12 °C in the southern mixed forests (Kamino et al., 2019; Ribeiro et al., 2011) (Fig. 1b). Coastal regions experience over 1800 mm yr<sup>-1</sup> of precipitation well-distributed throughout the year, while inland areas are relatively more seasonal, ranging from 1300 to 1600 mm yr<sup>-1</sup> (Kamino et al., 2019). Most annual precipitation in the regions influenced by the SACZ occurs during austral summer (Fig. 1c). Predominantly during austral autumn and winter, the South Atlantic anticyclone affects the region, leading to lower temperatures and reduced rainfall, whereas to the south of the SESA, cold fronts favor moderate precipitation levels (Kamino et al., 2019) (Fig. 1d). Wildfires are naturally rare in the Atlantic Forest due to high moisture and dense forest cover, however, large scale degradation, rising temperatures and extreme climatic events, such as those related to La Niña years, intensify wildfires frequency and severity (Jesus et al., 2022; da Silva Junior et al., 2020).
- (7) Extratropical Andes (ExTrAn), including Patagonia and Tierra del Fuego, primarily encompasses wet-temperate forests, alpine forests and grasslands. Mean annual temperature ranges from 3 to 16 °C between the latitudes 55 and 32°S (Fig. 1b). The zonal rainfall gradient is abruptly sharp, with the west part of the Andes cordillera receiving most of the precipitation, while drier conditions prevail in the eastern side (Coronato et al., 2005; Endlicher and Santana, 1988) (Fig. 1c,d). Precipitation is controlled by the Southern Westerly Winds (SWW), which shift southward during austral summer, while during austral winter they weaken and expand equatorward (Garreaud et al., 2009) (Fig. 1c,d). Fire activity in the region exhibits a gradient from fuel-limited conditions to the north of 32° S to moisture-limited conditions to the south of 40° S (Holz et al., 2012). Fire events are favored by intermediate productivity levels, with combined contribution of woody and herbaceous vegetation. Fires are primarily conditioned by dry and warm events in moisture-limited areas or anomalously positive rainfall over fuel-limited areas, usually associated with ENSO variability, changes in the Antarctic Oscillation, or in the Pacific Subtropical Anticyclone, which shifts the SWW poleward (Holz et al., 2012; Kitzberger et al., 2022; Kitzberger and Veblen, 1997).

Fig. 1 – Vegetation and climate settings of the Neotropics. Studied sites (white circles) and target subregions (numbered polygons) are depicted. (a) Ecoregions, TrSuMBF – Tropical subtropical moist broadleaf forests; TrSuDBF – Tropical subtropical dry broadleaf forests; TrSuGSSh – Tropical subtropical grasslands, savannas, and shrublands; TeBMiF – Temperate broadleaf mixed forests; TeCF – Temperate coniferous forests; FGS – Flooded grasslands and savannas; DXSh – Desert and xeric shrublands; MoGSh – Montane grasslands and shrublands; MeFWSc – Mediterranean forests, woodlands, and scrubs; TeGShS – Temperate grasslands, shrublands, and savannas (Olson et al., 2001). Mean burned area of the last 12 years derived from Laurent et al. (2018). (b) Mean annual temperature (MAT) (Hijmans et al., 2005). (c,d) Precipitation during the extended austral (c) summer (November – March) and (d) winter (May – September) averaged from 1960 to 2021 using the data from Climatic Research Unit (Harris et al., 2020). CLLJ – Caribbean Low-level jets; ITCZ – Intertropical Convergence Zone; LLJ – Low-level jets; SACZ – South Atlantic Convergence Zone; SASM – South American Summer Monsoon; SWW – Southern Westerly Winds.

#### 3 Material and methods




## 3.1 Compilation of records and data treatment

The study area spans the Neotropical realm between latitudes 28°N to 60°S and longitudes 33°W to 105°W (Fig. 2a). Pollen records were gathered from the Neotoma (210 entries) and Pangaea (6 entries) databases (Felden et al., 2023; Williams et al., 2018). Charcoal data were gathered from the Reading (94 entries) (Harrison et al., 2022), Neotoma (16

entries), DRYAD (4 entries) (McMichael et al., 2021), and Pangaea (2 entries) databases. Additionally, we digitalized and extracted 39 pollen and 15 charcoal records from publications without openly available data, directly from published diagrams by using WebPloterDigitalizer (WebPlotDigitizer version 5.2) (Fig. 2b) (for the full list of datasets and respective citations and sources, see Supplementary Table 1). For NEB, charcoal influx data is represented by three available curves from the region derived from De Oliveira et al. (1999), Bouimetarhan et al. (2018), and Ledru et al. (2006). A subset of these records was grouped into seven subregions (*Sect. 2*), delimited by similarities in dominant climate features and vegetation (Fig. 1 and 2a):








Northern Neotropics (NNeo) comprises 26 pollen and 25 charcoal records collected north of 7 °N, mostly from Central America, excluding high montane sites above 3000 m. Most frequent taxa include trees and shrubs such as *Pinus*, Moraceae/Urticaceae, *Quercus*, *Alnus*, *Acalypha*, *Bursera*, and *Piper* and herbs such as Poaceae, Cyperaceae, Asteraceae, and Amaranthaceae.

Amazonia includes mostly records from the eastern, southwestern, and northern borders of the Amazon River drainage basin (28 pollen, 31 charcoal). Most common woody taxa include Moraceae, Alchornea, Melastomataceae, Podocarpus, Myrtaceae, Mauritia, Cecropia, Euterpe, and Ilex, and herbs such as Poaceae, Asteraceae, and Cyperaceae.

*Tropical Andes (TrAn)* comprises 83 pollen and 24 charcoal records located between 9 °N and 24 °S, from Andean sites located above 2200 m (average altitude of the records: 3590 m). Main pollen taxa include herbs such as Poaceae, Asteraceae, Cyperaceae, and *Plantago*, while the woody component includes taxa such as Melastomataceae, *Weinmannia, Hedyosmum*, Moraceae/Urticaceae, *Alnus, Podocarpus*, and Myrica.

Northeastern Brazil (NEB) mostly encompasses the Caatinga, including marine records off the northern Brazilian margin, under major influence of NEB sources. The dearth of pollen (7) and charcoal (3) records prevent detailed vegetation and fire regime assessment. Some of the main taxa found in pollen records include herbs such as Poaceae, Cyperaceae, Asteraceae, Borreria, and Amaranthaceae, and woody taxa such as Cuphea, Alchornea, Arecaceae, Moraceae/Urticaceae, Dalbergia, Schefflera, Myrsine, Mimosa, and Platymiscium.

*Central-eastern Brazil (CEB)* is represented by 18 pollen and 8 charcoal records in the southeastern Cerrado. Characteristic pollen taxa include herbs such as Poaceae, Cyperaceae, Asteraceae, Apiaceae, and *Borreria*, and woody taxa such as Myrtaceae, *Cecropia*, Moraceae/Urticaceae, *Myroxylon*, *Mauritia*, and Melastomataceae.

Southeastern South America (SESA) encompasses 21 pollen and 12 charcoal records within the modern extent of the Atlantic Forest. Most common herbaceous taxa are Poaceae, Cyperaceae, Asteraceae, Eryngium, and Xyris while characteristic woody taxa include Myrtaceae, Moraceae/Urticaceae, Alchornea, Myrsine, Arecaceae, Melastomataceae, Ericaceae, Podocarpus, and Araucaria.

Extratropical Andes (ExTrAn) comprises 47 pollen records and 18 charcoal records located south of 32°S. Characteristic taxa from this subregion include herbs such as Poaceae, Cyperaceae, Asteraceae, Apiaceae, Misodendrum, and Brassicaceae and woody taxa such as Nothofagus, Austrocedrus, Cupressaceae, Ericaceae, Empetrum, Myrtaceae, and Podocarpus.

Datapoints outside the defined subregions (black dots in Fig. 2a) were excluded from subregional analyses. These include records that are either geographically isolated or located outside subregional definitions, e.g., high montane sites from NNeo > 3000 m; low altitudes from TrAn < 2200 m.

For Neotoma records, arboreal pollen (AP), which serves as an indicator of tree cover relative to herbaceous vegetation, was calculated as the percentage of woody taxa (trees, shrubs, and palms), considering taxa at the genus and family level divided by the total sum of trees and shrubs, palms, and herbs, excluding mangrove and aquatic taxa, fern spores, and

unidentified types. We used the Neotoma standardized classification of ecological groups (Supplementary Table 1). For manually extracted records, AP percentages were obtained from published pollen diagrams. Therefore, the criteria used to construct these AP curves may slightly differ from those applied in our calculations based on raw data. For instance, in CEB, *Mauritia* and Cyperaceae are often excluded from AP calculations, as these taxa are often over-represented due to strong local imprint from palm swamp vegetation (Barberi et al., 2000; Escobar-Torrez et al., 2023; Salgado-Labouriau et al., 1997). To maintain consistency within this specific region, we also excluded *Mauritia* and Cyperaceae from AP calculations from CEB records. Samples with pollen counts below 100 grains were removed. Records from sites highly influenced by coastal dynamics (> 15 % of mangrove taxa) were also removed from the AP composites. Although tree ferns can serve as important indicators of humid conditions in regions such as the Andes and Atlantic Forests, these were excluded from pollen sums due to inconsistent reporting across records, to ensure a standardized dataset suitable for regional-scale analyses.

For charcoal composites, records containing both micro- (< 100 µm) and macro- (> 100 µm) particles were included. Charcoal raw counts were converted into concentrations and then to influx using site-specific sedimentation rates to account for differences in sedimentation rates across sites (Marlon et al., 2016). Changes in charcoal records can be linked to past fire activity and used to infer shifts in fire regimes (Power et al., 2008; Daniau et al., 2010; Marlon et al., 2016). While our approach does not allow to resolve specific components of the fire regime (e.g., intensity, severity, frequency, seasonality, spatial extent of burned vegetation), it allows to identify collective changes in fire recorded as changes in charcoal influx within a given region over a long timescale. New age models were calculated for Neotoma charcoal entries as per Harrison et al. (2022). We used the 'Bacon' R package (Blaauw and Christen, 2011) and the IntCal20 (Reimer et al., 2020) and SHCal20 (Hogg et al., 2020) calibration curves for latitudes above 15°N and below 15°S, respectively, and a 50:50 mixed calibration curve for intermediate latitudes.

Transformation and standardization of AP and charcoal influx data followed Power et al. (2008) using the 'paleofire' R package (Blarquez et al., 2014). These steps are essential for appropriate comparisons between records that use different quantification techniques and report different charcoal particle sizes (Power et al., 2008). It also allows for comparisons among sites with different vegetation settings (e.g., swamp, peat, lake, and marine records) by emphasizing trends over absolute values. As such, values were transformed and standardized with a Box-Cox transformation ( $\alpha = 0.01$ ) and 0-1 range rescaling and converted to z-scores using a common base period of 0.2–21 ka, so that all sites have a common mean and variance. Composite curves were constructed by fitting a locally weighted regression (LOWESS) curve to the pooled transformed and rescaled data, with confidence intervals (2.5<sup>th</sup> and 97.5<sup>th</sup> percentiles) generated through 1000 bootstrap replicates. A two-stage smoothing approach was applied using a pre-bin half width of 20 yr and a LOWESS smoothing of 1000 yr window half width to produce low resolution curves and a smoothing of 400 yr window half width for higher resolution curves (Daniau et al., 2012; Marlon et al., 2008). In general, caution is warranted when interpreting trends during periods with wide confidence intervals or when the composite curve approaches the upper or lower bounds of the confidence interval, or exhibits outlier shifts. These cases usually relate to periods with few records and indicate greater uncertainty and sensitivity to individual records, thus reflecting local variability of specific sites.

Recent studies have highlighted limitations of z-score scaling, including the distortion of charcoal peaks and the lack of consistency in representing fire absence across sites, particularly in tropical ecosystems where the documentation of both presence and absence is important (McMichael et al., 2021; Gosling et al., 2021). To assess the impact of methodological choices on the resulting curves, we also explored the alternative approach based on Proportional Relative Scaling (PRS) as per McMichael et al. (2021) (Supplementary Material). We produced curves using PRS and PRS applying a base period

(0.2 – 21 ka). We compare these two relative scaling-based curves with our z-score curves. Importantly, by applying the same base period to both PRS and z-score calculations, we ensured consistency across datasets for comparative purposes, although this approach was not assessed in McMichael et al. (2021).

Additionally, we produced maps displaying site-specific AP percentages and z-scores for both AP and charcoal influx for all data points to illustrate spatial patterns across time slices of the last 21 ka. Each site-specific data point represents a mean z-score calculated relative to its own long-term mean (base period: 21 to 0.2 kyr BP). As a result, the colors of the dots within a single map reflect deviations from local baseline conditions and should not be directly compared across sites to infer geographic gradients or absolute levels of fire activity or tree cover. However, clusters with similar z-score trends, such as consistently positive values in a region, may indicate homogeneous responses and coherent regional patterns, for example a general expansion in tree cover or intensification of the fire regime. For mapping purposes, the z-scores were divided into five equidistant categories: > +0.8 (strong positive anomalies), +0.8 to +0.4 (positive anomalies), +0.4 to -0.4 (weak positive or negative anomalies), -0.4 to -0.8 (negative anomalies), and 

Fig. 2 – Spatial and temporal distribution of analyzed records. (a) Distribution of pollen and charcoal records compiled from Neotoma, Pangaea, and Reading databases, and manually extracted from publications (Felden et al., 2023; Harrison et al., 2022; Williams et al., 2018). Details on the reference, site location, and maximum spanning age from each site are available in Supplementary Table 1. Black circles indicate sedimentary records that were not included in the subregional analysis. Elevations greater than 500 m are represented in light gray, while areas above 1500 m altitude are shown in dark gray. Major rivers are displayed as black lines. (b) Number of pollen and charcoal records available for each 400-year time bin over the last 21 ka, providing an overview of temporal data coverage.

# 3.2 Radiocarbon ages from archeological sites






We generated summed probability density (SPD) curves as a proxy for human occupation trends (Contreras and Meadows, 2014; Williams, 2012), based on compiled <sup>14</sup>C ages from archaeological sites available from literature (Araujo et al., 2025; Goldberg et al., 2016) and the Mesoamerican Radiocarbon database (MesoRad, 2020) (Fig. A1). Compiled data from Goldberg et al. (2016) are limited to ages older than 2 ka <sup>14</sup>C ages. This approach assumes a proportional relationship

between human populations and the production of datable material, as well as the statistical representativeness of the actual dated samples in relation to the full spectrum of <sup>14</sup>C samples from a region (Contreras and Meadows, 2014).

We produced SPD curves using 'rcarbon' R package, with 600-yr moving average window size to reduce the effects of the calibration process and 100-yr bins to account for potential biases associated to strong inter-site variability in sample size (Crema and Bevan, 2021). We used calibration curve SHCal20 (Hogg et al., 2020) for Southern Hemisphere <sup>14</sup>C ages (Amazonia, NEB, CEB, SESA, TrAn) and IntCal20 (Reimer et al., 2020) for Northern Hemisphere <sup>14</sup>C ages (NNeo). SPD curves are shown from the oldest non-zero values.

For NNeo, we use  $^{14}$ C ages available from MesoRad (2020) (N = 1692) and at latitudes northern of 7°N from Goldberg et al. (2016) (N = 96). For Andean regions, such as TrAn (N = 873) and ExTrAn (N = 621), we use data compiled by Goldberg et al. (2016). For Amazonia (N = 751), NEB (N = 542), CEB (N = 593), and SESA (N =1589), we use data compiled by Araujo et al. (2025).

### 3.3 Modern fire patterns extraction and analysis

To assess the current relationship between climate, fire activity, and vegetation parameters, we extracted data from the global fire patch functional traits database (FRY) (Laurent et al., 2018). The FRY map resolution was rescaled from  $0.002246^{\circ} \times 0.002246^{\circ}$  to grids of  $0.45^{\circ} \times 0.45^{\circ}$  (2500 km² at the equator) using 20 yr of satellite monitoring data (2001 to 2020). For each rescaled grid, we obtained the mean fire radiative power, as a measure of fire intensity, and total burned area. Fire radiative power was then compared with WorldClim climate variables (Fick and Hijmans, 2017) (originally 10 minutes spatial resolution,  $0.1667^{\circ} \times 0.1667^{\circ}$ )—including mean annual temperature (MAT), maximum temperature of the warmest month, annual precipitation, precipitation of the driest quarter, and precipitation seasonality—and major vegetation types defined by ecoregions (Olson et al., 2001) (Fig. 3). For consistency, the climate models and ecoregion distribution were rescaled to match the FRY fire map.

#### 4 Results








#### 4.1 Modern fire, climate and vegetation relationship

Modern vegetation and fire patterns correlate with climate (Fig. 3). Tropical savannas are predominantly distributed in seasonal climates with annual precipitation ranging from 1000 to 2000 mm yr<sup>-1</sup> and mean annual temperature between 20 and 27 °C (Fig. 3a). Towards wetter climates, the conditions for the existence of savannas overlap with those of tropical moist forests, while towards drier climates, they overlap with those of xeric vegetation (Fig. 3a,b).

Regarding fire activity, tropical moist forests and xeric vegetation exhibit weaker fire activity in relation to savannas (Fig. 3c,d). Fire activity also weakens towards cooler subtropical and temperate savannas and montane grasslands. Fire radiative power, which is linearly correlated with the total burned area (Fig. 3e), is more intense in tropical savannas, specifically at intermediate annual rainfall levels (875–2000 mm yr<sup>-1</sup>, peaking at ca. 1520 mm yr<sup>-1</sup>, Fig. 3c) and precipitation under 100 mm during the driest month (Fig. 3d). Fire intensity rises with mean annual temperatures above 21 °C, reaching maximum values as temperatures rise (Fig. 3c). Regions with maximum temperatures of the warmest month above 29 °C concentrate most of the fire activity (Fig. 3d). The fire season in tropical America occurs during spring (i.e., March to May in the northern tropics and August to October in the southern tropics) (Fig. 3f), while along the southwestern flank of the Andes the fire season occurs during austral autumn.

Fig. 3 – Climate, vegetation, and fire patterns across the Neotropics. (a-b) Distribution of ecoregions (Olson et al., 2001) and (c-e) fire radiative power (2001–2020) (Laurent et al., 2018) across the climate space. (a,c) Mean annual temperature × annual precipitation; (b,d) maximum temperature of the warmest month × precipitation of the driest month. Black dots denote records compiled in this study. (e) Relationship between total burned area and mean fire radiative power, averaged per 2500 km² grid cell. (f) Monthly relative burned area for the Northern and Southern Hemisphere regions of the Neotropics from MODIS (Fire Information for Resource Management System – FIRMS: <a href="https://firms.modaps.eosdis.nasa.gov/">https://firms.modaps.eosdis.nasa.gov/</a>).

# 4.2 Pollen and charcoal records





# 4.2.1 Representativeness of the dataset

The existing records are distributed across the main subregions of the Neotropics, ranging from the moist tropical forests to the semiarid xeric vegetation (Fig. 1 and 2). While the Andes are particularly well sampled (Fig. 2a), areas such as central Amazonia, arid and semiarid regions (e.g., northeastern Brazil), and central regions of the Neotropics located between 20 and 40°S (Fig. 2a) are underrepresented.

In terms of chronological representation, site density increases towards the Holocene (Fig. 2b). Across the Neotropics, considering time bins of 400-yr intervals, the LGM (21–19 ka) is supported by an average of 30 pollen and 20 charcoal records, followed by the deglacial period (19–11.7 ka; 62 and 35, respectively), and the Holocene (

**Fig. 4** – **Arboreal pollen and charcoal influx trends.** (a-g) Arboreal pollen and (h-m) charcoal influx z-score values derived from compilations for studied subregions. Green and red curves: 1000-yr window smoothed, black curves: 400-yr window smoothed. Brown curves show the number of records by 400-yr bins. Gray areas represent 2.5<sup>th</sup> and 97.5<sup>th</sup> confidence intervals. Note: Enough charcoal records from Northeastern Brazil were not available to generate a composite curve. Large charcoal anomalies that extend beyond +2 or -2 are indicated by circled arrows, which also depict low confidence periods.

# 5 Discussion

365

370

# 5.1 Modern climate-fire-vegetation relationships

The most intense fire activity occurs in the warm regions that combine intermediate annual precipitation levels (875–2000 mm yr<sup>-1</sup>) and a marked dry season (less than 100 mm in the driest month, Fig. 3c-e). Sufficient moisture supports biomass

growth, while the dry season grants the necessary flammability for fire events (Fig. 3c-e). These fire-prone conditions are typical of tropical savannas and grasslands (Fig. 3a,b), such as in CEB, where a regular frequency of fire events is key in maintaining biodiversity and the physiognomy of vegetation (Bernardino et al., 2022; Mistry, 1998). In modern day CEB, however, fire regime is mostly limited by fuel moisture (moisture-limited condition, negative biomass-fire correlation) (Alvarado et al., 2020). In arid and semiarid environments, such as NEB, or high-altitude areas, such as higher areas of TrAn, fire activity is hindered due to biomass limitation (fuel-limited conditions, positive biomass-fire correlation) (Fig. 3a-e). On the other hand, under wet conditions of tropical rainforests, such as Amazonia or parts of NNeo, fire activity is limited due to constant fuel moisture (moisture-limited condition) (Fig. 3a,b). Natural and anthropogenic wildfires are more frequent at the dry-wet season transition, when biomass flammability is at its highest, which typically corresponds to the austral spring in Southern Hemisphere tropical regions (Fig. 3f) (Mistry, 1998; Ramos-Neto and Pivello, 2000). The modern fire regime has been heavily modified by human activity through the intensification of wildfires burning both fire-adapted and fire-sensitive vegetation (Argibay et al., 2020; Hantson et al., 2015; Pivello, 2011). Despite this limitation, insights can still be obtained on the feedbacks between fire, vegetation, and climate (Fig. 3a-e) as also suggested by global and local fire analyses, which show a combined climate and anthropogenic control on fire (Hantson et al., 2015; Kitzberger et al., 2022), reinforcing the interconnectedness of these factors.

## 5.2 Vegetation and fire regime changes over the last 21 ka

# 5.2.1 Northern Neotropics (NNeo)

During the LGM, high levels of tree cover and weak fire regimes (Fig. 5d,e) are consistent with estimates of 4–5°C drop in mean annual temperatures (Correa-Metrio et al., 2012) and wet conditions (Hodell et al., 2008; Deplazes et al., 2013; Fig. 5a,c). Throughout the deglaciation, a marked decrease in tree cover and intensification of the fire regime relate to predominant drier phases promoted by southward displacements of the ITCZ associated with millennial-scale events (HS1 and the YD) (Fig. 5b,c) (Deplazes et al., 2013; Haug et al., 2001). The wetter interval linked to BA/ACR (Deplazes et al., 2013; Fig. 5b) is not clearly detected in our analyses. Additionally, human occupation in the NNeo began during the late Pleistocene (Ardelean et al., 2020), although in smaller populations (Fig. 5d), and likely started contributing to environmental changes. These combined changes importantly affected megafaunal populations in NNeo, yet the implications of their decline and subsequent extinction for fire and vegetation dynamics remain unclear (Dávila et al., 2019; Rozas-Davila et al., 2021). Nevertheless, late Pleistocene tree cover and fire dynamics suggest predominant hydroclimate control (Fig. 5b-f) and an overall negative correlation between tree cover and fire activity (Fig. A2a) suggests moisture-limited conditions for fire.

At the onset of the Holocene, during the EH, a marked shift towards expansion of tree cover and minimum fire activity (Fig. 5 e,f; Fig. A2a) were induced by increasing wetter conditions (Haug et al., 2001; Hodell et al., 2008). During this period, human populations began engaging in agricultural activities, domesticating maize and squash by ca. 9 ka (Piperno et al., 2009), likely impacting local fire regimes. However, the gradual long-term tree cover decrease and fire activity increase synchronous with a progressive transition to drier conditions (Haug et al., 2001), still suggest a main climatic driver from the EH until ca. 4 ka (Fig. 5b,e,f). From ca. 4 ka onwards, the expanding anthropogenic pressure in Central America became a clear driver of vegetation and fire changes (Fig. 5d) (Harvey et al., 2019; Leyden, 2002). The LH drop in tree cover decouples from the more gradual climate-driven decrease observed since the onset of the Holocene (Fig. 5a,e; Fig. A2a). This latter shift in AP is coeval with the demographic expansion of Mesoamerican populations (Fig. 5d). Fire

activity also stays relatively high during this period (Fig. 5f). At ca. 1.2 ka, the increase in tree cover and decrease of fire activity was possibly related to the rapid decrease in populations likely associated with the collapse of the Maya civilization (Gill et al., 2007; Haug et al., 2003).

Fig. 5 – Northern Neotropics (NNeo) vegetation, fire, climate regimes, and human populations: (a) Pollen-based mean annual air temperature reconstruction from Petén-Itzá core PI-06 (Correa-Metrio et al., 2012). (b) Bulk sediment Ti content (Haug et al., 2001) and (c) bulk sediment reflectance (Deplazes et al., 2013) for Cariaco Basin. (d) Summed probability density (SPD) of <sup>14</sup>C ages from archeological sites in Central America (MesoRad, 2020) and northern South America (Goldberg et al., 2016) (N = 1788). (e) Arboreal pollen (AP) and (f) charcoal influx z-scores composites using 1000-yr (green and red, respectively) and 400 yr (black) smoothing half-window. Gray areas represent 2.5<sup>th</sup> and 97.5<sup>th</sup> confidence intervals. (g) Number (#) of records with available pollen (green) and charcoal (red) data in a 400-yr time bin.

## 5.2.2 Amazonia




The gigantic extent of Amazonia and its equatorial positioning results in heterogeneous meridional and zonal environmental patterns for the region. Limited number of records in the LGM and deglacial period results in poor spatial and temporal constraints of tree cover and fire activity variability, thus preventing a generalization and detailed assessment of the observed patterns (Fig. 6f). Additionally, caution is needed when interpreting vegetation changes from fluvial and floodplain records, as edaphic factors, rather than climate, may control the observed patterns.

During the LGM, the region featured reduced tree cover, mainly at the ecotones and eastern areas, and weak fire activity (Fig. 6e,f), although its western and core regions remained mostly forested (Akabane et al., 2024; Colinvaux et al., 1996; Haberle and Maslin, 1999; Urrego et al., 2005). This pattern of low tree cover was likely a response to 4–6 °C colder

temperatures (Bush et al., 2001; Colinvaux et al., 1996; Stute et al., 1995), low CO<sub>2atm</sub> (Fig. 6a) (Bereiter et al., 2015), and reduced rainfall in eastern Amazonia (Fig. 6b,c) (Häggi et al., 2017; Wang et al., 2017). Reduced fire activity, despite low tree cover, may also have been a consequence of colder conditions and reduced convective activity, which would result in fewer lightning ignitions, and absence of widespread human impacts.





During the deglacial period, our data indicate oscillating tree cover patterns during HS1 and highest deglacial AP values by 13–12 ka, coinciding with minimum fire activity (Fig. 6e,f), an intensified SASM (Cheng et al., 2013; Mosblech et al., 2012) and increased CO<sub>2atm</sub> (Fig. 6a,b). These pronounced oscillations in AP and charcoal z-scores may arise from contrasting meridional changes in precipitation patterns associated with millennial-scale events. For instance, during HS1 and the YD, while drier conditions expanded over northern Amazonia (Akabane et al., 2024; Deplazes et al., 2013; Zular et al., 2019), southern areas experienced wetter conditions (Campos et al., 2019; Mosblech et al., 2012; Novello et al., 2017). The subsequent EH decline in AP was likely driven by a weakening of the SASM and the stepwise increase in fire activity (Fig. 6e,f), which was mainly concentrated to the eastern and southern areas (Gosling et al., 2021) and may have played an important role in reducing tree cover over ecotones. Moreover, the end of the Pleistocene marks the beginning of human occupation of the Amazon basin, which increased in the EH (Fig. 6d), when they were already engaging in resource management practices (Neves et al., 2021). The impacts of megafaunal extinction also initiated long-term changes in both nutrient distribution and species turnover (Doughty et al., 2013, 2016a). However, its correlation with overall tree cover and fire regime changes remains elusive for the region.

During the MH, a stepwise increase in tree cover is recorded at 7 ka, mostly reflecting forest expansion in northern Amazonia (Behling and Hooghiemstra, 2000) and coinciding with monsoon strengthening (Fig. 6b,e). Meanwhile, some increase in fire activity, amid forest expansion and progressively wetter conditions, may relate to expanding human populations in Amazonia (Cordeiro et al., 2014; Gosling et al., 2021; Riris and Arroyo-Kalin, 2019). The absence of a clear decrease in tree cover as consequence of anthropogenic activity is likely due to agroforestry practices that allowed for long fallow periods and forest recovery (Iriarte et al., 2020).

In the LH, gradual tree cover expansion is mainly driven by the southward forest expansion that formed the modern extent of the rainforest (Fontes et al., 2017; Mayle et al., 2000) associated with a further intensification of monsoon strength (Baker and Fritz, 2015) and moisture increase. Some increase in fire activity after ca. 3 ka has been potentially favored by dryness associated to intensifying ENSO activity (Kanner et al., 2013; Mark et al., 2022) in combination to expanding human populations (Fig. 6d). The decline in fire activity over the last 0.5 ka coincides with the indigenous demographic collapse following the European contact (Fig. 6d,f).

**Fig. 6** – **Amazonia vegetation, fire, climate regimes, and human populations: (a)** Atmospheric concentration of CO<sub>2</sub> (Bereiter et al., 2015). Speleothem δ<sup>18</sup>O from **(b)** El-Condor (ELC), Cueva del Diamante (NAR), and Santiago (San) caves in western Amazonia (Cheng et al., 2013; Mosblech et al., 2012) and from **(c)** Paraíso cave in eastern Amazonia (Wang et al., 2017). **(d)** Summed probability density (SPD) of <sup>14</sup>C ages from archeological sites (N = 732) (Araujo et al., 2025). **(e)** Arboreal pollen (AP) and **(f)** charcoal influx z-scores composites using 1000-yr (green and red, respectively) and 400 yr (black) smoothing half-window. Gray areas represent 2.5<sup>th</sup> and 97.5<sup>th</sup> confidence intervals. **(g)** Number (#) of records with available pollen (green) and charcoal (red) data in a 400-yr time bin.

#### 5.2.3 Tropical Andes (TrAn)






Climate changes in the Andes were heterogeneous and asynchronous (Bush and Flenley, 2007). During the LGM, predominating open vegetation (Fig. 7e) was likely controlled by 5–8 °C cooler-than-modern temperatures exerting the main limiting factor for tree cover development, even though moist conditions prevailed (Baker et al., 2001; Bush et al., 2004; Cheng et al., 2013; Paduano et al., 2003; Valencia et al., 2010) (Fig. 7a,b). Subsequently, TrAn saw an increase in arboreal taxa from 18 to 14.8 ka, when the region became warmer and significantly wetter (Fig. 7a) (Martin et al., 2018; Palacios et al., 2020). Despite uncertainties related to the limited number of records, a slight fire increase in the second part of HS1 may have resulted from higher biomass availability (Fig. 7d). From 14 to 12.9 ka, pervasive decrease in tree cover throughout the BA/ACR until the onset of the YD likely relate to drier conditions in the southern TrAn and overall glacier expansion (Jomelli et al., 2014), which is then followed by a tree cover recovery throughout the YD potentially associated to further warming and the return of wetter conditions (Fig. 7a,d). Positive correlation of increasing fire activity throughout the LGM and deglacial period along with increasing trends of tree cover (Fig. A2c) point to fuel-limited conditions for fire, although prevailing wet conditions during the late Pleistocene may have also played a key role in restraining fire (Fig. 7a,b).

During the EH, tree cover achieved its highest elevation during the Holocene (Fig. 7a). This tree cover increase was likely maintained by warmer temperatures and rising of the tree line, despite relatively drier conditions in response to a weaker

SASM, as suggested by  $\delta^{18}$ O data from ice cores and speleothems (Cheng et al., 2013; Thompson et al., 1998; Vuille et al., 2003) (Fig. 7a). As in NNeo and Amazonia, evidence for human agricultural activities in TrAn began during this period (Pagán-Jiménez et al., 2016). Additionally, an overall decline of megafauna in tropical Andes resulted in sensitive ecological consequences associated with vegetation turnover, e.g., the encroachment of both palatable and woody taxa, as well as fuel build-up (Bush et al., 2022; Pym et al., 2023). Several interacting factors, such as warming, hydroclimate changes, initial expansion of human populations, and megafaunal decline, collectively contributed to intensifying the fire regime in the region (Bush et al., 2022; Pym et al., 2023) (Fig. 7f). Consequently, increased fire activity hampered further development of a continuous tree cover above the tree line resulting in a sharper vegetation boundary (Rehm and Feeley, 2015). This suggests a complex interplay of abiotic and biotic drivers shaping the observed pattern.








The MH is characterized by a stepwise intensification of the fire regime, with tree cover only showing a slight decreasing trend, as indicated by our regional-integrating AP composite (Fig. 7d,e). However, subdividing TrAn at 8 °S in northern and southern sectors reveals distinct trends (Fig. A3): while fire activity increased across both regions, tree cover remained stable in the north but declines in the south. These trends are consistent with heterogeneous hydroclimatic patterns along Andes, with drier conditions recorded by lowered lake levels, mainly in the Altiplano (Baker et al., 2001; Bush and Flenley, 2007; Hillyer et al., 2009) and persistent wet conditions recorded in central sectors of the TrAn (Bustamante and Panizo, 2016; Cheng et al., 2013; Polissar et al., 2013) (Fig. 7a). Modern climate patterns indicate that anomalies in equatorial Pacific SST can produce different regional impacts in precipitation by affecting moisture-laden easterlies and the Bolivian High (Garreaud et al., 2003; Poveda et al., 2020; Vuille, 1999). For instance, reduced ENSO variability has been suggested to decrease moisture balance in parts of the Andes, particularly in the southern TrAn, parts of the northern TrAn and NNeo (Fig. A3; Fig. 7c) (Polissar et al., 2013). Accordingly, a potential correspondence between the zonal Pacific SST gradient anomaly and tree cover trends during the EH and MH may exist for southern TrAn but not for northern TrAn (Fig. A3). Furthermore, while  $\delta^{18}O$  in ice and speleothem cores and  $\delta D$  from lake sediments primarily reflect rainy season precipitation (Cheng et al., 2013; Fornace et al., 2014; Vuille et al., 2003), fluctuations in lake levels are influenced by annual precipitation (Theissen et al., 2008). This may alternatively suggest that despite a gradual increase in summer precipitation throughout the Holocene, the MH featured a decrease in annual precipitation over the Altiplano. Nevertheless, the maintenance of tree cover was also favored by the persistence of moist microclimates (Ledru et al., 2013; Nascimento et al., 2019). These conditions, as well as anthropogenic activities, would have favored a regional intensification of the fire regime and decline in tree cover. By ca. 6 ka, agropastoral systems based on maize agriculture and llama herding became widespread (Nascimento et al., 2020).

The LH marks a major decrease in tree cover and high fire activity in the whole TrAn (Fig. 7c), probably related to the expansion of human impacts in the region and increasing sedentism (Goldberg et al., 2016; Valencia et al., 2010). Anthropogenic activity likely maintained fire activity at similar rates to the previous MH dry phase (Fig. 7f) despite moisture increase, maintaining a lowered and sharper tree line (Schiferl et al., 2023). During the last 0.5 ka, tree cover expansion and decreased fire activity may point to the abandonment of sites as consequence of the European contact and demographic collapse in the region (Koch et al., 2019).

Fig. 7 – Tropical Andes (TrAn) vegetation, fire, climate regimes, and human populations: (a) Speleothem δ<sup>18</sup>O from El-Condor (ELC), Cueva del Diamante (NAR), and Santiago (San) caves (Cheng et al., 2013; Mosblech et al., 2012). (b) Freshwater benthic diatom (%) from Titicaca Lake (Baker et al., 2001). (c) Summed density probability of <sup>14</sup>C ages from archeological sites in TrAn (Goldberg et al., 2016) (N = 949). (d) Arboreal pollen (AP) and (e) charcoal influx z-scores composites using 1000-yr (green and red, respectively) and 400 yr (black) smoothing half-window. Gray areas represent 2.5<sup>th</sup> and 97.5<sup>th</sup> confidence intervals. (f) Number (#) of records with available pollen (green) and charcoal (red) data in a 400-yr time bin.

# 5.2.4 Northeastern Brazil (NEB)





In NEB, the scarcity of records decreases the precision and accuracy of the observed trends (Fig. 8f) and prevents producing a composite curve using charcoal data. Therefore, we only discuss major features of tree cover and fire dynamics.

During the LGM, the region experienced low tree cover driven by prevailing dry conditions (Cruz et al., 2009; Dupont et al., 2001; Mulitza et al., 2017). The deglaciation period is marked by tree cover expansion, associated with phases of southward displaced ITCZ during HS1 and YD (Fig. 8a,b,d) (Bouimetarhan et al., 2018; Cruz et al., 2009; Dupont et al., 2001; Mendes et al., 2019; Venancio et al., 2020). These periods featured the onset of forest corridors connecting Amazon and Atlantic forests and decreased fire activity during YD (Bouimetarhan et al., 2018; Dupont et al., 2009; Ledru and de Araújo, 2023; De Oliveira et al., 1999), which is depicted as the interval with highest AP z-scores over the last 21 ka (Fig. 8d).

During the EH, decrease in tree cover is likely a consequence to both climate changes and increasing human impacts. This period features drier conditions relative to YD in northern NEB due to the northward displacement of the ITCZ (Fig. 8a) (Mendes et al., 2019; Prado et al., 2013a; Venancio et al., 2020) and the expansion of human populations in the region (Araujo et al., 2025) (Fig. 8c). In the subsequent MH period, increasing trends in tree cover (Fig. 8d) may reflect relatively wet conditions due to a weak Nordeste Low (Cruz et al., 2009; Prado et al., 2013b) (Fig. 8b) and southward shifted seasonal migration range of the ITCZ (Chiessi et al., 2021), in addition to decreased human populations in inland parts of NEB (Araujo et al., 2025) (Fig. 8c). The LH was marked by a decrease in tree cover and intensification of the fire regime, likely in response to the establishment of the modern semiarid conditions over most NEB (Chiessi et al., 2021; Cruz et al., 2009; Utida et al., 2020) and rapid increase in human populations (Fig. 8c) (Araujo et al., 2025).

The relationship between tree cover and fire in the NEB remains elusive due to the limited number of records (Fig. 8e). Charcoal records suggest a negative correlation between tree cover and fire (Fig. 8f). This observation, however, is apparently counterintuitive, given that Caatinga currently exhibits the opposite pattern (fuel-limited conditions), where the lack of fuel inhibits fire activity (Argibay et al., 2020) (Fig. 4a,c). We propose two potential explanations for this apparent discrepancy. First, the fire records are located at the margins of the semiarid region, thus capturing influences from adjacent tropical savannas and forests, rather than exclusively reflecting xeric vegetation-fire dynamics. Second, increasing human occupation during the LH (Fig. 8c) may have intensified burning activities, even as natural fire frequencies declined. While NEB contains the earliest evidence of human occupation of the Americas (Boëda et al., 2014), only the LH saw a proliferation of more sedentary, possibly agricultural, ceramic-producing societies (Oliveira, 2002).

Fig. 8 – Northeastern Brazil (NEB) vegetation, fire, climate regimes, and human occupation: (a)  $\delta D$  n- $C_{29}$  alkane from GeoB16202-2 (Mulitza et al., 2017). (b) Speleothem  $\delta^{18}O$  from Rio Grande do Norte cave (RN) (Cruz et al., 2009). (c) Summed density probability of  $^{14}C$  ages from archeological sites in NEB (N = 542) (Araujo et al., 2025). (d) Arboreal pollen (AP) z-scores composites using 1000-yr (green) and 400 yr (black) smoothing half-window. Gray areas represent 2.5th and 97.5th confidence intervals. (e) charcoal influx z-scores from single sites (yellow: Ledru et al., 2006; brown: De Oliveira et al., 1999; red: Bouimetarhan et al., 2018). (f) Number (#) of records with available pollen data in a 400-yr time bin.

#### 5.2.5 Central-eastern Brazil (CEB)

During the LGM, tree cover and fire activity exhibit high trends; however, the scarcity of records hampers a regional generalization and underscores the need for additional data from this period (Fig. 9f). During the deglacial period, high tree cover and low fire activity trends were favored by periods of intensified rainfall in the region, i.e., HS1 and the YD (Fig. 9a,b,d,e) (Campos et al., 2019; Martins et al., 2023; Meier et al., 2022; Stríkis et al., 2015, 2018), which allowed the widespread migration of cold- and moist-adapted tree taxa through central Brazil (Pinaya et al., 2019). An increase in fire activity centered in ca. 16.5 ka may result from a short dry incursion during HS1 (Stríkis et al., 2015), which has yet to be confirmed, as the scarcity of data prevents detailed assessment of the observed pattern (Fig. 9e).

Our compilation suggests a prevailing decrease in tree cover and intensification in fire activity from the YD to the EH (Fig. 9d,e). These trends were likely a response to a weaker SASM/SACZ led by low austral summer insolation, which yielded drier conditions in the region (Cruz et al., 2005; Prado et al., 2013a; Wong et al., 2023). The stepwise increase in fire activity, peaking at ca. 10.5 ka, further contributed to the tree cover rapid decrease (Fig. 9d,e). Additionally, human activity probably also contributed to tree cover and fire trends during this period, as archaeological records show well-established occupations from ca. 13 ka onwards and expanding population in the EH (Fig. 9c) (Araujo et al., 2025; Strauss et al., 2020). Furthermore, the megafauna functional extinction during this period likely initiated long-term changes in vegetation composition (Raczka et al., 2018) and potential increase of fuel loads (as in Gill, 2014). This effect, however, seems to have been secondary, as a decrease in tree cover is opposite to the expected response by megafaunal extinction alone (Doughty et al., 2016b; Macias et al., 2014).

Throughout the MH and the LH, the progressive increase in rainfall driven by a progressive strengthening of the SACZ (Cruz et al., 2005; Meier et al., 2022) (Fig. 9b) favored the expansion of tree cover and the attenuation of fire activity (Fig. 9d,e). In the LH, despite a second wave of human impacts in the region after an occupation hiatus in the MH (Araujo et al., 2005, 2025), tree cover exhibits a continuous increase and fire activity a decreasing trend (Fig. 9c,d,e).

In CEB, long-term tree cover changes are negatively correlated with fire activity (Fig. 9d,e, Fig. A2d). This pattern points to a feedback mechanism in which herbaceous vegetation facilitates fire activity, while fire, in turn, contributes to the dominance of herbs by limiting tree encroachment. Conversely, moisture-driven development of woody formations leads to the suppression of fire, which further contributes to tree cover expansion (Fig. 9d,e).

Fig. 9 – Central-eastern Brazil (CEB) vegetation, fire, climate regimes, and human occupation: (a) Speleothem  $\delta^{18}$ O from Lapa sem Fim (LSF) and Lapa Grande (LG) caves (Stríkis et al., 2011, 2018). Downcore  $\ln(K/Al)$  from marine sediment core M125-35-3, which reflects changes in the clay mineral composition and increases with chemical weathering intensity and hence, moisture availability (Meier et al., 2022). (b) Summed density probability of  $^{14}$ C ages from archeological sites in CEB (N = 481). (c) Arboreal pollen (AP) and (d) charcoal influx z-scores composites using 1000-yr (green and red, respectively) and 400 yr (black) smoothing half-window. Charcoal z-score negative anomaly reaching -2.2 is indicated by a circled arrow. Gray areas represent 2.5<sup>th</sup> and 97.5<sup>th</sup> confidence intervals. (e) Number (#) of records with available pollen (green) and charcoal (red) data in a 400-yr time bin.

#### 5.2.6 Southeastern South America (SESA)

During the LGM, the low arboreal pollen z-scores reflect the dominating open physiognomies (Fig. 10c) (Behling, 2002b; Gu et al., 2018), despite moist conditions sustained by a strong SASM/SACZ influence (Cruz et al., 2005). This predominating open vegetation was likely controlled by 3–7 °C lower temperatures (Behling, 2002a; Chiessi et al., 2015) and reduced CO<sub>2atm</sub> levels (Fig. 10a,b) (Bereiter et al., 2015). Moreover, stronger Antarctic cold fronts may have shifted the woody savanna and forest boundaries further north, favoring the prevalence of open grasslands. The presence of a diverse megafauna potentially contributed by restricting both woody encroachment and fuel accumulation (Furquim et al., 2024; Macias et al., 2014; Prates and Perez, 2021; da Rosa et al., 2023). The combination of low woody biomass, higher impact of herbivory, cold and moist conditions restricted fire activity during this period (Fig. 10f). The long-term and gradual trend of tree cover expansion and fire intensification from the LGM to the EH suggests that scarcity of fuel was a limiting factor for fire activity during the Pleistocene (positive tree cover-fire correlation) (Fig. 10e,f, Fig. A2e). Alternatively, or concurrently, both trends may have been driven by progressive deglacial warming. Notably, our composite curves of tree cover and fire activity show no clear correlation with millennial-scale hydroclimate variability throughout most of the late Pleistocene, suggesting that moisture availability was not the primary driver, with temperature and CO<sub>2atm</sub> exerting greater control. However, Campos et al. (2019) suggested that no generalized increase in precipitation occurred in SESA during

HS1 and the YD. Furthermore, while human occupation began at ca. 14-13 ka in the region, impacts were likely still limited (Araujo et al., 2025; Araujo and Correa, 2016; Suárez, 2017).






During the EH, a peak in fire activity (Fig. 10f) was facilitated by the availability of biomass coupled with warmer temperatures and relatively drier conditions (Fig. 10b,c,e,f). The intensified fire regime, on the other hand, probably contributed to the retraction of forest formations during this period (Fig. 10e,f). Throughout the Holocene, a continuous increase in arboreal biomass decoupled from CO<sub>2atm</sub> and temperature trends suggests a forest expansion mainly driven by increasing precipitation and the gradual suppression of fire activity (Fig. 10c,e,f). The extinction of megafauna by the end of the EH potentially contributed, at least in part, to the expansion of woody taxa (Behling et al., 2023; Macias et al., 2014). From 6 to 3.5 ka, the continuous expansion of tree cover is slowed down, coinciding with a peak in fire activity and expanding human populations in the region (Fig. 10d-f). In the last 2 ka, despite an increase in human populations, tree cover and fire activity do not exhibit a clear response.

The Pleistocene to Holocene transition likely represents a transition from temperature/ $CO_{2atm}$ -limited to moisture-limited conditions, when an inflexion in the correlation of tree cover and fire is observed (Fig. A2e). Over the 21-kyr analyzed period, tree cover and fire activity show a positive correlation with both increasing in the long-term but with a negative correlation on detrended data (r = -0.48, p 

Fig. 10 – Southeastern South America (SESA) vegetation, fire, climate regimes, and human occupation: (a) Atmospheric concentration of CO<sub>2</sub> (Bereiter et al., 2015). (b) Estimated mean annual temperature (Chiessi et al., 2015). (c) Speleothem  $\delta^{18}$ O from Botuverá cave (Wang et al., 2007). (d) Summed density probability of  $^{14}$ C ages from archeological sites in SESA (N = 1701) (Araujo et al. 2025). (e) Arboreal pollen (AP) and (f) charcoal influx z-scores composites using 1000-yr (green and red, respectively) and 400 yr

(black) smoothing half-window. Gray areas represent 2.5<sup>th</sup> and 97.5<sup>th</sup> confidence intervals. **(g)** Number (#) of records with available pollen (green) and charcoal (red) data in a 400-yr time bin. Large charcoal negative anomaly extending beyond -2 is indicated by a circled arrow.

#### 5.2.7 Extratropical Andes (ExTrAn)









During the LGM, ExTrAn was characterized by open physiognomies and a weak fire regime (Fig. 11e,f). This was likely consequence of significantly cold conditions (Fig. 11a,b) (Massaferro et al., 2009), while both wet and dry conditions have also been suggested along ExTrAn (Montade et al., 2013; Moreno et al., 2018). After ca. 18 ka, tree cover increases along with warming temperatures (Kaiser et al., 2005) and rising CO<sub>2atm</sub> (Fig. 11a,b), with an accelerated tree cover increase during HS1 coinciding with a warming phase and retreating glaciers in the region (Fig. 11b,e,f) (Barker et al., 2009; Kaiser et al., 2005; Moreno et al., 2015; Palacios et al., 2020). This warming period also coincides with evidences of human presence in the region (Dillehay et al., 2015). Fire activity features a stepwise increase from 13.8 to 12 ka, corresponding to the second deglacial warming phase (Fig. 11a,b,e), under decreasing, albeit still high, precipitation levels (Montade et al., 2019) and drier summers (Moreno, 2020). This threshold in the fire regime was likely induced by warming temperatures and availability of biomass. The positive correlation between woody biomass and fire activity supports a fuel-limited fire regime in the region, and/or suggests that observed increases in tree cover and fire activity were both driven by deglacial warming (Fig. A2f). Nevertheless, peak fire activity in the region is currently achieved at intermediate levels of both woody and herbaceous biomass (Holz et al., 2012), supporting the role of increasing woody vegetation in creating optimal conditions for fire. This is also suggested by the relevant contribution of wood charcoal in lakes from the region (Whitlock et al., 2006). Human populations only started to expand after the ACR and likely also contributed to the intensification of the fire regime (Fig. 11d) (Perez et al., 2016; Salemme and Miotti, 2008). At ca. 13-12 ka, Nothofagus emerges as the predominant taxon and marks the widespread expansion of temperate forests. This expansion exhibited latitudinal variability, with a gradual trend in southern areas and a steeper increase after 15 ka in northern regions (Nanavati et al., 2019). As a result of combined human pressure and environmental changes, the extinction of megaherbivores could have contributed to arboreal growth and fuel accumulation; however, the extent of this impact is still unclear (Abarzúa et al., 2020; Villavicencio et al., 2016).

During the EH, peak fire activity potentially contributed to the deacceleration of tree cover expansion (Fig. 11d,e), which were both favored by a decrease in moisture due to the weakened and poleward shifted Southern Westerly Winds (SWW) (Lamy et al., 2010; Moreno et al., 2021; Nehme et al., 2023) (Fig. 11c) coupled with relatively warmer conditions. The EH climate amelioration also allowed human populations to spread and colonize other localities in the region (Miotti and Salemme, 2003; Perez et al., 2016; Salemme and Miotti, 2008), expanding its contribution to increased fire activity. In the MH, the expansion of forests and decrease in fire activity (Fig. 11e,f) was potentially driven by increased moisture due to a strengthening and equatorward shift of the northern boundary of the SWW (Fig. 1c,d) (Razik et al., 2013; Villa-Martínez et al., 2003) and reduced ENSO (Mark et al., 2022; Rein et al., 2005). Unlike other regions of South America (e.g., TrAn, CEB, NEB), the MH marks a rapid growth in human populations in ExTrAn, which continued through the LH and may have benefited from expanding forests (Fig. 11e) and consequent increase in availability of resources. By 5.5 ka, the region experiences the highest tree cover, whereas fire activity attains peak values after 3.5 ka, concurrent with a major expansion of human populations (Perez et al., 2016) and intensification of ENSO variability (Mark et al., 2022; Rein et al., 2005; Whitlock et al., 2006).

Fig. 11 – Extratropical Andes (ExTrAn) vegetation, fire, climate regimes, and human occupation: (a) Surface mean annual temperature anomaly reconstructed from EPICA Dome C, Antarctica (Jouzel et al., 2007). (b) Sea surface temperature reconstruction for the eastern South Pacific (Kaiser et al., 2005). (c) Rainfall estimates from Lake Aculeo (34°S; northern portion of the Southern Westerly Winds) based on multiproxy lake-level reconstructions (Jenny et al., 2003). (d) Summed density probability of <sup>14</sup>C ages from archeological sites in ExTrAn (N = 621) (Goldberg et al., 2016). (e) Arboreal pollen (AP) and (f) charcoal influx z-scores composites using 1000-yr (green and red, respectively) and 400 yr (black) smoothing half-window. Gray areas represent 2.5<sup>th</sup> and 97.5<sup>th</sup> confidence intervals. (g) Number (#) of records with available pollen (green) and charcoal (red) data in a 400-yr time bin.

#### 5.3 Controls on neotropical vegetation and fire regime

#### 5.3.1 Late Pleistocene (21 – 11.7 ka)






The different changes in vegetation and fire activity observed across the Neotropics highlights the influence of competing and context-dependent drivers that operate on different spatial and temporal scales. In general, low levels of CO<sub>2atm</sub>, such as during the LGM, reduce photosynthetic efficiency, mainly in C3 plants, limiting tree biomass potential growth and favoring C4 grasses (Boom et al., 2002; Foley, 1999; Maksic et al., 2022). A stronger impact of low CO<sub>2atm</sub> is expected on vegetation in warm tropical regions (e.g., NNeo, Amazonia, CEB, NEB) compared to cold regions (e.g., ExTrAn, SESA, TrAn) (Sage and Coleman, 2001). Reduced CO<sub>2atm</sub> is also suggested by modelling data to weaken the fire regime by altering availability and properties of biomass (Haas et al., 2023), while it can also increase the severity of fires by slowing post-burn tree recovery (Bond et al., 2003).

In sub- and extra-tropical latitudes (ExTrAn, SESA) and high-altitude sites (TrAn), temperatures 3–8°C lower than present likely constrained biomass growth during LGM, even in locally moist areas (Fig. 12a) (Cruz et al., 2005; Massaferro et al., 2009; Moreno et al., 2018). While cooler temperatures can improve water-use efficiency, frost events can produce long-term plant mortality and shape forest-savanna boundaries (Hoffmann et al., 2019; Inouye, 2000). Weak fire regime over these regions likely resulted from limited biomass in addition to lower temperatures and high moisture levels (Fig. 12g).

In warmer tropical regions (NNeo, Amazonia, CEB, NEB) precipitation seems to have played a pivotal role in the control of vegetation and fire dynamics. In NNeo, high tree cover and low fire activity were sustained by moist conditions (Fig. 12a,d). Strengthened SASM and an east-west precipitation dipole (wet west Amazonia and dry central-east Amazonia and NEB) (Cheng et al., 2013; Cruz et al., 2009; Kukla et al., 2023; Wang et al., 2017), led to decreased tree cover in

Amazonian ecotones and in NEB (Fig. 12a,d). In CEB, tree cover and fire patterns were heterogeneous (Fig. 12d) and influenced by both climatic and edaphic conditions. Nevertheless, colder and wetter conditions favored the increase in tree cover and migration of woody taxa across central Brazil (Pinaya et al., 2024). Fire activity remained mostly weak in tropical regions (Fig. 12g), likely constrained by low biomass availability due to reduced CO<sub>2atm</sub> and dryness in NEB (fuel-limited conditions) or by persistent moisture in NNeo and western Amazonia (moist-limited conditions).

The deglaciation is marked by progressive warming (Shakun et al., 2012) and rising CO<sub>2atm</sub> levels (Bereiter et al., 2015), along with substantial shifts in precipitation linked to millennial-scale events. In the Neotropics, HS1 (18-14.8 ka) and the YD (12.9-11.7 ka) were characterized by wetter conditions in south tropical latitudes (Campos et al., 2019; Meier et al., 2022; Mulitza et al., 2017) and drier conditions in north tropical latitudes (Deplazes et al., 2013; Zular et al., 2019), driven by southward ITCZ shifts. While our analyses can address the long-term trends shaped by these events, the low availability of high temporal resolution and continuous records with robust chronological control hinders a more detailed assessment of site-specific vegetation and fire anomalies over time.

Between 19 and 14.8 ka, Southern Hemisphere tropical regions (TrAn, NEB, SESA, ExTrAn) exhibit increasing tree cover and decreasing fire activity, mostly driven by rising temperatures and enhanced rainfall (Fig. 12e,h) (Campos et al., 2019; Shakun et al., 2012). In contrast, NNeo exhibits the opposite trends (Fig. 12e). In the second part of deglaciation, 14.8–11.7 ka, a generalized increase in tree cover and fire activity (Fig. 12f,i) coincides with further warming (Shakun et al., 2012) and CO<sub>2atm</sub> rise (Bereiter et al., 2015). TrAn and the northern parts of ExTrAn indicate an increase in tree cover (Fig. 12f), likely related to the combination of warming and wet conditions (Baker and Fritz, 2015; Montade et al., 2019). Notably, vegetation and fire responses in NEB (wetter) and SESA (drier) may reflect the prevailing precipitation dipole pattern (Campos et al., 2022; Cruz et al., 2009; Wong et al., 2023), although lower temperatures could have still played a key role in limiting tree cover in SESA (Fig. 12c,f). The stepwise intensification in southern hemisphere fire during this period, ca. 14–13 ka, agrees with a worldwide shift in fire regime intensification (Daniau et al., 2012), indicating that warmer conditions, contingent to fuel availability, were critical for this shift.

Fig. 12 – Late Pleistocene maps of site-specific arboreal pollen percentages and anomalies of average arboreal pollen and charcoal influx z-score values by time period: (a-c) arboreal pollen percentages, (d-f) arboreal pollen (AP) and (g-h) charcoal influx. Three time slices are considered: (a, d, g) Last Glacial Maximum (21 – 19 ka); (b, e, h) first part of the deglacial period encompassing the first stepwise warming (19 – 14.8 ka); (c, f, i) and second part of the deglacial period, encompassing the second stepwise warming (14.8 – 11.7 ka). Sites with positive or negative z-score anomalies indicate a record with predominantly higher or lower tree cover/fire activity than the average over the last 21 kyr. Elevations greater than 500 m are represented in light gray, while areas above 1500 m altitude are shown in dark gray. Major rivers are displayed as black lines.

# 5.3.2 Holocene (11.7 – 0 ka)

735

740

The greater availability of continuous records spanning the Holocene allows for a more detailed spatial assessment of vegetation and fire dynamics (Fig. 13). The similarities of tree cover and fire activity changes with precipitation shifts

suggest that moisture availability became a more important driver of Holocene vegetation and fire dynamics in the whole Neotropics.

Compared to the YD, the EH is characterized by negative tree cover trends in Amazonia, NEB, and CEB, slow tree cover increase in SESA and ExTrAn, where fire activity also increased, and opposite trends in NNeo (Fig. 13a,d,g). These changes are coherent with hydroclimate patterns related to a northward-displaced ITCZ and weaker SASM/SACZ resulting in drier conditions in most southern tropical regions (Cheng et al., 2013; Cruz et al., 2005), relatively drier but still humid northeastern Brazil (Cruz et al., 2009; Venancio et al., 2020), and wetter conditions in northern Neotropical latitudes (Haug et al., 2001). This period also featured a weaker, southward-displaced SWW, but relatively strengthened in its core (ca. 53°S) (Lamy et al., 2010), contributing to drier conditions in most of ExTrAn. Additionally, evidence of increasing human populations in Central and South America (Araujo et al., 2025; Goldberg et al., 2016; MesoRad, 2020) and plant domestication in NNeo, TrAn and Amazonia (Piperno, 2011) suggests that, although still limited, they played an active role as ecosystem engineers, influencing fire regimes and landscape dynamics.

In the MH, tree cover expanded over northern Amazonia, NEB, CEB, SESA, and ExTrAn (Fig. 13b,e). This pattern coincides with gradual precipitation changes driven by a southward expanded migration range of the ITCZ (Chiessi et al., 2021; Haug et al., 2001), a gradual intensification of the SASM/SACZ (Cheng et al., 2013; Prado et al., 2013a; Wong et al., 2023), and a weakening of the Nordeste Low (Cruz et al., 2009). The increase in fire activity over SESA and southwestern and western Amazonia during this period may have also resulted from increasing human activity over these regions (Araujo et al., 2025; Brugger et al., 2016; Lombardo et al., 2020), while in most of the southern Neotropics this period is marked by an occupation hiatus (Araujo et al., 2005, 2025). Although humans have been present in the Neotropics since the late Pleistocene (Goebel et al., 2008), their large-scale influence on tree cover and fire regimes became more pronounced by the end of MH, as consequence of a marked demographic expansion (Gill et al., 2007; Goldberg et al., 2016; Maezumi et al., 2018). This period also featured an intensification of the SWW and a northward migration of its northern boundary (Lamy et al., 2010; Razik et al., 2013) and a reduced strength/frequency of ENSO variability compared to the EH (Koutavas and Joanides, 2012; Polissar et al., 2013). This may have contributed to the ExTrAn moisture and arboreal vegetation increase and fire decrease, while in the Altiplano, centered around 12 °S, tree cover reduced and fire intensified amid dry conditions (Fig. 13e,h; Fig. A3).

During the LH, tree cover expanded in southern Amazonia, SESA, and part of CEB (Fig. 13c,f). This pattern was facilitated by increased moisture in these regions due to the strengthening of the SASM/SACZ (Cheng et al., 2013; Cruz et al., 2005). While our study did not assess for vegetation compositional changes that might highlight human impacts in these regions (Flantua and Hooghiemstra, 2023), intensified fire activity, despite increased moisture, may point to human influence. In contrast, tree cover declined in NNeo and NEB, where fire activity intensified. These declines were likely a result of both intensified human activity and climatic shifts toward drier conditions. The LH period marks the onset of semi-arid conditions in NEB (Cruz et al., 2009; Chiessi et al., 2021) and precipitation reduction in NNeo (Haug et al., 2001). Meanwhile, in TrAn, tree cover also declined despite moist conditions related to a strengthened SASM. Notably, NNeo and TrAn became densely populated during the LH, and the environmental impacts of human activities specially in these areas likely outweighed those driven by climate alone. In ExTrAn, tree cover remained relatively stable and fire activity increased, despite the rising moisture (Fletcher and Moreno, 2012; Lamy et al., 2010), also suggesting the influence of human activities in this region. In the Neotropics, numerous pollen records indicate human activity, particularly in the last 2 ka (Flantua et al., 2016; Flantua and Hooghiemstra, 2023), in line with our observations.

Fig. 13 – Holocene maps of site-specific arboreal pollen percentages and anomalies of average arboreal pollen and charcoal influx z-score values by time period: (a-c) arboreal pollen percentages, (d-f) arboreal pollen (AP) and (g-h) charcoal influx. Three time slices are considered: (a, d, g) early Holocene (11.7 - 8.2 ka); (b, e, h) mid Holocene (8.2 - 4.2 ka); and (c, f, i) late Holocene (4.2 - 0.0 ka). Sites with positive or negative z-score anomalies indicate a record with predominantly higher or lower tree cover/fire activity than the average over the last 21 kyr. Elevations greater than 500 m are represented in light gray, while areas above 1500 m altitude are shown in dark gray. Major rivers are displayed as black lines.

#### 6 Conclusions

Our assessment of modern climate-vegetation-fire dynamics, combined with a compilation of pollen and charcoal records from the Neotropics contributes to elucidating key environmental controls on vegetation and fire changes in the region over the last 21,000 yr. Our findings reveal contrasting shifts in vegetation and fire activity across the Neotropics, highlighting the complex interplay of various competing drivers, such as temperature, CO<sub>2atm</sub>, precipitation, vegetation-fire feedback, in addition to human impacts. We also indicate that the relative importance of different parameters affecting climate–vegetation–fire interactions vary across temporal and spatial scales.

In the southern latitudes (ExTrAn, SESA) and high Andes (TrAn), 3–8°C lower temperatures were the critical limiting factor for tree cover development during the glacial period. In contrast, in the warmer tropical regions (NNeo, Amazonia, CEB, NEB) precipitation played a pivotal role. We thereby suggest that shifts towards open arboreal cover during the glacial period should not be interpreted solely as indicators of dry conditions, particularly in regions where low temperatures and CO<sub>2atm</sub> constrained biomass growth, i.e., extra- and sub-tropical and high-montane regions. Fire activity, in turn, exhibits a non-linear response, increasing with biomass availability in fuel-limited conditions (ExTrAn, SESA, TrAn), but decreasing with moisture availability under moisture-limited conditions (NNeo, CEB). On the other hand, intensification in fire activity can hamper woody biomass growth. The deglacial stepwise increase in fire activity in several subregions of the Neotropics also suggests that warming thresholds can trigger rapid intensification of fire regimes, provided sufficient biomass is available.

During the Holocene, when variations in temperature and CO<sub>2atm</sub> were less pronounced, precipitation became a primary climatic determinant of tree cover and fire dynamics in the Neotropics. For instance, long-term Holocene increase in tree cover in Amazonia, CEB, and SESA were likely promoted by progressively increasing moisture, in opposition to NNeo and NEB. In addition to environmental controls, especially in the later parts of the Holocene, accelerated human demographic growth promoted widespread landscape transformation. The Holocene intensification of the fire regime probably resulted from a combination of warming and drying in some regions (NNeo, NEB) and direct human impacts (e.g., NNeo, TrAn), which further induced low tree cover states or delayed tree cover recovery. This finding raises concern for the future, as potential increases in extreme hot and dry events across parts of the Neotropics are likely to intensify fire regimes and tree cover loss in the region.

Our compilation underscores the scarcity of records spanning the last 21,000 years with both high temporal resolution and precise chronological control. Therefore, further downcore studies are required to better constrain the effects of short-lasting events on vegetation and fire dynamics. These efforts are crucial to better constrain the impacts of short-lived climatic events and human impacts on vegetation and fire dynamics, which may help to assess more rapid environmental responses as expected in the future.

# 7 Appendices

Fig. A1 – Location of supporting data used in this study. Sites with (a) pollen and (b) charcoal records used in the regional synthesis. (c) Sites from which data were extracted for this study and locations of radiocarbon-dated archaeological sites used to calculate summed probability distributions. 1 – Péten-Itza (PI-6) (Correa-Metrio et al., 2012); 2 – ODP site 1002 (Haug et al., 2001) and MD03-2621 (Deplazes et al., 2013); 3 – Santiago cave (Mosblech et al., 2012); 4 – Cueva del Diamante (NAR) and El Condor (ELC) caves (Cheng et al., 2013); 5 – Paraíso cave (Wang et al., 2017); 6 – Laguna Pallcacocha (Mark et al., 2022); 7 – Titicaca Lake (Baker et al., 2001); 8 – GeoB16202-2 (Mulitza et al., 2017); 9 – Rainha (RN) cave (Cruz et al., 2009); 10 – Lapa sem Fim (LSF) cave (Stríkis et al., 2018); 11 – Lapa Grande (LG) cave (Stríkis et al., 2011); 12 – M125-35-3 (Meier et al., 2022); 13 – Botuverá cave (Cruz et al., 2005); 14 – GeoB6211-2 (Chiessi et al., 2015); 15 – Lake Aculeo (Jenny et al., 2003); 16 – ODP site 1233 (Kaiser et al., 2005). Dots represent radiocarbon dates from archeological sites (Araujo et al., 2025; Goldberg et al., 2016; MesoRad, 2020). Elevations greater than 500 m are represented in light gray, while areas above 1500 m altitude are shown in dark gray.

Fig. A2 – Correlation between arboreal pollen z-scores and charcoal influx z-scores by subregion. Correlations were calculated for periods in which at least four records were available for both AP and charcoal influx z-score composites (black dots), periods with lower availability of records are shown (gray dots) but not included in calculations. A strong and significant positive correlation between tree cover and fire activity (c, e, f) indicates fuel-limited conditions, where an increase in biomass availability enhances fire potential. Conversely, a strong significant negative correlation (b) suggests climate-limited conditions, where climate variables, such as moisture, predominantly regulate fire dynamics.

Age (ka)

Fig. A3 – Tropical Andes (TrAn) subdivided in northern and southern sectors at 8 °S: (a) East Pacific SST anomaly (Koutavas and Joanides, 2012; Lea et al., 2006). (b) Zonal SST gradient anomaly between east and west Pacific (Koutavas and Joanides, 2012). (c) El Niño events per 100 years estimated from XRF data from Lake Pallcacocha (Mark et al., 2022). (d) Arboreal pollen (AP) and (e) charcoal influx z-scores composites from all records from TrAn located north of 8 °S. (f) Arboreal pollen (AP) and (g) charcoal influx z-scores composites from all records from TrAn located south of 8 °S. AP and charcoal-influx z-scores were produced using 1000-yr (green and red, respectively) and 400 yr (black) smoothing half-window. Gray areas represent 2.5th and 97.5th confidence intervals.

#### 8 Code availability







The analyses were primarily performed using code from already developed R packages "paleofire", "Neotoma" and "rearbon". However, the specific scripts used, based on these packages, can be found on GitHub at <a href="https://github.com/tkakabane/APcomp">https://github.com/tkakabane/APcomp</a>

# 9 Data availability

The authors declare that all data supporting the findings of our study are publicly available from the web or upon request to the authors. Pollen and charcoal data are available from Neotoma Paleoecology (http://www.neotomadb.org), Pangaea (https://www.pangaea.de/), and Reading databases. Present climate models are available from WorldClim (https://www.worldclim.org/), modern fire activity from the global fire patch functional traits database (FRY), and Wildlife terrestrial ecoregions be found the World can on Fund website (https://www.worldwildlife.org/publications/terrestrial-ecoregions-of-the-world).

#### 10 Supplementary data

**Supplementary data 1** – Site\_info: Details of the sites included in the study: site coordinates, site name, altitude, maximum and minimum estimated ages, and publication reference. Composite curves: Results from the composites using 1000-yr and 400-yr smoothing half-width. Neotoma\_ecogroup: Taxa classification into trees and shrubs (TRSH), palms (PALM), upland herbs (UPHE), and mangrove (MANG).

#### 11 Author contribution

TKA and ALD designed the experiments and TKA carried them out. TKA, CMC, ALD, and PEO prepared the manuscript with contributions from all co-authors. VH extracted data for analyses of modern fire pattern characterization. CMC, ALD, PEO, JW, ACC, DJBJ, MHS, and TAS contributed to interpretations, discussions and critical revision of the manuscript.

#### 12 Acknowledgments

TKA thanks the financial support from FAPESP (grants 2019/19948-0 and 2021/13129-8) and CAPES-COFECUB program (grant 88887.989386/2024-00). This study was financed, in part, by the São Paulo Research Foundation (FAPESP), Brazil, processes number 2018/15123-4 and 2019/24349-9. CMC acknowledges the financial support from CNPq (grant 312458/2020-7), and the CAPES-COFECUB program (grants 8881.712022/2022-1 and 49558SM). We thank UMR EPOC 5805 (France) for the support in conducting this research. We thank the insightful comments from the reviewers Nicholas O'Mara, Raquel Franco Cassino, and Paula A Rodríguez-Zorro, the community member Kees Nooren, and the editor Christian Franzke. Data were obtained from the Neotoma Paleoecology (<a href="https://www.neotomadb.org">https://www.neotomadb.org</a>), and Reading databases. The work of data contributors, data stewards, and the community involved in these databases are gratefully acknowledged.

## 13 Competing interests

The authors declare that they have no conflict of interest.

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
