# Peer review of "Vegetation and fire regimes in the Neotropics over the last 21,000 years"

_EGUsphere, 2025_

## Community Comment (CC1)

The article is well written, and has beautiful figures that nicely supports the results. I very much liked the first part of the article that analysis the relation between modern fire activity and current vegetation and climate. This part of their work could be an article on its own.

The second part of their work is very much restricted to the lack of enough long records, as the authors also emphasize in the last paragraph of their conclusions. This easily leads to wrong interpretations. The authors for example found for the NNeo a significant negative correlation between charcoal influx, and Arboreal pollen percentage. However, this correlation is heavily influenced by one charcoal record from lake Tulane in Florida, which record is very different from the other long record used (see figure). The Tulane charcoal record is extracted from the Reading Palaeofire database, also occur in Neotoma, but can't be found in the referred article (Grimm et al., 2006). Another charcoal record from lake Tulane (Watts and Hansen, 1988), also in the Reading Palaeofire database, is very different (see figure), but hasn't been used. An updated record for lake Tulane is likely to be published soon (Perrotti et al., 2023).

I

would suggest that the authors concentrate their work on the Holocene, or the last 6000 years, with a minimum number of palaeofire records for each subregion of at least ~10. They should define a minimum number of records for each subregion, and adjust the period studied accordingly. The authors should use a pre-binning of 400 or 500 years, instead of 20 years. In their current analysis many datapoints from low resolution records are missing.

Unfortunately, for many sites or the pollen or the charcoal record has been used. It would have been nice if the authors first of all investigate the relation between the charcoal and pollen record from individual sites. This would mean that more effort should have been put in the digitalization of records. It is at least a message to the palaeoecology community that more records should be added to the online databases. As long as the number of records included in the analysis is low, compared to the enormous size and heterogeneity of the study region, one should be careful with interpretation of the data, even if correlations seem significant.

Watts, W.A., and Hansen, B.C.S., 1988. Environments of Florida in the late Wisconsin and Holocene. In: Purdy BA (ed). Wet site archaeology. Telford Press, Caldwell, pp 307–324.

Perrotti et al., 2023. Does fire drive Quaternary ecosystem transformation at Lake Tulane, Florida? In: Abstracts of the 2nd Conservation Paleobiology Symposium. Bulletin of the Florida Museum of Natural History 60(2): 103.

---

## Author Comment (AC1)

**Community (Kees Nooren)**

The article is well written, and has beautiful figures that nicely supports the results. I very much liked the first part of the article that analysis the relation between modern fire activity and current vegetation and climate. This part of their work could be an article on its own.

**Response#1.** We sincerely thank Kees Nooren for the positive feedback.

The second part of their work is very much restricted to the lack of enough long records, as the authors also emphasize in the last paragraph of their conclusions. This easily leads to wrong interpretations. The authors for example found for the NNeo a significant negative correlation between charcoal influx, and Arboreal pollen percentage. However, this correlation is heavily influenced by one charcoal record from lake Tulane in Florida, which record is very different from the other long record used (see figure). The Tulane charcoal record is extracted from the Reading Palaeofire database, also occur in Neotoma, but can't be found in the referred article (Grimm et al., 2006). Another charcoal record from lake Tulane (WaΣs and Hansen, 1988), also in the Reading Palaeofire database, is very different (see figure), but hasn't been used. An updated record for lake Tulane is likely to be published soon (Perroţ et al., 2023). I would suggest that the authors concentrate their work on the Holocene, or the last 6000 years, with a minimum number of palaeofire records for each subregion of at least ~10. They should define a minimum number of records for each subregion, and adjust the period studied accordingly. The authors should use a pre-binning of 400 or 500 years, instead of 20 years. In their current analysis many datapoints from low resolution records are missing.

**Response#2.** Thank you for the remark on specific records from Florida, USA. However, despite some variability, we still find a significant negative correlation between charcoal influx and arboreal pollen when considering only Holocene pollen and charcoal records, as shown in Fig. R4.1.

[Figure]

**Fig R4.1.** Correlation of charcoal influx (Nneo CHAR) and arboreal pollen (Nneo TRSH) z-scores for the northern Neotropics over (a) the last 6 ka (p < 0.01, r2 = 0.33, r = -0.57), and (b) the last 11.7 ka (p < 0.01, r2 = 0.31, r = -0.56).

We agree that the Pleistocene period is represented by fewer and lower-resolution records, resulting in less robust spatial and temporal coverage, which is acknowledged in the manuscript. Nevertheless, we believe that analyzing the full 21 ka period provides a more valuable contribution to understanding both vegetation and fire dynamics, as well as their uncertainties, than restricting the study to a shorter interval with smaller changes in some of the boundary conditions (e.g., atmospheric $CO_2$ concentration, temperature). That said, we acknowledge that interpretations for periods with a smaller number of records should be treated with greater caution. In response, we will further include in the methods a note of caution about the intervals with lower confidence:

*"Caution is warranted when interpreting trends during periods with wide confidence intervals or when the composite curve approaches the upper or lower bounds of the confidence interval. These cases usually relate to periods with few records and indicate greater uncertainty and sensitivity to individual records. Strong fluctuations during such periods are likely highly uncertain and may reflect local variability of specific sites."*

We applied the two-stage smoothing method with a pre-binning half-window of 20 years (40-year bins) to prevent high-resolution records from disproportionately influencing the composite. This was followed by smoothing using half-windows of 400 and 1000 years. While changing these parameters does not significantly affect long-term trends, it can influence short-term variability. We therefore interpret short-term patterns during the Pleistocene with caution, as sparse data lead to large confidence intervals and unstable composites. Bootstrap resampling further indicates periods of disagreement among records or where individual records may dominate. In general, z-score values are very stable in terms of trends to varying pre-binning half-window values, although half-window smoothing can yield more stable curves, while losing details. For a brief discussion on this matter, please see our Response #3 and Fig. R2.3 to Reviewer #2 (Raquel F. Cassino). As such, increasing the pre-binning to 400 or 500 years would require a broader smoothing window, which could diminish the resolution of Holocene features without necessarily improving confidence in the Pleistocene, a period for which we already refrain from drawing further inferences due to the low data reliability.

Regarding the Tulane record, we will also include the record from Watts and Hansen (1988) in our analysis. However, we will keep the Tulane record available from the Reading Database. This record is also available and referenced in Neotoma (DOI: 10.21233/a8jq-7f35) with the following dataset notes: *"Grimm et al (2006) publication describes core, but charcoal dataset not used."*. In fact, the data available from the Reading Database were directly provided by Eric Grimm and Jim

Clark for the compilation published by Daniau et al. (2010) and later incorporated into the ACER compilation by Sanchez-Goñi et al. (2017). Thus, although it is not in the 2006 publication, this data was produced by the same authors and has been previously used in other publications. As for the record likely to be published soon (Perrott et al., 2023), it cannot be included in the current synthesis as it is not yet available.

References:

Daniau, et al.: Fire regimes during the Last Glacial, Quat. Sci. Rev., 29, 2918–2930, https://doi.org/10.1016/j.quascirev.2009.11.008, 2010.

Sanchez-Goñi, et al.: The ACER pollen and charcoal database: A global resource to document vegetation and fire response to abrupt climate changes during the last glacial period, Earth Syst. Sci. Data, 9, 679–695, https://doi.org/10.5194/essd-9-679-2017, 2017.

Grimm, et al.: Evidence for warm wet Heinrich events in Florida, Quat. Sci. Rev., 25, 2197–2211, https://doi.org/10.1016/j.quascirev.2006.04.008, 2006.

Unfortunately, for many sites or the pollen or the charcoal record has been used. It would have been nice if the authors first of all investigate the relation between the charcoal and pollen record from individual sites. This would mean that more effort should have been put in the digitalizing of records. It is at least a message to the palaeoecology community that more records should be added to the online databases. As long as the number of records included in the analysis is low, compared to the enormous size and heterogeneity of the study region, one should be careful with interpretation of the data, even if correlations seem significant.

**Response#3.** We agree that this suggestion represents a very interesting and valuable approach. However, incorporating such an analysis would considerably broaden the scope of the manuscript and further increase its already substantial size. Moreover, it would require selecting a subset of our compiled database that includes only records with both pollen and charcoal data, which would represent large structural changes to the entire study. While we recognize the potential of this approach, we believe it extends beyond the current scope and would be better suited for future research.

Watts, W.A., and Hansen, B.C.S., 1988. Environments of Florida in the late Wisconsin and Holocene. In: Purdy BA (ed). Wet site archaeology. Telford Press, Caldwell, pp 307–324.

Perroţti et al., 2023. Does fire drive Quaternary ecosystem transformation at Lake Tulane, Florida? In: Abstracts of the 2nd Conservation Paleobiology Symposium. Bulletin of the Florida Museum of Natural History 60(2): 103.

---

## Author Comment (AC2)

**Reviewer #1 (Nicholas O'Mara)**

**Review of Earth System Dynamics manuscript: egusphere-2025-1424**

This new compilation is impressive, and the analysis is thorough and detailed. I really appreciate the authors efforts to parse the records by climate/ecology to undergo a nuanced dissection of the competing influences of many factors on vegetation structure and fire regime and how these differ across geographies. The manuscript is very well written. They frame and motivate the problem well, the arguments are logical, the figures are clear and impactful, and it is overall quite pleasant to read. This study warrants speedy publication in Earth System Dynamics following minor revisions. I break down my review into major overarching comments followed by in-line comments and recommendations.

**Response #1**. We sincerely appreciate the positive assessment of our manuscript.

**Overarching comments:**

Throughout the manuscript, the term "biomass" appears to be used interchangeably with "tree cover". As a means of estimating vegetation structural change, the authors use the fraction of arboreal pollen in sediment cores. This method estimates the fraction of vegetation in a region which is composed of trees, however this is not a measure of biomass *per se*. All else equal, more trees on a landscape would equate to more biomass, but a ratio alone does not tell you this. Grasses can make up significant portions of the total biomass of ecosystems, particularly in tropical savannas (*e.g.*, Cerrado). The authors should take a careful look at the instances where they make claims about changes in biomass when they are actually measuring tree pollen fraction to infer changes in tree cover. Grass biomass is an important fuel source especially in tropical savannas like the Cerrado, so I urge caution to the authors on broadly equating increased tree cover with biomass in the context of fuel availability. I flag such instances in the in-line comments.

**Response #2**. Thank you very much for your observation. We will modify such instances accordingly, following your in-line comments by clearly stating woody or arboreal biomass when directly related to our compiled arboreal pollen data.

The description of the charcoal records is insufficient. The authors spend a decent portion of their methods section describing the dominant pollen types in the records that they compiled for each region but only list the number of records for charcoal without further description. Charcoal comes in many forms which record different aspects of fire regimes across multiple spatial scales. For instance, are all of the charcoal in the synthesis microcharcoal? Or are macrocharcoal particle records also included? One must read between the lines and look at the column title in the supplement table to infer this. A more complete description of the charcoal records

including at a minimum the size fraction of the records used in this synthesis is needed.

**Response #3**. We agree that descriptions regarding charcoal data could be clarified and improved. Accordingly, we will have revised the methods to ensure greater accuracy on this topic. More specifically, after standardization, we analyzed both macroscopic (>100μm) and microscopic (<100μm) charcoal data jointly, as performed in Power et al. (2008, 2010), Mooney et al. (2011), Gosling et al. (2021). The following sentence was included:

"*For charcoal composites, both micro- (< 100 μm) and macro- (> 100 μm) particles were included. Charcoal raw counts were converted into concentrations and then to influx using site-specific sedimentation rates, to account for differences in sedimentation rates across sites (Marlon et al., 2016).*"

References:

Gosling, et al.: Scarce fire activity in north and north-western Amazonian forests during the last 10,000 years, Plant Ecol. Divers., 14, 143–156, https://doi.org/10.1080/17550874.2021.2008040, 2021.

Marlon, et al.: Reconstructions of biomass burning from sediment-charcoal records to improve data-model comparisons, Biogeosciences, 13, 3225–3244, https://doi.org/10.5194/bg-13-3225-2016, 2016.

Mooney, S. D., et al.: Late Quaternary fire regimes of Australasia, Quat. Sci. Rev., 30, 28–46, https://doi.org/10.1016/j.quascirev.2010.10.010, 2011.

Power, M. J., et al.: Changes in fire regimes since the last glacial maximum: An assessment based on a global synthesis and analysis of charcoal data, Clim. Dyn., 30, 887–907, https://doi.org/10.1007/s00382-007-0334-x, 2008.

Power, M. J., et al: Fire history and the global charcoal database: A new tool for hypothesis testing and data exploration, Palaeogeogr. Palaeoclimatol. Palaeoecol., 291, 52–59, https://doi.org/10.1016/j.palaeo.2009.09.014, 2010.

The use of charcoal/pollen ratio records in this context surprises me. The authors say they multiply such records by the sedimentation rates of the cores to get an influx like unit, but this cannot be done. For such data to be ecologically meaningful, either (1) the pollen accumulation rate would have to be linearly correlated with the sedimentation rate and not a product of vegetation coverage within the watershed of the lake or (nearby river in the case of marine cores), or (2) the pollen accumulation rate would have to have such low variance that the change in charcoal accumulation rate drives the observed signal. Without convincing evidence in support of either of those scenarios, one cannot expect the charcoal numbers in these ratios to reflect changes in burning on the landscape. I suggest the authors remove such records from the compilation.

**Response #4**. We appreciate the thoughtful comment regarding the use of pollen/charcoal ratios in our analysis. We agree with the argument, and we will exclude the ratio-based records from the z-score composite curves to maintain

methodological consistency. Only four records were based on the pollen/charcoal ratio, two of them were not included in any subregion, one was included in NEB, and one in NNeo. However, for NEB where no composite curve was generated due to the scarcity of available charcoal data, we will keep the pollen/charcoal ratio for qualitative comparison (red curve in Fig. R1.1e – see below). This will be kept as simple charcoal/pollen ratios z-scores to provide at least some regional perspective. Additionally, two other curves were included in the new Fig. 8 (Fig. R1.1a,e), one from a record containing both charcoal and pollen data (Ledru et al., 2006) and another of δD-based reconstructed precipitation from a marine core off NEB.

[Figure]

**Fig. R1.1. Updated Fig. 8** – Northeastern Brazil (NEB) vegetation, fire, climate regimes, and human occupation: (a) δD $n$-$C_{29}$ alkane from GeoB16202-2 (Mulitza et al., 2017). (b) Speleothem $\delta^{18}O$ from Rio Grande do Norte cave (RN) (Cruz et al., 2009). (c) Summed density probability of $^{14}C$ ages from archeological sites in NEB (N = 542) (Araujo et al., 2025). (d) Arboreal pollen (AP) z-scores composites using 1000-yr (green) and 400 yr (black) smoothing half-window. Gray areas represent 2.5th and 97.5th confidence intervals. (e) Charcoal influx z-scores from single sites (yellow: Ledru et al. (2006); brown: De Oliveira et al., 1999; red: Bouimetarhan et al., 2018). (f) Number (#) of records with available pollen data in a 400-yr time bin.

Reference:

Ledru, M. P., et al.: Millenial-scale climatic and vegetation changes in a northern Cerrado (Northeast, Brazil) since the Last Glacial Maximum, Quat. Sci. Rev., 25, 1110–1126, https://doi.org/10.1016/j.quascirev.2005.10.005, 2006.

Figure 1 would benefit from a panel plotting either the fire radiative power or burned area data used in the modern analysis. The amount of burning across these biomes is highly variable. A map of where fires occur today would help the reader in interpreting the paleorecord compilations presented here.

**Response #5**. Thank you for the suggestion. We will include fire activity along with vegetation distribution. A preliminary updated version of Fig. 1a that includes fire activity can be checked below (Fig. R1.2).

[Figure]

**Fig. R1.2. Updated Fig. 1 – Vegetation and climate settings of the Neotropics**. Studied sites (white circles) and target subregions (numbered polygons) are depicted. **(a)** Ecoregions from Oslon et al. (2001): TrSuMBF – Tropical subtropical moist broadleaf forests; TrSuDBF – Tropical subtropical dry broadleaf forests; TrSuGSS – Tropical subtropical grasslands, savannas, and shrublands; TeBMiF – Temperate broadleaf mixed forests; TeCF – Temperate coniferous forests; FGS – Flooded grasslands and savannas; DXSh – Desert and xeric shrublands; MoGSh – Montane grasslands and shrublands; MeFWSc – Mediterranean forests, woodlands, and scrubs . Mean burned area of the last 12 years from Laurent et al. (2018).

I would encourage the authors to emphasize that the majority of the sites in this study, except those in region 5 (Central Eastern Brazil) in the Cerrado, are currently situated in more forested regions that do not burn as much as tropical savannas. This would tie in within the results of the modern analysis (Figure 3) to show that one might expect more fire in the past if grasses where a more dominant fraction of the local vegetation.

**Response #6**. This helpful comment prompted us to strengthen the connection between Section 5.1, "Modern climate–fire–vegetation relationships", and the

subsequent discussion of regional patterns. We would like to emphasize, however, that some areas currently covered by mostly forested vegetation were dominated by more open vegetation types in the past. This is particularly evident in Southeastern South America (SESA) (Fig. R1.3) and Southern Andes (SAn). In SESA, for example, we observe a shifting relationship between tree cover and fire activity over time, with both arboreal pollen (AP) and charcoal increasing during Termination 1, followed by a continued rise in AP and stabilization of fire activity. Therefore, while only the Cerrado and high-altitude Andes are predominantly covered by open vegetation under modern conditions, regions such as SESA and SAn were also covered by open vegetation in the past and likely exhibited different fire–vegetation dynamics (e.g., Fig R1.3).

[Figure]

**Fig. R1.3** – Correlation between arboreal pollen z-scores and charcoal influx z-scores in Southeastern South America (SESA).

*"These fire-prone conditions are typical of tropical savannas and grasslands (Fig. 3a,b), such as in CEB, where a regular frequency of fire events is key in maintaining biodiversity and the physiognomy of vegetation (Bernardino et al., 2022; Mistry, 1998). In modern day CEB, however, fire regime is mostly limited by fuel moisture (moisture-limited condition, negative biomass-fire correlation), except towards its transition to semiarid conditions (Fig. 3a,b) (Alvarado et al., 2020). In arid and semiarid environments, such as NEB, or high-altitude areas, such as CAn, fire activity is generally hindered due to biomass limitation (fuel-limited conditions, positive biomass-fire correlation) (Fig. 3a-e). On the other hand, under wet conditions of tropical rainforests, such as Amazonia or parts of NNeo, fire activity is limited due to constant fuel moisture (moisture-limited condition) (Fig. 3a,b)."*

Figures 12 and 13. I really like the time snapshot analysis presented in these figures. However, it is a little bit difficult to have to read across the panels and compare the colors between points to see if the z-scores increased or decreased through time by comparing the color to the previous time slice. When I first looked at these maps, I

expected that they were displaying the trends rather than the mean z-scores for the time slice. One minor change you could make to these figures would be to change the markers depending on whether the mean z-score increased or decreased compared to the last time slice. *E.g.*, upward triangle for increase above some threshold between 21-19 to 19-14.8, downward triangle for decrease below some threshold, and circle for no change outside of some threshold. This would really help as the reader is going through your later portion of the discussion when you are providing a broad overview of the trends through time. Additionally, the time slice labels are backward, they should be in chronological order, *e.g.*, "(c) 21 – 19 ka". You have it right in the figure caption, just reversed in the panel labels.

**Response #7**. We improved the description on how to interpret these maps in the methodology.

"*Each site-specific data point represents a mean z score calculated relative to its own long-term mean (base period: 21,000 to 200 yr BP). As a result, the colors of the dots within a single map reflect deviations from local baseline conditions and should not be directly compared across sites to infer geographic gradients or absolute levels of fire activity or tree cover. However, spatial clusters of similar z-score trends, such as consistently positive values in a region, may indicate coherent regional patterns, for example a general expansion in tree cover or intensification of the fire regime.*"

We will also fix the backward labels mentioned. However, we prefer not to represent changes using different symbols, as this may introduce even more complexity. In some cases, sites may not display continuous changes between time slices due to, e.g. hiatuses or sampling resolution, which would require an additional symbol to distinguish these instances from increases, decreases, or stable conditions. We decided to use this approach as it has already been successfully used in other studies (e.g., Marlon et al., 2016; Mooney et al., 2011; Power et al., 2008). As a side note, in Power et al. (2008), triangles are used, but simply to indicate positive or negative z-scores for a given timeslice.

References:

Marlon, et al. Reconstructions of biomass burning from sediment-charcoal records to improve data-model comparisons, Biogeosciences, 13, 3225–3244, https://doi.org/10.5194/bg-13-3225-2016, 2016.

Mooney, S. D. et al.: Late Quaternary fire regimes of Australasia, Quat. Sci. Rev., 30, 28–46, https://doi.org/10.1016/j.quascirev.2010.10.010, 2011.

Power, et al.: Changes in fire regimes since the last glacial maximum: An assessment based on a global synthesis and analysis of charcoal data, Clim. Dyn., 30, 887–907, https://doi.org/10.1007/s00382-007-0334-x, 2008.

Regarding data availability, I suggest that the authors make the smoothed z-score time series of AP and charcoal influx available in the supplement or permanently stored in a public archive. These new synthesis curves are valuable information for paleoclimatologists, paleoecologists, anthropologists, climate modelers, *etc.,* who might wish to compare them with their data. As I am sure the authors are aware from the webplotdigitizing they did for this study, it is always a relief when the key data for a paper are easily accessible for future analysis and plots don't need to be unnecessarily and painstakingly recreated.

**Response #8**. We agree. All composite curves will be provided as Supplementary data.

**In-line comments:**

19-20: I suggest you remove the "in the one hand" and "on the other hand" they are just not necessary to the point of the sentence which is already clear and concise.
Agree.

24-25: Perhaps add "with additional impacts from human activity" to the end of the sentence which starts with "Temperature …"
Agree.

26: "Biomass growth" this should be "tree growth"
Agree, but we opt for "arboreal growth".

46: "process" should be "processes"
Agree.

50: Glacial/interglacial cycles were occurring (although much more muted) in the Neogene. I suggest you change this to something like "onset of pronounced glacial/interglacial cycles in the Quaternary"
Agree.

51: Change "were responsible for" to "played a significant role in"
Agree.

52: Please clarify here if you mean in setting the modern ecosystem distributions or modulating changing ecosystem distributions through time.
Agree.

56-57: "Weaker fire regime" is not very clear to a general reader. I am okay with the use of weaker and stronger fire regimes throughout the paper, but take a sentence here to explain what characteristics constitute and "weak" versus "strong" fire regime. Additionally, you could maybe also clarify also here what "fire activity" means because you use this general term in the text as well.
Agree. We will provide some explanation in the method section, to avoid a break in the introduction for explaining the concept. While our approach does not allow us to distinguish specific aspects of the fire regime such as intensity, severity, frequency, seasonality, spatial extent of the burned vegetation, we use the broader

terms "fire regime" or "fire activity" to reflect the general response of fire to environmental changes, which is directly interpreted from values of charcoal influx. "*Changes in charcoal records can be linked to past fire activity and used to infer shifts in fire regimes. While our approach does not allow us to resolve specific components of the fire regime (e.g., intensity, severity, frequency, seasonality, spatial extent of burned vegetation), we consider fire regime as the collective changes in charcoal influx trends within a given region over a long timescale. We use fire activity as a more general term to describe variability in charcoal influx.*"

Also see our Response #5 to Reviewer #3 (Paula A. Rodríguez-Zorro).

81 and throughout: You interchange "mm.yr-1" and "mm yr⁻¹" I recommend you adopt the second in all cases, the added period is unnecessary.

Agree.

86-87: "...marked by weak seasonality with mean monthly temperatures ranging between 25 and 27 °C and mean annual precipitation of..." would be better as a comma-separated list: ""...marked by weak seasonality, mean monthly temperatures ranging between 25 and 27 °C, and mean annual precipitation of..."

Agree.

93-94: It is not clear to me which positive feedback loop you are referring to. *E.g.* fire impacts on vegetation structure and knock-on effects on temperature and future fire likelihoods, fires emitting greenhouse gases leading to overall warming and thus more fires, *etc*. Please clarify.

Agree. We will change to "*Persistent moist conditions of the rainforest naturally inhibit wildfires. However, initial fire events whether triggered by severe droughts (e.g., related to El Niño), ongoing climate changes, and/or human impacts further increase forest flammability through canopy degradation and fuel accumulation. This favors subsequent fire events, thereby fostering a positive feedback loop (Brando et al., 2020; Bush et al., 2008; Cochrane et al., 1999; Nepstad et al., 1999).*"

99: "area" should be "areas"

Agree.

103-104: "moist-laden" should be "moisture-laden"

Agree.

199: I suggest you change "...open grasslands to closed shrublands and woodlands..." to "...open grasslands and savannas to closed shrublands and woodlands..."

Agree.

121: "...the occurrence South Atlantic..." should be "...the occurrence of the South Atlantic..."

Agree.

125-145: This is largely a style choice, so up to you, but in all preceding paragraphs you list temperature ranges from low to high (which is convention) but here you list

high to low. I see that you might be doing this intentionally as you describe the regions from north to south, but it is a little weird to read temperature ranges from 16 to 3 °C.

Thank you for noting this. The reversed temperature sequence (16 to 3°C) was indeed intentional to match the north-to-south latitudinal progression (32°S to 55°S). We will adapt the text to "Mean annual temperature ranges from 3 to 16 °C between the latitudes 55 and 32°S".

140: "...southward-displaced..." should be "...southwardly displaced..."

We will slightly modify this part to "*Precipitation is controlled by the Southern Westerly Winds (SWW), which shift southward during austral summer,* [...]"

165: For example, here is the only place in the text you mention microcharcoal. See overarching comments for method recommendations.

This comment was addressed in the overarching comments above.

201-203: "For charcoal composites, ..." You need to either provide strong justification that this is a viable method or remove such records from the compilation. See overarching comment.

Agree.

295: "Despite represented by..." should be "Despite being represented by..."

Agree.

303: Is this a mistaken paragraph break at the end of this line?

This was intentional as an introduction sentence for the following paragraphs in which we describe the distinct regional patterns. We included a ":" to the end of this line to make it clearer.

315: Figure 4 should have y-axis labels. A single label common for all subplots would be fine. The two columns appear unnecessarily squished together horizontally; I think you can add a little separation between the two which should give ample room for the y-axis labels.

Indeed. Thank you for the suggestion. We will fix Fig. 4 accordingly.

341: "...high level..." and "...fire regime..." should be "...high levels..." and "...fire regimes..."

Agree.

349: "p-values" can just be "p"

Agree.

351: "condition" should be "conditions"

Agree.

354: drop the "and" before "likely"

Agree.

359: "...coeval to..." should be "...coeval with..."

Agree.

357-358: I do not really see a slope break? The decline appears to be part of the larger trend of declining AP % since the EH. So, I would avoid calling it abrupt.

We will remove the word "abrupt." Our intention was to convey that the Late Holocene decline in tree cover appears more pronounced than the gradual decrease observed since the Early Holocene, which we interpret as a shift likely amplified by direct human activity.

378: "...featured a reduced..." should be "...featured reduced..."

Agree.

380-382: I thought Amazonia is generally moisture limited, so isn't it surprising that low rainfall leads to low fire activity?

Yes, indeed, multiple interacting forcings contribute to a complex response. For example, the Amazon is a highly heterogeneous and predominantly fire-limited ecosystem, with marked west–east and north–south gradients. During the LGM, eastern Amazonia is interpreted to have been drier (Häggi et al., 2017; Wang et al., 2017) while western Amazonia was wetter, or as wet as modern conditions (Baker et al., 2001; Cheng et al., 2013). In addition, the southern and eastern Amazon rainforests were relatively smaller compared to their pre-Industrial extension (Mayle et al., 2000), possibly as a consequence of lower $CO_2$ (Maksic et al., 2022) and drier conditions (Fontes et al., 2017).

Despite these regional contrasts, charcoal records consistently indicate low fire activity during the LGM and part of Termination 1 across the eastern (Hermanowski et al., 2012), central-northern (Blaus et al., 2024; Bush et al., 2004) and southern (Cordeiro et al., 2014; Fontes et al., 2017) Amazonia. A different scenario is reported for southwestern Amazonia (Burbridge et al., 2004) with decreasing fire activity towards the Holocene. Over eastern and southern Amazonia, forest vegetation was largely replaced by savannas during the late Pleistocene. This seemingly counterintuitive pattern may be explained by colder temperatures, reduced convective activity (and consequently fewer lightning ignitions), or even increased megafauna herbivory limiting fuel accumulation. We will incorporate this point into the discussion.

References:

Baker, et al.: The history of South American tropical precipitation for the past 25,000 years, Science, 291, 640–643, https://doi.org/10.1126/science.291.5504.640, 2001.

Blaus, et al.: Climate, vegetation, and fire, during the last deglaciation in northwestern Amazonia, Quat. Sci. Rev., 332, 108662, https://doi.org/10.1016/j.quascirev.2024.108662, 2024.

Burbridge, et al.: Fifty-thousand-year vegetation and climate history of Noel Kempff Mercado National Park, Bolivian Amazon, Quat. Res., 61, 215–230, https://doi.org/10.1016/j.yqres.2003.12.004, 2004.

Bush, et al.: Amazonian paleoecological histories: one hill, three watersheds, Palaeogeogr. Palaeoclimatol. Palaeoecol., 214, 359–393, https://doi.org/10.1016/S0031-0182(04)00401-8, 2004.

Cheng, et al.: Climate change patterns in Amazonia and biodiversity, Nat. Commun., 4, 1411, https://doi.org/10.1038/ncomms2415, 2013.

Cordeiro, et al.: Palaeofires in Amazon: Interplay between land use change and palaeoclimatic events, Palaeogeogr. Palaeoclimatol. Palaeoecol., 415, 137–151,

https://doi.org/10.1016/j.palaeo.2014.07.020, 2014.

Fontes, et al.: Paleoenvironmental dynamics in South Amazonia, Brazil, during the last 35,000 years inferred from pollen and geochemical records of Lago do Saci, Quat. Sci. Rev., 173, 161–180, https://doi.org/10.1016/j.quascirev.2017.08.021, 2017.

Häggi, et al.: Response of the Amazon rainforest to late Pleistocene climate variability, Earth Planet. Sci. Lett., 479, 50–59, https://doi.org/10.1016/j.epsl.2017.09.013, 2017.

Hermanowski, et al.: Palaeoenvironmental dynamics and underlying climatic changes in southeast Amazonia (Serra Sul dos Carajás, Brazil) during the late Pleistocene and Holocene, Palaeogeogr. Palaeoclimatol. Palaeoecol., 365–366, 227–246, https://doi.org/10.1016/j.palaeo.2012.09.030, 2012.

Maksic, et al.: Brazilian biomes distribution: Past and future, Palaeogeogr. Palaeoclimatol. Palaeoecol., 585, 110717, https://doi.org/10.1016/j.palaeo.2021.110717, 2022.

Mayle, et al.: Millennial-Scale Dynamics of Southern Amazonian Rain Forests, Science (80-. )., 290, 2291–2294, https://doi.org/10.1126/science.290.5500.2291, 2000.

Wang, et al. Hydroclimate changes across the Amazon lowlands over the past 45,000 years, Nature, 541, 204–207, https://doi.org/10.1038/nature20787, 2017.

385-389: This intra-region spatial variability is really interesting. Are there enough records to split Amazonia up further and potentially make curves of E vs W or N vs S? If so, that could make for a nice supplementary figure. But if not, okay!

This would be indeed very interesting. However, to carry out such analysis we would have to limit our scope to the Holocene and focus on Amazonia. Particularly, more detailed discussions on Holocene fire and vegetation in Amazonia have been done in Gosling et al. (2021), Smith et al. (2018), and Mayle and Power (2008), for example. Still, we will include some of these nuances and references related to regional heterogeneity in the discussion.

References:

Gosling, et al.: Scarce fire activity in north and north-western Amazonian forests during the last 10,000 years, Plant Ecol. Divers., 14, 143–156, https://doi.org/10.1080/17550874.2021.2008040, 2021.

Mayle, F. E. and Power, M. J.: Impact of a drier Early–Mid-Holocene climate upon Amazonian forests, Philos. Trans. R. Soc. B Biol. Sci., 363, 1829–1838, https://doi.org/10.1098/rstb.2007.0019, 2008.

Smith, R. J. and Mayle, F. E.: Impact of mid- to late Holocene precipitation changes on vegetation across lowland tropical South America: A paleo-data synthesis, Quat. Res. (United States), 89, 134–155, https://doi.org/10.1017/qua.2017.89, 2018.

436: "…the influence different…" should be "…the influence of different…"

Agree.

438: "moist-laden" should be "moisture-laden"

Agree.

441: "…speleothem cores suggest primarily reflect…" remove either "suggest" or "primarily reflect"

Agree.

444-446: This is an interesting point. I wonder if you could comment further here about the potential impacts of llama grazing on grassy fuel loads?

It is challenging to disentangle the specific impacts of llama grazing from broader human influences at this stage. However, if considering grazing pressure alone, increased llama activity could have reduced grassy fuel loads, consistent with findings from other regions where megafauna or livestock grazing limits fine flammable biomass (Blackhall et al., 2017; Fuhlendorf and Engle, 2004; Furquim et al., 2024; Gill et al., 2009), even though the grazing-fire relationship is not always straightforward (e.g., Blackhall et al., 2017). Additionally, the continued anthropogenic use of fire may have sustained relatively high fire activity, potentially offsetting the fuel-reducing effects of llama grazing alone. Warming conditions and an increase in fuel availability could also have contributed to more intense fires.

While our primary goal is to provide a broader overview of vegetation and fire dynamics, we acknowledge the relevance of this point. In response to this and a comment from Reviewer #2, we will include a few remarks throughout the discussion considering the influence of grazing, even though it was likely of secondary importance considering the timespan of our analyses and human impacts during the Holocene.

References:

Blackhall, et al.: Effects of biological legacies and herbivory on fuels and flammability traits: A long-term experimental study of alternative stable states, J. Ecol., 105, 1309–1322, https://doi.org/10.1111/1365-2745.12796, 2017.

Fuhlendorf, S. D. and Engle, D. M.: Application of the fire-grazing interaction to restore a shifting mosaic on tallgrass prairie, J. Appl. Ecol., 41, 604–614, https://doi.org/10.1111/j.0021-8901.2004.00937.x, 2004.

Furquim, et al.: Interactive effects of fire and grazing on vegetation structure and plant species composition in subtropical grasslands, Appl. Veg. Sci., 27, 1–13, https://doi.org/10.1111/avsc.12800, 2024.

Gill, et al.: Pleistocene Megafaunal Collapse, Novel Plant Communities, and Enhanced Fire Regimes in North America, Science, 326, 1100–1103, https://doi.org/10.1126/science.1179504, 2009.

501-505: It seems to me that the high rainfall observed during HS1 and during the YD ◊ EH match peaks in the charcoal influx and BA matched a trough in the charcoal influx, suggesting fuel limited conditions which would make sense for the Cerrado savanna vegetation.

These specific millennial-scale changes may suggest fuel-limited conditions during Termination 1. However, the long-term negative correlation between tree cover and fire (Fig. 9c-d of the original version of the manuscript) points to moisture-limited conditions. Uncertainties in regional patterns and chronology pose additional challenges in interpreting these short-term variabilities. Moreover, most of the Cerrado today is considered moisture-limited, with the exception to its transition to drier vegetation types (Alvarado et al., 2020). Taken together, we see robust arguments not to change the original interpretation on this specific point.

Reference:

Alvarado, et al.: Thresholds of fire response to moisture and fuel load differ between tropical savannas and grasslands across continents, Glob. Ecol. Biogeogr., 29, 331–344, https://doi.org/10.1111/geb.13034, 2020.

*Also, what do you think happened* ca. *19-17 ka? Why are the z-scores for charcoal influx so low? It's hard to tell from the figure, but does the number of records go to zero at points in this interval? It would be good to comment on this. And perhaps obscure portions of the curve which might have very low confidence due to lack of sufficient data.*

This is likely an artifact resulting from the scarcity of data during this period, which makes the composite curve highly unstable and sensitive to specific records. Thus, we are not interpreting such features. The negative value reaches –2.2.

We have now included in the method section the following note of caution:

"*Caution is warranted when interpreting trends during periods with wide confidence intervals or when the composite curve approaches the upper or lower bounds of the confidence interval, or exhibits outlier shifts. These cases usually relate to periods with few records and indicate greater uncertainty and sensitivity to individual records. Strong fluctuations during such periods are likely highly uncertain and may reflect local variability of specific sites.*"

517: "Biomass growth" should be "tree growth"

Agree.

517-518: "moist-laden" should be "moisture-laden"

Agree.

535-536: I would recommend "biomass" in both cases should be "tree cover"

Done for the second case. In the first case we changed to woody biomass.

545: "biomass" should be "tree cover"

Changed to arboreal biomass.

566: "This was likely consequence from significantly…" should be "This was likely the consequence of significantly…"

Agree.

568: "tree cover increases along warming…" should be "tree cover increases along with warming…"

Agree.

574: "biomass" should be "tree cover"

Changed to woody biomass.

589-590: The human population was already quite high by 7 ka, so why is the influence on fires delayed until 3.5 ka?

Humans were very likely to influence fire during the Mid-Holocene and even earlier. However, their regional impact became more evident (in our results) when both tree

cover and humidity increased over the region and, unexpectedly, fire activity also increased. This likely suggests a superimposed impact of human activity on the background natural trend.

604: "biomass" should be "tree biomass"

Agree.

606: I think it is worth pointing out to the readers that Haas et al., (2023) is a modeling study and we do not yet fully understand the impacts of low atmospheric $CO_2$ on global fire regimes.

Agree.

Overall, I really enjoyed this well-written and interesting paper, and I am excited to see the manuscript published following these revisions. Nice work!

Thank you very much for the throughout revision and insightful comments.

---

## Author Comment (AC3)

Review of the manuscript entitled "Vegetation and fire regimes in the Neotropics over the last 21,000 years"

The manuscript presents an excellent and timely synthesis, combining analyses of modern data with an extensive compilation of fossil records and archaeological evidence. The work is well written, clearly organized, and supported by high-quality figures that effectively convey the results.

**Response #1.** We are grateful for the comments from Raquel F. Cassino on our manuscript.

I would like to offer a few comments and questions that I hope may help to further strengthen the manuscript:

1-Definition of arboreal pollen

The authors calculated arboreal pollen (AP) percentages as the sum of woody taxa (trees and palms), excluding mangrove and aquatic taxa, fern spores, and unidentified types. This approach is widely used in paleoecological studies and serves as a valuable proxy for reconstructing past vegetation structure. However, given the floristic complexity of tropical ecosystems, I would like to kindly suggest that the authors provide further clarification regarding the taxonomic criteria used to define AP. Specifically, many plant families in these ecosystems include both arboreal and non-arboreal life forms, which may significantly influence AP percentages depending on how these groups were categorized. It would be helpful to know how the authors distinguished between arboreal and non-arboreal taxa within such families, and whether any standardized criteria were applied in this process.

For instance, I wonder whether the palm *Mauritia flexuosa* was considered part of the AP in the Cerrado records. Given that Mauritia palms are often highly abundant in local swamp environments (veredas), their inclusion could potentially inflate AP percentages without necessarily indicating a broader regional forest expansion. Clarifying this point would be particularly valuable for interpreting AP trends in relation to regional woody cover dynamics. Providing these additional details could enhance the transparency and reproducibility of the study, and also refine the paleoecological interpretations drawn from the AP trends.

**Response #2.** The ecological types are based on Neotoma standardized classification. We will include a spreadsheet with all used taxa and corresponding classification (TRSH – trees and shrubs, PALM - palms, UPHE – upland herbs) in the supplementary table.

We will modify it to read: *"For Neotoma records, arboreal pollen (AP), which serves as an indicator of tree cover relative to herbaceous vegetation, was calculated as the percentage of woody taxa (trees, shrubs (TRSH), and palms (PALM), considering taxa at the genus and family level) divided by the total sum of trees and shrubs (TRSH), palms (PALM), and herbs (UPHE), excluding mangrove and aquatic taxa, fern spores, and unidentified types. The classification of ecological groups used in these calculations follow Neotoma standardized classifications (Supplementary table 1). For manually extracted records, AP percentages were obtained from published pollen diagrams. Therefore, the criteria used to construct these AP curves may slightly differ from those applied in our calculations based on raw data. For instance, in CEB, Mauritia and Cyperaceae are often excluded from AP calculations, as these taxa are often over-represented due to strong local imprint from palm swamp vegetation (Barberi et al., 2000; Escobar-Torrez et al., 2023; Salgado-Labouriau et al., 1997). To maintain consistency for this specific region, we also excluded Mauritia and Cyperaceae from our AP calculations."*

Particularly for Central-Eastern Brazil (CEB), most records were manually extracted from published pollen diagrams. As such, we had to rely on the available AP curves, which may differ slightly from our calculation methods using datasets for which raw data were accessible (e.g., those obtained from Neotoma or Pangaea).

For example, in the case of Lagoa Feia (Cassino et al., 2020; Escobar-Torrez et al., 2023), we used the AP curve available in Escobar-Torrez et al. (2023), which we assume does not include Mauritia in the AP calculation.

In the Chapada dos Veadeiros record (Ferraz-Vicentini, 1999), *Mauritia* is rare and occurs in low percentages, so its inclusion or exclusion does not meaningfully affect the results. For other records, such as Cromínia (Salgado-Labouriau et al., 1997) and Vereda de Águas Emendadas (Barberi et al., 2000), we calculated AP using the sum of 100% excluding the "vereda" group. This was not the case for Lagoa Bonita (Barberi, 2001), in which we had initially included all AP, but will now correct the record by excluding the "Pólen de brejo e vereda" group. For Vereda São José (Cassino et al., 2018), available in Neotoma, we will remove *Mauritia* to ensure consistency with the manually extracted records from the region.

Nevertheless, the newly calculated AP composite closely resembles the previous one and does not result in any change to our interpretations (Fig. R2.1). Additionally, we will include a new precipitation curve based on speleothem records from a site located in the northern parts of CEB (Stríkis et al., 2011, 2018), providing further regional context.

[Figure]

**Fig. R2.1. Updated Fig. 9 –** Central-eastern Brazil (CEB) vegetation, fire, climate regimes, and human occupation: (a) Speleothem δ18O from Lapa sem Fim (LSF) and Lapa Grande (LG) caves (Stríkis et al., 2011, 2018). Downcore ln(K/Al) from marine sediment core M125-35-3, which reflects changes in the clay mineral composition and increases with chemical weathering intensity and hence, moisture availability (Meier et al., 2022). (b) Summed density probability of $^{14}$C ages from archeological sites in CEB (N = 481). (c) Arboreal pollen (AP) and (d) charcoal influx z-scores composites using 1000-yr (green and red, respectively) and 400 yr (black) smoothing half-window. Charcoal z-score negative anomaly reaching -2.2 is indicated by a circled arrow. Gray areas represent 2.5th and 97.5th confidence intervals. (e) Number (#) of records with available pollen (green) and charcoal (red) data in a 400-yr time bin.

References:

Barberi.: Mudanças paleoambientais na região dos cerrados do planalto central durante o quaternário tardio: o estudo da lagoa bonita (DF), University of São Paulo, 210 pp., https://doi.org/https://doi.org/10.11606/T.44.2001.tde-04112015-161453, 2001.

Barberi et al.: Paleovegetation and paleoclimate of "Vereda de Aguas Emendadas", central Brazil, J. South Am. Earth Sci., 13, 241–254, https://doi.org/10.1016/S0895-9811(00)00022-5, 2000.

Cassino, et al.: A Late Quaternary palynological record of a palm swamp in the Cerrado of central Brazil interpreted using modern analog data, Palaeogeogr. Palaeoclimatol. Palaeoecol., 490, 1–16, https://doi.org/10.1016/j.palaeo.2017.08.036, 2018.

Cassino, et al.: Vegetation and fire variability in the central Cerrados (Brazil) during the Pleistocene-Holocene transition was influenced by oscillations in the SASM boundary belt, https://doi.org/10.1016/j.quascirev.2020.106209, 2020.

Escobar-Torrez et al.: Long-and short-term vegetation change and inferred climate dynamics and anthropogenic activity in the central Cerrado during the Holocene, J. Quat. Sci., https://doi.org/10.1002/jqs.3567, 2023.

Ferraz-Vicentini.: História do Fogo no Cerrado: Uma Análise palinológica, 1999.

Salgado-Labouriau et al.: Late quaternary vegetational and climatic changes in cerrado and palm swamp from Central Brazil, Palaeogeogr. Palaeoclimatol. Palaeoecol., 128, 215–226, https://doi.org/10.1016/S0031-0182(96)00018-1, 1997.

Stríkis, et al.: Abrupt variations in South American monsoon rainfall during the Holocene based on a speleothem record from central-eastern Brazil, Geology, 39, 1075–1078, https://doi.org/10.1130/G32098.1, 2011.

Stríkis, et al.: South American monsoon response to iceberg discharge in the North Atlantic, Proc. Natl. Acad. Sci. U. S. A., 115, 3788–3793, https://doi.org/10.1073/pnas.1717784115, 2018.

**2-Charcoal data scaling**

The use of z-score transformation for scaling charcoal data, as applied by the authors, is a widely accepted and established method in paleo-fire research, allowing for effective comparison of variability within and between records. However, recent studies (e.g., McMichael et al., 2021; Gosling et al., 2021) have highlighted some potential limitations of z-score scaling, particularly its tendency to distort the structure of charcoal peaks by inflating small-scale peaks and minimizing the influence of large peaks. Moreover, as noted by McMichael et al. (2021), this method "does not retain a consistent value that represents the absence of fire across sites", which can be especially relevant in tropical ecosystems where documenting both the presence and absence of fire is critical for understanding fire regime variability.

As an alternative, proportional relative scaling has been suggested (Gosling et al., 2021), which transforms charcoal data to a 0–100 scale while retaining the zero value to consistently represent fire absence. This approach also avoids upweighting rare charcoal finds in otherwise charcoal-poor sequences, potentially providing a more ecologically meaningful representation of fire activity.

Given these recent discussions, I wonder whether the authors considered alternative scaling methods, and if so, what motivated the decision to apply the z-score transformation. I believe that elaborating on this methodological choice could be valuable for readers and for future studies in similar tropical contexts.

**Response #3.** We appreciate the suggestion regarding data scaling. We have considered the use of relative scaling (0–1) for both AP and charcoal influx data. For charcoal, the use of z-scores, as highlighted in the comment, is a widely accepted and established method in paleo-fire research. We chose to use z-scores as a meaningful way of comparing deviations from local norms, which is particularly valuable when comparing across sites with different vegetation types and fire

regimes, as well as among studies with different methodologies for charcoal quantification. In contrast, relative scaling would potentially downplay or exaggerate local variations, especially if a site's maximum or minimum values are represented by outliers.

The approach proposed by McMichael et al. (2021) seems promising for regions or time periods with low or near absence of fire activity. However, for an intercomparison study like ours the chosen method must be able to also appropriately represent areas with substantial fire activity. These arguments lead us to choose z-scores.

Given your suggestion and to verify differences between the two methods, we produced new curves using PRS, as per McMichael et al. (2021) (Fig. R2.2) and PRS applying a base period (0.2 – 21 ka, PRS.bp). We compare these two relative scaling-based curves with our z-score curves. Importantly, we applied the base period to the PRS method to ensure consistency with our z-score calculations, which also rely on a defined base period. This, however, was not tested in McMichael et al. (2021).

[Figure]

**Fig. R2.2**. Comparison of charcoal influx data transformed into z-scores with proportional relative scaling (PRS) and proportional relative scaling with a 0.2 – 21 ka base period (PRS.bp). All curves use pre-binning half-window of 20 yr and half-window smoothing of 1000 yr.

[Figure]

**Fig R2.2**. Continued.

These comparisons show that despite changes in the amplitude of variability, all regions hold mostly coherent patterns of changes and a similar temporal structure. However, some key differences also arise. We briefly discuss them below.

Although all composites for NNeo indicate increased fire activity between 18 and 13 ka and during the LGM for CEB, z-scores and PRS.bp suggest a stronger rise, while PRS shows only a mild increase. The use of base period produces very similar trends between both methods, which is particularly important when including records with peak values outside our time of interest. Moreover, variabilities during the Pleistocene are more sensitive to site-specific data, due to the low availability of records.

On the other hand, in Amazonia, although the overall temporal patterns are similar across methods, a key divergence is observed: both PRS and PRS.bp suggest higher fire activity during the LGM compared to the Holocene, whereas z-scores indicate the opposite trend. In this case, z-scores seem more consistent with the known long-term fire history in the region. Several Amazonian records spanning Pleistocene ages suggest higher fire activity during the Holocene (Blaus et al., 2024; Bush et al., 2004; Colinvaux et al., 1997; Fontes et al., 2017; Hermanowski et al., 2012). An exception is southwestern Amazonia, where high charcoal concentrations occur during the Pleistocene, followed by a decline in the late Holocene as rainforest expanded (Burbridge et al., 2004).

Given that Amazonia exhibited important discrepancies between methods and is the region in which PRS is likely to best perform due to its rare frequency of fire events, we tested both z-scores and PRS.bp with multiple settings by applying combinations of sizes for bin half-window (binhw: 1500, 1000, 500, 300, and 40 years) and smoothing half-window (hf: 3000, 1500, 1000, and 400 years) (Fig. R2.3). Results show that PRS.bp is more sensitive to binhw variations, with Holocene base levels increasing systematically with larger bin sizes. Notably, while for binhw of 40 and 300 Pleistocene values are usually higher than those of the Holocene, the opposite is produced by applying binhw of 500, 1000 and 1500 years. In contrast, z-scores remain stable across different settings, consistently showing higher Holocene fire levels and no systematic variation linked to bin size.

[Figure]

**Fig. R2.3** Charcoal influx composites for Amazonia using z-scores and proportional relative scaling (PRS.bp; base period: 0.2–21 ka), applying different smoothing half-windows (hf) and bin half-window (binhw) lengths in years.

Considering these observations, z-scores appear more stable and better suited to capturing long-term trends in fire activity. While PRS seems very promising, particularly for regions such as Amazonia where fire activity is rare, it still requires further testing. For instance, how it performs across different regions and time spans, as well as its sensitivity to variations in base periods and binning parameters. Thus, we see strong arguments to keep z-scores. This brief discussion on the comparison between these two methods will be included in a Supplementary Information.

References:

Blaus, et al.: Climate, vegetation, and fire, during the last deglaciation in northwestern Amazonia, Quat. Sci. Rev., 332, 108662, https://doi.org/10.1016/j.quascirev.2024.108662, 2024.

Burbridge, et al.: Fifty-thousand-year vegetation and climate history of Noel Kempff Mercado National Park, Bolivian Amazon, Quat. Res., 61, 215–230, https://doi.org/10.1016/j.yqres.2003.12.004, 2004.

Bush, et al.: Amazonian paleoecological histories: one hill, three watersheds, Palaeogeogr. Palaeoclimatol. Palaeoecol., 214, 359–393, https://doi.org/10.1016/S0031-0182(04)00401-8, 2004.

Colinvaux, et al.: Glacial and Postglacial Pollen Records from the Ecuadorian Andes and Amazon, Quat. Res., 48, 69–78, https://doi.org/10.1006/qres.1997.1908, 1997.

Fontes, et al.: Paleoenvironmental dynamics in South Amazonia, Brazil, during the last 35,000 years inferred from pollen and geochemical records of Lago do Saci, Quat. Sci. Rev., 173, 161–180, https://doi.org/10.1016/j.quascirev.2017.08.021, 2017.

Hermanowski, et al.: Environmental changes in southeastern Amazonia during the last 25,000 yr revealed from a paleoecological record, Quat. Res., 77, 138–148, https://doi.org/10.1016/j.yqres.2011.10.009, 2012.

3-Relationship between tree cover, biomass, and fire

The interpretation proposed by the authors—linking positive correlations between tree cover and fire frequency to fuel-limited regimes, and negative correlations to moisture-limited regimes (e.g. lines 349-350; 421-422; 535-537; 554-555;574-575)—is broadly consistent with established ecological theory on fire-vegetation-climate interactions. This framework provides a useful lens for interpreting paleoecological data, particularly in ecosystems where fuel availability is a key constraint on fire activity.

Indeed, in ecosystems where fire regimes are typically fuel-limited, such as deserts, xeric shrublands, and dry savannas and grasslands, fire occurrence is constrained by low biomass production and the discontinuity of fuel, despite often dry climatic conditions (e.g., Krawchuk et al., 2009). In these cases, it is true that an increase in biomass is necessary to reach the threshold at which fire can propagate across the landscape (e.g., Pausas & Ribeiro, 2013).

However, it is important to recognize that this relationship is not linear. Once a certain level of biomass is reached—sufficient to produce continuous fuel loads—

further increases in biomass, particularly through increased tree cover, do not necessarily lead to higher fire frequency. In fact, dense woody cover can reduce fire frequency (which is acknowledged by the authors in lines 516-517) by suppressing the herbaceous layer, increasing shading and moisture retention, and creating microclimatic conditions less favorable to combustion (e.g., Staver et al., 2011).

Moreover, the assumption that increasing tree cover directly facilitates increased fire activity under fuel-limited conditions may not always hold. In some cases, both variables—tree cover and fire frequency—could increase independently as parallel responses to external climatic drivers. For instance, a scenario in which warmer temperatures promote tree expansion, while drier conditions simultaneously enhance fire frequency, could result in a positive correlation that does not necessarily reflect a causal, fuel-mediated link. Such a situation might be relevant to the patterns observed in some of the analyzed regions.

Conversely, in cases of negative correlations, it is important to consider that a reduction in fire frequency could also be a precondition for tree cover expansion, rather than its consequence (e.g., Staver et al., 2011). These alternative causal pathways underscore the importance of considering the directionality of the relationships and the potential influence of external climatic factors.

Additionally, it is crucial to emphasize that in many fire-prone ecosystems, especially savannas, herbaceous fuels—notably C4 grasses—are the primary drivers of fire regimes, while woody biomass plays a secondary role (e.g., Bond & Keeley, 2005). Therefore, correlations between tree cover and fire activity may not fully capture the dynamics of fuel availability and fire propagation (the authors seem to consider throughout the ms that arboreal cover (interpreted from AP) equals biomass growth and fuel availability).

Overall, the interpretations made by the authors are reasonable within their theoretical framework, but incorporating these additional ecological nuances could further strengthen the discussion and provide a more comprehensive understanding of the complex interplay between vegetation dynamics and fire regimes.

**Response #4.** Following specific comments from Reviewer #1 (Nicholas O'Mara), we will have fixed this generalization of AP as biomass. We will include in the methods that AP is relative to herbaceous vegetation:

"*Arboreal pollen (AP), which serves as an indicator of tree cover relative to herbaceous vegetation, was calculated as the percentage of woody taxa (trees and palms, considering taxa at the genus and family level) (TRSH and PALM) divided by the total sum of trees and shrubs (TRSH), palms (PALM), and herbs (UPHE), excluding mangrove and aquatic taxa, fern spores, and unidentified types.*"

We will also fix part of the CEB section 5.2.5 in order to account for the importance of herbaceous vegetation to fire dynamics, as follows:

5.2.5   Central-eastern Brazil (CEB): "*In CEB, long-term tree cover changes are negatively correlated with fire activity. This pattern points to a feedback mechanism in which herbaceous vegetation facilitates fire activity, while fire, in turn, contributes to the dominance of herbs by limiting tree encroachment. Conversely, moisture-driven development of woody formations leads to the suppression of fire, which further contributes to tree cover expansion.*"

Furthermore, we agree with the pointed concerns that fire activity and tree cover relationships are often multidirectional, and attributing clear causality is difficult, especially when both variables may be responding to the same external forcing. Thus, we will adapt the text to provide a more nuanced explanation about the potential forcings and tree cover-fire relationships:

5.2.6   Southeastern South America (SESA): "*This suggests that the intensified fire regime may have been driven by the long-term increase in fuel availability from woody biomass, or that both tree cover and fire responded to the same external forcing (i.e., deglacial warming). Nevertheless, in the short-term, fire likely acted as a limiting factor for tree cover development.*"

5.2.7   Extratropical Andes (ExTrAn) (this section before named "Southern Andes" will be renamed after changes related to our Response #3 to Reviewer #3, Paula A. Rodríguez-Zorro): "*The strong positive correlation between woody biomass and fire activity supports a fuel-limited fire regime in the region, and/or suggests that observed increases in tree cover and fire activity were both driven by deglacial warming (Fig. A1f). Nevertheless, peak fire activity in the region is currently achieved at intermediate levels of both woody and herbaceous biomass (Holz et al., 2012), supporting the role of increasing woody vegetation in creating optimal conditions for fire. Human populations started to expand after the ACR and likely also contributed to the intensification of the fire regime (Fig. 11d) (Perez et al., 2016; Salemme and Miotti, 2008).*"

4-Potential role of megafauna extinction:

One additional factor that may have influenced vegetation structure and fire regimes in the Neotropics during the late Quaternary is the extinction of megafauna at the Pleistocene-Holocene transition (or later - e.g. Faria et al., 2025). Large herbivores are known to play a critical role in shaping vegetation through grazing, browsing, and trampling, thereby modulating fuel loads and fire regimes (Gill et al., 2009; Doughty et al., 2016). The disappearance of these animals in (some parts of) South America may have contributed to changes in woody cover and fuel accumulation, potentially influencing fire dynamics independently or

synergistically with climatic and anthropogenic factors. While I understand that this topic may be beyond the primary scope of the manuscript, I wonder whether the authors considered this as a possible additional driver in some regions, or whether any of the available paleoecological records reflect such transitions.

**Response #5**. We appreciate the insightful comment. Although this aspect was not considered in our continental-scale analysis, it raises an important point that we will acknowledge in the revised text. Quantifying the impact of megafauna remains challenging due to the limited availability of downcore records across Neotropics that directly assess megafaunal population changes, vegetation and fire (e.g., Bush et al., 2022; Raczka et al., 2018, 2019; Rozas-Davila et al., 2016, 2021).

We also consider these ecological changes as secondary to the major climatic shifts associated with Termination 1, as many of the observed trends can be correlated with substantial changes in temperature and precipitation during this period and during the Holocene to increasing human activities. Nevertheless, it is plausible that megafauna played a competing role in limiting fuel accumulation and constraining the encroachment of woody taxa in open savannas and grasslands. We will include remarks on these effects in the discussion and include references on the role of megafauna and grazing.

For instance:

5.2.1 Northern Neotropics (NNeo): "*Although millennial-scale climate and vegetation changes importantly affected megafaunal populations in NNeo, the extent of megafaunal impacts in the region, and the consequences of their extinction on fire and vegetation dynamics remain unclear (Dávila et al., 2019; Rozas-Davila et al., 2021).*"

5.2.2 Amazonia: "*The impacts of megafaunal extinction also initiated long-term changes in both nutrient distribution and species turnover (Doughty et al., 2013, 2016a). However, its correlation with overall tree cover and fire regime changes remains elusive for the region.*".

5.2.5 Central-eastern Brazil (CEB): "*Additionally, human activity probably also contributed to tree cover and fire trends during this period, as archaeological records show well-established occupations from ca. 13 ka onwards and expanding population in the EH. Furthermore, the megafauna functional extinction during this period may have contributed to changes in vegetation composition (Raczka et al., 2018) and potential increase of fuel loads (e.g., Gill, 2014). This effect, however, may have been secondary, as a decrease in tree cover is opposite to the expected response by megafaunal extinction alone (Doughty et al., 2016b; Macias et al., 2014).*"

5.2.6  Southeastern South America (SESA): "*The presence of a diverse megafauna may have further contributed by restricting both woody encroachment and fuel accumulation (Furquim et al., 2024; Macias et al., 2014; Prates and Perez, 2021; da Rosa et al., 2023)*."

References:

Bush, et al.: A palaeoecological perspective on the transformation of the tropical Andes by early human activity, Philos. Trans. R. Soc. B Biol. Sci., 377, https://doi.org/10.1098/rstb.2020.0497, 2022.

Doughty et al.: The impact of the megafauna extinctions on savanna woody cover in South America, Ecography (Cop.)., 39, 213–222, https://doi.org/10.1111/ecog.01593, 2016.

Furquim, et al.: Interactive effects of fire and grazing on vegetation structure and plant species composition in subtropical grasslands, Appl. Veg. Sci., 27, 1–13, https://doi.org/10.1111/avsc.12800, 2024.

Gill.: Ecological impacts of the late Quaternary megaherbivore extinctions, New Phytol., 201, 1163–1169, https://doi.org/10.1111/nph.12576, 2014.

Macias, et al.: Grazing and neighborhood interactions limit woody encroachment in wet subtropical savannas, Basic Appl. Ecol., 15, 661–668, https://doi.org/10.1016/j.baae.2014.09.008, 2014.

Prates and Perez.: Late Pleistocene South American megafaunal extinctions associated with rise of Fishtail points and human population, Nat. Commun., 12, 1–11, https://doi.org/10.1038/s41467-021-22506-4, 2021.

da Rosa et al.: A Look into the Past: Fossils from the Campos Sulinos Region, in: South Brazilian Grasslands, Springer International Publishing, Cham, 45–81, https://doi.org/10.1007/978-3-031-42580-6_3, 2023.

Raczka, et al.: The collapse of megafaunal populations in southeastern Brazil, Quat. Res. (United States), 89, 103–118, https://doi.org/10.1017/qua.2017.60, 2018.

Raczka, et al.: A human role in Andean megafaunal extinction?, Quat. Sci. Rev., 205, 154–165, https://doi.org/10.1016/j.quascirev.2018.12.005, 2019.

Rozas-Davila, et al.: The functional extinction of Andean megafauna, Ecology, 97, 2533–2539, https://doi.org/10.1002/ecy.1531, 2016.

Rozas-Davila, et al.: When the grass wasn't greener: Megafaunal ecology and paleodroughts, Quat. Sci. Rev., 266, 107073, https://doi.org/10.1016/j.quascirev.2021.107073, 2021.

Final remarks

Overall, I believe this manuscript makes an important and valuable contribution to our understanding of long-term vegetation and fire dynamics in the Neotropics. It integrates multiple lines of evidence in a thoughtful and rigorous way, and I am confident it will be a useful reference for researchers working in this field. I appreciate the opportunity to revise this study and hope my comments are helpful to further refine this already excellent manuscript.

We sincerely thank Raquel F. Cassino for her thoughtful comments and the careful evaluation of our manuscript.

**Minor comments:**

Lines 237 - 241: replace "use" by "used"

Agree.

Line 436: "which suggests the influence "of" different mechanisms"

Agree.

Line 441: "Furthermore, while δ18O in ice and speleothem cores suggest primarily reflect rainy season precipitation" - remove "suggest".

Agree.

References:

Bond, W.J., & Keeley, J.E. 2005. Fire as a global 'herbivore': the ecology and evolution of flammable ecosystems. Trends in Ecology & Evolution, 20(7), 387–394.

Doughty, C. E., Wolf, A., & Malhi, Y. 2016. The legacy of the Pleistocene megafauna extinctions on nutrient availability in Amazonia. Nature Geoscience, 9, 800–803.

Faria, F. H. C., Carvalho, I. S., Araújo-Júnior, H. I., Ximenes, C. L., & Facincani, E. M. 2025. 3,500 years BP: The last survival of the mammal megafauna in the Americas. Journal of South American Earth Sciences, 153, 105367.

Gill, J. L., Williams, J. W., Jackson, S. T., Lininger, K. B., & Robinson, G. S. 2009. Pleistocene megafaunal collapse, novel plant communities, and enhanced fire regimes in North America. Science, 326(5956), 1100-1103.

Gosling, W. D., Maezumi, S. Y., Heijink, B. M., Nascimento, M. N., Raczka, M. F., van der Sande, M. T., Bush, M. B., & McMichael, C. N. H. 2021. Scarce fire activity in north and north-western Amazonian forests during the last 10,000 years. Plant Ecology & Diversity, 14(1), 89–99.

Krawchuk, M.A., et al. 2009. Global pyrogeography: the current and future distribution of wildfire. PLoS ONE, 4(4): e5102.

McMichael, C. N. H., Heijink, B. M., Bush, M. B., & Gosling, W. D. 2021. '. Frontiers in Biogeography, 13(1), e49431

Pausas, J.G., & Ribeiro, E. 2013. The global fire–productivity relationship. Global Ecology and Biogeography, 22(6), 728–736.

Staver, A.C., Archibald, S., & Levin, S.A. 2011. The global extent and determinants of savanna and forest as alternative biome states. Science, 334(6053), 230–232.

Citation: https://doi.org/10.5194/egusphere-2025-1424-RC2

---

## Author Comment (AC4)

The manuscript presented by Akabane et al., presents a synthesis of how vegetation and fire regimes in the Neotropics have responded to climatic changes over the last 21,000 years in seven subregions of the Neotropics. The authors used a modern analysis of vegetation distribution and fire activity in relation to current climatic conditions, complemented by a compilation of 243 vegetation and 127 fire records to assess changes over the past 21 ka BP.

The manuscript is well-structured and comprehensible, offering a significant contribution to the understanding of long-term ecosystem dynamics in the regions studied. Although there is a scarcity of paleo records in some areas, this research is a good example of the importance of databases and open data in enhancing our understanding of Neotropical ecosystems, and it calls for additional research to address the existing knowledge gaps.

**Response #1.** We thank Paula A. Rodríguez-Zorro very much for her evaluation of our manuscript.

GENERAL COMMENTS

The title proposed by the authors "Vegetation and fire regimes in the Neotropics over the last 21,000 years" suggest an extensive examination of the entire neotropical region. However, this study is restricted to seven specific subregions, omitting the northern Andes, parts of Bolivia and Chile, and large portions of Brazil and Argentina. This selection was partly due to data availability, yet it resulted in a focus on these seven subregions rather than a comprehensive analysis of the Neotropics.

**Response #2.** We appreciate the observation regarding the generality implied by our title. While it is true that our study focuses on seven subregions, we would like to emphasize that these regions collectively represent by far most of the pollen and charcoal records currently available from the Neotropics, covering a substantial portion of the region. Therefore, we believe that the chosen regions still allow us to draw meaningful conclusions about continental-scale vegetation and fire regime dynamics. Consequently, we feel that the title remains appropriate, even if it does not capture every local detail of the entire region.

In the section on vegetation settings, the authors provide a detailed description of the selected subregions, emphasizing the primary climatic drivers, such as the ITCZ, SASM, and ENSO. However, the analysis and discussion neglect the direct climatic influences from the Pacific Ocean (and records from the west flank of the Andes), particularly ENSO. The authors note in section 195: "Datapoints outside the defined subregions (black dots in Fig. 2a) were excluded from subregional analyses because they were either isolated or located outside subregional definitions, e.g., high

montane sites from NNeo >3000 m; low altitudes from CAn < 2200 m; or positioned on the Andean west flank, which has a distinct climate control relative to the east flank and Altiplano". The climatic dynamics influenced by the Pacific Ocean cannot be overlooked, particularly given that their effects can be detected globally (e.g. ENSO). If the analyzed records did not exhibit a clear signal, this should be explicitly stated. Similarly, if the sampling resolution is insufficient for the western flank, as has been acknowledged for other regions with limited data coverage such as Northeastern Brazil (NEB), this limitation should also be addressed. In the same line, it is unclear why ecosystems from high mountains, such as paramos, are excluded from the analysis. Several of them have proven useful for understanding past climatic variability (e.g. Haggemans et al., 2022, Espinoza et al., 2022, Ledru et al., 2022). It is worth mentioning that the northern part of the Andes was completely excluded in this study.

**Response #3.** Thanks for the relevant comment. We will include records from the Northern Andes (i.e., records located north of 0° latitude) down to 24 °S, and use all available sites above 2200 m a.s.l. Therefore, the section currently designated *Central Andes* section will be renamed to *Tropical Andes* to more accurately reflect its broader latitudinal scope. To maintain consistency in terminology, the section previously designated as *Southern Andes* will be renamed *Extratropical Andes*.

[Figure]

**Figure.** Cyan dots depict records which will be included in the analysis for the *Tropical Andes* section.

We will also include discussions on potential ENSO impacts. However, given the long-term scope of our study and the broad spatial and temporal coverage of our dataset, it is difficult to assess short-term variability. Moreover, the simultaneous intensification of ENSO during the late Holocene, after a damped mid Holocene phase (Carré et al., 2014; Koutavas et al., 2006; Moy et al., 2002) coincides with

increasing human influence on the landscape and land-use patterns, making it challenging to disentangle their respective effects (e.g., Nascimento et al., 2020). Nevertheless, we will briefly include discussions on how ENSO may have contributed to the observed changes and incorporate the updated ENSO activity curve from Laguna Pallcacocha (Mark et al., 2022) to Fig. 7 from the manuscript (see preliminary version of the updated figure below, Fig. R3.1). Additionally, we will introduce a new figure in the Appendix (Fig. R3.2), which separates the northern and southern Tropical Andes at 8°S. The discussion in the renamed section "5.2.3 Tropical Andes (TrAn)" will be slightly adapted to reflect these changes, as exemplified below.

[Figure]

**Fig. R3.1 Updated Fig.7**. Tropical Andes (TrAn) vegetation, fire, climate regimes, and human populations: (a) Speleothem δ18O from El-Condor (ELC), Cueva del Diamante (NAR), and Santiago (San) caves (Cheng et al., 2013; Mosblech et al., 2012). (b) Freshwater benthic diatom (%) from Titicaca Lake (Baker et al., 2001). (c) XRF-based reconstructed ENSO frequency from Laguna Pallcacocha (Mark et al., 2022). (d) Summed density probability of 14C ages from archeological sites in CAn (Goldberg et al., 2016) (N = 949). (e) Arboreal pollen (AP) and (f) charcoal influx z-scores composites using 1000-yr (green and red, respectively) and 400 yr (black) smoothing half-window. Gray areas represent 2.5th and 97.5th confidence intervals. (g) Number (#) of records with available pollen (green) and charcoal (red) data in a 400-yr time bin.

**5.2.3 Tropical Andes (TrAn)**

"The MH is characterized by a stepwise intensification of the fire regime, with tree cover remaining relatively stable, as indicated by our regional-integrating AP composite (Fig. 7d,e). However, subdividing TrAn at 8 °S in northern and southern sectors reveals distinct trends (Fig. A3): while fire activity increased across both regions, tree cover remained stable in the north but declines in the south. These trends are consistent with evidence of heterogeneous hydroclimatic patterns along Andes, with drier conditions recorded by lowered lake levels, mainly in the Altiplano region (Baker et al., 2001; Bush and Flenley, 2007; Hillyer et al., 2009) and persistent wet conditions recorded in central sectors of the TrAn (Bustamante and Panizo, 2016; Cheng et al., 2013; Polissar et al., 2013) (Fig. 7a). Modern climate patterns indicate that anomalies in equatorial Pacific SST can produce different regional impacts in precipitation by affecting moisture-laden easterlies and the Bolivian High (Garreaud et al., 2003; Poveda et al., 2020; Vuille, 1999). For instance, reduced ENSO variability has been suggested to decrease moisture balance in parts of the Andes, particularly in the southern TrAn, parts of the northern TrAn and NNeo (Fig. A3; Fig. 7c) (Polissar et al., 2013). Accordingly, we observe an apparent negative correspondence between the zonal Pacific SST gradient anomaly and tree cover trends during the EH and MH in southern TrAn, but no apparent control on northern TrAn (Fig. A3). Furthermore, while $\delta^{18}$O in ice and speleothem cores primarily reflect rainy season precipitation (Cheng et al., 2013; Vuille et al., 2003), fluctuations in lake levels are influenced by annual precipitation (Theissen et al., 2008). This may alternatively suggest that despite a gradual increase in summer precipitation throughout the Holocene, the MH featured a decrease in annual precipitation over the Altiplano. Nevertheless, the maintenance of some tree cover may indicate the persistence of moist microclimates (Ledru et al., 2013; Nascimento et al., 2019). These conditions, as well as anthropogenic activities, would have favored a regional intensification of the fire regime and decline in tree cover. By ca. 6 ka, agropastoral systems based on maize agriculture and llama herding became widespread (Nascimento et al., 2020)."

[Figure]

**Fig. R3.2. Figure that will be included in the Appendix as Fig. A3 – Tropical Andes (TrAn) subdivided in northern and southern sectors at 8 °S: (a)** East Pacific SST anomaly (Koutavas and Joanides, 2012; Lea et al., 2006). **(b)** Zonal SST gradient anomaly between east and west Pacific (Koutavas and Joanides, 2012). **(c)**) El Niño events per 100 years estimated from XRF data from Lake Pallcacocha (Mark et al., 2022). **(d)** Arboreal pollen (AP) and **(e)** charcoal influx z-scores composites from all records from TrAn located north of 8 °S. **(f)** Arboreal pollen (AP) and **(g)** charcoal influx z-scores composites from all records from TrAn located south of 8 °S. AP and charcoal-influx z-scores were produced using 1000-yr (green and red, respectively) and 400 yr (black) smoothing half-window. Gray areas represent 2.5th and 97.5th confidence intervals.

References:

Carré, et al.: Holocene history of ENSO variance and asymmetry in the eastern tropical Pacific, Science., 345, 1045–1048, https://doi.org/10.1126/science.1252220, 2014.

Koutavas, et al.: Mid-Holocene El Niño-Southern Oscillation (ENSO) attenuation revealed by individual foraminifera in eastern tropical Pacific sediments, Geology, 34, 993–996, https://doi.org/10.1130/G22810A.1, 2006.

Koutavas, A. and Joanides, S.: El Niño-Southern Oscillation extrema in the Holocene and Last Glacial Maximum, Paleoceanography, 27, 1–15, https://doi.org/10.1029/2012PA002378, 2012.

Lea, et al.: Paleoclimate history of Galápagos surface waters over the last 135,000 yr, Quat. Sci. Rev., 25, 1152–1167, https://doi.org/10.1016/j.quascirev.2005.11.010, 2006.

Mark, et al.: XRF analysis of Laguna Pallcacocha sediments yields new insights into Holocene El Niño development, Earth Planet. Sci. Lett., 593, 117657, https://doi.org/10.1016/j.epsl.2022.117657, 2022.

Moy, et al.: Variability of El Niño/Southern Oscillation activity at millennial timescales during the Holocene epoch, Nature, 420, 162–165, https://doi.org/10.1038/nature01194, 2002.

Nascimento, et al.: The adoption of agropastoralism and increased ENSO frequency in the Andes, Quat. Sci. Rev., 243, 106471, https://doi.org/10.1016/j.quascirev.2020.106471, 2020.

In the methodology, it is not entirely clear how the authors have standardized the pollen data. It draws my attention to the part where they have "excluded mangrove and aquatic taxa, fern spores and unidentified types". In the case of fern spores, the authors should evaluate or clarify if they have considered tree ferns in their AP composite. In some regions, such as the Atlantic and Andean Forests, tree ferns, like *Cyathea*, *Lophosoria*, or *Dicksonia* species thrive under specific conditions, with water availability being a common factor. In pollen records, they serve as key indicators of humid conditions and are included in the pollen sum (Kesler et al., 2011, Salazar et al., 2013, de Gasper et al., 2021).

**Response #4**. We addressed the concern regarding the standardization of the methodology in our Response #2 to Reviewer #2 (Raquel F. Cassino).

Regarding tree ferns, while we recognize their value as indicators of humid conditions in regions such as the Andes and Atlantic Forests, we excluded them from the pollen sum due to inconsistent reporting across records. Including them would have limited comparability among sites. To maintain a standardized dataset suitable for regional-scale analyses, we chose to exclude all fern spores from the pollen sum. We acknowledge that this may compromise some site-specific interpretations, but it improves the overall inter-comparability of the dataset.

The following paragraph will be included in the method section:

"*Although tree ferns can serve as important indicators of humid conditions in regions such as the Andes and Atlantic Forests, we excluded them from the pollen sum due to inconsistent reporting across records, in order to ensure a standardized dataset suitable for regional-scale analyses.*"

Regarding fire dynamics, it is unclear how the authors determined high or low fire activity. The methodology from the compiled data is not evident. Additionally, the type of data used to reconstruct fire regimes, whether macrocharcoal, microcharcoal, or both is not specified.

**Response #5.** We use the methodology described in Blarquez et al. (2014). This method has also been used in other compilation studies (Daniau et al., 2012; Marlon et al., 2013; Mooney et al., 2011; Power et al., 2010). The composite z-score curve represents interpolated values using a LOWESS derived from individual site z-scores, which indicate how far charcoal influx at each site deviates from the mean. Positive and negative composite z-scores correspond to periods of above- and below-average charcoal influx, respectively, which we interpret as stronger or weaker fire regimes. While our approach does not allow us to distinguish specific aspects of the fire regime such as intensity, severity, frequency, seasonality, spatial extent of the burned vegetation, we use the broader terms "fire regime" or "fire activity" to reflect the general response of fire to environmental changes, which is directly interpreted from values of charcoal influx. Still, prompted by this comment, we will improve a specific passage of the methods by clarifying this aspect of our interpretations as follows:

"*Changes in charcoal records can be linked to past fire activity and used to infer shifts in fire regimes. While our approach does not allow to resolve specific components of the fire regime (e.g., intensity, severity, frequency, seasonality, spatial extent of burned vegetation), we consider fire regime as the collective changes in charcoal influx trends within a given region over long timescales. We use fire activity as a more general term to describe variability in charcoal influx.*".

Regarding charcoal descriptions, it was indeed insufficiently explained in the manuscript. Both macroscopic and microscopic charcoal data were treated jointly. We will provide further methodological details about charcoal records (see our Response #3 to Reviewer #1 Nicholas O'Mara).

References:

Blarquez et al.: Computers & Geosciences paleofire : An R package to analyse sedimentary charcoal records from the Global Charcoal Database to reconstruct past biomass burning $, Comput. Geosci., 72, 255–261, https://doi.org/10.1016/j.cageo.2014.07.020, 2014.

Daniau et al.: Predictability of biomass burning in response to climate changes, Global Biogeochem. Cycles, 26, 1–12, https://doi.org/10.1029/2011GB004249, 2012.

Marlon et al.: Global biomass burning: A synthesis and review of Holocene paleofire records and their controls, Quat. Sci. Rev., 65, 5–25, https://doi.org/10.1016/j.quascirev.2012.11.029, 2013.

Mooney et al.: Late Quaternary fire regimes of Australasia, Quat. Sci. Rev., 30, 28–46, https://doi.org/10.1016/j.quascirev.2010.10.010, 2011.

Power et al.: Fire history and the global charcoal database: A new tool for hypothesis testing and data exploration, Palaeogeogr. Palaeoclimatol. Palaeoecol., 291, 52–59, https://doi.org/10.1016/j.palaeo.2009.09.014, 2010.

Similarly, the authors should pay careful attention to the interpretation of fire dynamics in ecosystems such as savannas, in which the fuel for fire primarily comes from grasses. The authors used AP to determine the available biomass to be burned;

however, for these types of systems, grasses and herbs should also be considered (Alvarado et al., 2020).

**Response #6**. Thanks for this comment. We will revise parts of the text to more clearly distinguish between the terms "biomass" and "tree cover". We agree with the observation, and this is particularly evident for central-eastern Brazil, where fire activity has an opposite pattern to AP. Please, also see our Response #4 to Reviewer #2 (Raquel F. Cassino).

5.2.5 Central-eastern Brazil (CEB) "*In CEB, long-term tree cover changes are negatively correlated with fire activity (Fig. 9b,c, Fig. A1d). This pattern points to a feedback mechanism in which herbaceous vegetation facilitates fire activity, while fire, in turn, contributes to the dominance of herbs by limiting tree encroachment. Conversely, moisture-driven development of woody formations leads to the suppression of fire, which further contributes to tree cover expansion (Fig. 9b,c, Fig. A1d).*"

I hope these suggestions are useful to complement this outstanding review, and I congratulate the authors on their efforts to contribute to the understanding of Neotropical ecosystems at different time scales.

We thank Paula A. Rodríguez-Zorro very much for all observations and comments about our manuscript.

REFERENCES

Steiger, N.J., Smerdon, J.E., Seager, R. *et al.* (2021). ENSO-driven coupled megadroughts in North and South America over the last millennium. *Nat. Geosci.* 14, 739–744

Hagemans et al., (2022). Intensification of ENSO frequency drives forest disturbance in the Andes during the Holocene. *Q. Sci. Rev.* 294, 107762

Espinoza, I. G., Franco-Gaviria, F., Castañeda, I., Robinson, C., Room, A., Berrío, J. C., et al. (2022). Holocene fires and ecological novelty in the high colombian cordillera oriental. *Front. Ecol. Evol.* 10:895152.

Ledru, M.-P., Aquino-Alfonso, O., Finsinger, W., Samaniego, P., & Hidalgo, S. (2022). Changes in the vegetation and water cycle of the Ecuadorian páramo during the last 5000 years. *The Holocene*, *32*(9), 950-963.

Kessler, M., Kluge, J., Hemp, A., Ohlemüller, R. (2011). A global comparative analysis of elevational species richness patterns of ferns. *Glob. Ecol. Biogeogr*. 20, 868–880.

Salazar, L., Homeier, J., Kessler, M., Abrahamczyk, S., Lehnert, M., Krömer, T., & Kluge, J. (2013). Diversity patterns of ferns along elevational gradients in Andean tropical forests. *Plant Ecology & Diversity*, *8*(1), 13–24

De Gasper, A. L., Grittz, G. S., Russi, C. H., Schwartz, C. E., Rodrigues, A. V. (2021). Expected impacts of climate change on tree ferns distribution and diversity patterns in subtropical Atlantic Forest. *Perspect. Ecol. Conserv.* 19, 369–378.

Alvarado, S. T., Andela, N., Silva, T. S., and Archibald, S.(2020) Thresholds of fire response to moisture and fuel load differ between tropical savannas and grasslands across continents, *Global Ecol. Biogeogr.*, 29, 331–344

---

## Author Response (AR1)

Dear Editor, dear Reviewers,

We are very grateful for the positive evaluation of our manuscript and the detailed feedback provided by the three reviewers. We greatly appreciate the constructive comments and suggestions, which encouraged us to improve the clarity and quality of the manuscript. Some of the main changes address the following points:

- 1. A more cautious and precise use of the term *biomass* throughout the manuscript.
- 2. Expansion of the former "Central Andes" section to include records from the northern Andes, now renamed "Tropical Andes." This update involved moderate revisions to the corresponding discussion and the addition of a new figure in the Appendix to enhance regional detail.
- 3. Inclusion of brief discussions on the potential influence of megafauna, although such effects are generally not detectable within the scope of our regionally integrated analysis.
- 4. Addition of figures in the Appendix to improve clarity regarding the underlying data, along with a new supplementary section briefly comparing different methodological approaches.
- 5. Addition of further information to the supplementary table, including raw data from the composite curve and growth habit information for the taxa used.
- 6. Enhancements in the description of methods to improve transparency and reproducibility.

Other minor corrections and changes throughout the text are highlighted in the version with tracked changes.

Please find below our detailed, point-by-point responses, which follow the structure and are in the same line of our previous replies to the reviewers. Comments from the reviewers are shown in pink, and our responses are provided in black. Line numbers refer to the clean version of the revised manuscript.

On behalf of all coauthors,

Thomas Kenji Akabane

**Reviewer #1 (Nicholas O'Mara)**

**Review of Earth System Dynamics manuscript: egusphere-2025-1424**

This new compilation is impressive, and the analysis is thorough and detailed. I really appreciate the authors efforts to parse the records by climate/ecology to undergo a nuanced dissection of the competing influences of many factors on vegetation structure and fire regime and how these differ across geographies. The manuscript is very well written. They frame and motivate the problem well, the arguments are logical, the figures are clear and impactful, and it is overall quite pleasant to read. This study warrants speedy publication in Earth System Dynamics following minor revisions. I break down my review into major overarching comments followed by in-line comments and recommendations.

**Response #1**. We sincerely appreciate the positive assessment of our manuscript.**

**Overarching comments:**

Throughout the manuscript, the term "biomass" appears to be used interchangeably with "tree cover". As a means of estimating vegetation structural change, the authors use the fraction of arboreal pollen in sediment cores. This method estimates the fraction of vegetation in a region which is composed of trees, however this is not a measure of biomass *per se*. All else equal, more trees on a landscape would equate to more biomass, but a ratio alone does not tell you this. Grasses can make up significant portions of the total biomass of ecosystems, particularly in tropical savannas (*e.g.*, Cerrado). The authors should take a careful look at the instances where they make claims about changes in biomass when they are actually measuring tree pollen fraction to infer changes in tree cover. Grass biomass is an important fuel source especially in tropical savannas like the Cerrado, so I urge caution to the authors on broadly equating increased tree cover with biomass in the context of fuel availability. I flag such instances in the in-line comments.

**Response #2**. Thank you very much for your observation. We modified such instances accordingly, following your in-line comments by clearly stating woody or arboreal biomass when directly related to our compiled arboreal pollen data.

The description of the charcoal records is insufficient. The authors spend a decent portion of their methods section describing the dominant pollen types in the records that they compiled for each region but only list the number of records for charcoal without further description. Charcoal comes in many forms which record different aspects of fire regimes across multiple spatial scales. For instance, are all of the charcoal in the synthesis microcharcoal? Or are macrocharcoal particle records also included? One must read between the lines and look at the column title in the

supplement table to infer this. A more complete description of the charcoal records including at a minimum the size fraction of the records used in this synthesis is needed.

**Response #3**. We agree that descriptions regarding charcoal data could be clarified and improved. Accordingly, we have revised the methods to ensure greater accuracy on this topic. More specifically, after standardization, we analyzed both macroscopic (>100 $\mu$ m) and microscopic (<100 $\mu$ m) charcoal data jointly, as performed in Power et al. (2008, 2010), Mooney et al. (2011), Gosling et al. (2021). The following sentence was included:

"For charcoal composites, records containing both micro- (< 100  $\mu$ m) and macro- (> 100  $\mu$ m) particles were included. Charcoal raw counts were converted into concentrations and then to influx using site-specific sedimentation rates to account for differences in sedimentation rates across sites (Marlon et al., 2016). Changes in charcoal records can be linked to past fire activity and used to infer shifts in fire regimes (Power et al., 2008; Daniau et al., 2010; Marlon et al., 2016)." (lines 219-222)

**References**

Gosling, et al.: Scarce fire activity in north and north-western Amazonian forests during the last 10,000 years, Plant Ecol. Divers., 14, 143–156, https://doi.org/10.1080/17550874.2021.2008040, 2021.

Marlon, et al.: Reconstructions of biomass burning from sediment-charcoal records to improve data-model comparisons, Biogeosciences, 13, 3225–3244, https://doi.org/10.5194/bg-13-3225-2016, 2016.

Mooney, S. D., et al.: Late Quaternary fire regimes of Australasia, Quat. Sci. Rev., 30, 28–46, https://doi.org/10.1016/j.quascirev.2010.10.010, 2011.

Power, M. J., et al.: Changes in fire regimes since the last glacial maximum: An assessment based on a global synthesis and analysis of charcoal data, Clim. Dyn., 30, 887–907, https://doi.org/10.1007/s00382-007-0334-x, 2008.

Power, M. J., et al: Fire history and the global charcoal database: A new tool for hypothesis testing and data exploration, Palaeogeogr. Palaeoclimatol. Palaeoecol., 291, 52–59, https://doi.org/10.1016/j.palaeo.2009.09.014, 2010.

The use of charcoal/pollen ratio records in this context surprises me. The authors say they multiply such records by the sedimentation rates of the cores to get an influx like unit, but this cannot be done. For such data to be ecologically meaningful, either (1) the pollen accumulation rate would have to be linearly correlated with the sedimentation rate and not a product of vegetation coverage within the watershed of the lake or (nearby river in the case of marine cores), or (2) the pollen accumulation rate would have to have such low variance that the change in charcoal accumulation rate drives the observed signal. Without convincing evidence in support of either of those scenarios, one cannot expect the charcoal numbers in these ratios to reflect changes in burning on the landscape. I suggest the authors remove such records from the compilation.

Response #4. We appreciate the thoughtful comment regarding the use of pollen/charcoal ratios in our analysis. We agree with the argument, and we excluded the ratio-based records from the z-score composite curves to maintain methodological consistency. Only four records were based on the pollen/charcoal ratio, two of them were not included in any subregion, one was included in NEB, and one in NNeo. However, for NEB where no composite curve was generated due to the scarcity of available charcoal data, we kept the pollen/charcoal ratio for qualitative comparison (red curve in Fig. R1.1e – see below). This is kept as simple charcoal/pollen ratios z-scores to provide at least some regional perspective. Additionally, two other curves were included in the new Fig. 8 (lines 557-563; Fig. R1.1a,e), one from a record containing both charcoal and pollen data (Ledru et al., 2006) and another of  $\delta D$ -based reconstructed precipitation from a marine core off NEB.

Fig. R1.1. Updated Fig. 8 – Northeastern Brazil (NEB) vegetation, fire, climate regimes, and human occupation: (a)  $\delta D \, n$ -C29 alkane from GeoB16202-2 (Mulitza et al., 2017). (b) Speleothem  $\delta^{18}O$  from Rio Grande do Norte cave (RN) (Cruz et al., 2009). (c) Summed density probability of 14C ages from archeological sites in NEB (N = 542) (Araujo et al., 2025). (d) Arboreal pollen (AP) z-scores composites using 1000-yr (green) and 400 yr (black) smoothing half-window. Gray areas represent 2.5th and 97.5th confidence intervals. (e) Charcoal influx z-scores from single sites (yellow: Ledru et al. (2006); brown: De Oliveira et al., 1999; red: Bouimetarhan et al., 2018). (f) Number (#) of records with available pollen data in a 400-yr time bin.

**Reference:**

Ledru, M. P., et al.: Millenial-scale climatic and vegetation changes in a northern Cerrado (Northeast, Brazil) since the Last Glacial Maximum, Quat. Sci. Rev., 25, 1110–1126, https://doi.org/10.1016/j.quascirev.2005.10.005, 2006.

Figure 1 would benefit from a panel plotting either the fire radiative power or burned area data used in the modern analysis. The amount of burning across these biomes is highly variable. A map of where fires occur today would help the reader in interpreting the paleorecord compilations presented here.

**Response #5**. Thank you for the suggestion. We have now included fire activity along with vegetation distribution to Fig 1a from the manuscript.

I would encourage the authors to emphasize that the majority of the sites in this study, except those in region 5 (Central Eastern Brazil) in the Cerrado, are currently situated in more forested regions that do not burn as much as tropical savannas. This would tie in within the results of the modern analysis (Figure 3) to show that one might expect more fire in the past if grasses where a more dominant fraction of the local vegetation.

Response #6. This helpful comment prompted us to strengthen the connection between Section 5.1, "Modern climate-fire-vegetation relationships", and the subsequent discussion of regional patterns. We would like to emphasize, however, that some areas currently covered by mostly forested vegetation were dominated by more open vegetation types in the past. This is particularly evident in Southeastern South America (SESA) (Fig. R1.2) and Southern Andes (SAn). In SESA, for example, we observe a shifting relationship between tree cover and fire activity over time, with both arboreal pollen (AP) and charcoal increasing during Termination 1, followed by a continued rise in AP and stabilization of fire activity. Therefore, while only the Cerrado and high-altitude Andes are predominantly covered by open vegetation under modern conditions, regions such as SESA and SAn were also covered by open vegetation in the past and likely exhibited different fire-vegetation dynamics (e.g., Fig R1.2).

**Fig. R1.2** – Correlation between arboreal pollen z-scores and charcoal influx z-scores in Southeastern South America (SESA).

"These fire-prone conditions are typical of tropical savannas and grasslands (Fig. 3a,b), such as in CEB, where a regular frequency of fire events is key in maintaining

biodiversity and the physiognomy of vegetation (Bernardino et al., 2022; Mistry, 1998). In modern day CEB, however, fire regime is mostly limited by fuel moisture (moisture-limited condition, negative biomass-fire correlation) (Alvarado et al., 2020). In arid and semiarid environments, such as NEB, or high-altitude areas, such as higher areas of TrAn, fire activity is hindered due to biomass limitation (fuel-limited conditions, positive biomass-fire correlation) (Fig. 3a-e). On the other hand, under wet conditions of tropical rainforests, such as Amazonia or parts of NNeo, fire activity is limited due to constant fuel moisture (moisture-limited condition) (Fig. 3a,b)." (lines 371-378)

Figures 12 and 13. I really like the time snapshot analysis presented in these figures. However, it is a little bit difficult to have to read across the panels and compare the colors between points to see if the z-scores increased or decreased through time by comparing the color to the previous time slice. When I first looked at these maps, I expected that they were displaying the trends rather than the mean z-scores for the time slice. One minor change you could make to these figures would be to change the markers depending on whether the mean z-score increased or decreased compared to the last time slice. E.g., upward triangle for increase above some threshold between 21-19 to 19-14.8, downward triangle for decrease below some threshold, and circle for no change outside of some threshold. This would really help as the reader is going through your later portion of the discussion when you are providing a broad overview of the trends through time. Additionally, the time slice labels are backward, they should be in chronological order, e.g., "(c) 21 – 19 ka". You have it right in the figure caption, just reversed in the panel labels.

**Response #7**. We improved the description on how to interpret these maps in the methodology.**

"Additionally, we produced maps displaying site-specific AP percentages and z-scores for both AP and charcoal influx for all data points to illustrate spatial patterns across time slices of the last 21 ka. Each site-specific data point represents a mean z-score calculated relative to its own long-term mean (base period: 21 to 0.2 kyr BP). As a result, the colors of the dots within a single map reflect deviations from local baseline conditions and should not be directly compared across sites to infer geographic gradients or absolute levels of fire activity or tree cover. However, clusters with similar z-score trends, such as consistently positive values in a region, may indicate homogeneous responses and coherent regional patterns, for example a general expansion in tree cover or intensification of the fire regime." (lines 251-257)

We also fixed the backward labels mentioned. However, we prefer not to represent changes using different symbols, as this may introduce even more complexity. In some cases, sites may not display continuous changes between time slices due to, e.g. hiatuses or sampling resolution, which would require an additional symbol to

distinguish these instances from increases, decreases, or stable conditions. We decided to use this approach as it has already been successfully used in other studies (e.g., Marlon et al., 2016; Mooney et al., 2011; Power et al., 2008). As a side note, in Power et al. (2008), triangles are used, but simply to indicate positive or negative z-scores for a given timeslice.

**References:**

Marlon, et al. Reconstructions of biomass burning from sediment-charcoal records to improve data-model comparisons, Biogeosciences, 13, 3225–3244, https://doi.org/10.5194/bg-13-3225-2016, 2016.

Mooney, S. D. et al.: Late Quaternary fire regimes of Australasia, Quat. Sci. Rev., 30, 28–46, https://doi.org/10.1016/j.quascirev.2010.10.010, 2011.

Power, et al.: Changes in fire regimes since the last glacial maximum: An assessment based on a global synthesis and analysis of charcoal data, Clim. Dyn., 30, 887–907, https://doi.org/10.1007/s00382-007-0334-x, 2008.

Regarding data availability, I suggest that the authors make the smoothed z-score time series of AP and charcoal influx available in the supplement or permanently stored in a public archive. These new synthesis curves are valuable information for paleoclimatologists, paleoecologists, anthropologists, climate modelers, *etc.*, who might wish to compare them with their data. As I am sure the authors are aware from the webplotdigitizing they did for this study, it is always a relief when the key data for a paper are easily accessible for future analysis and plots don't need to be unnecessarily and painstakingly recreated.

**Response #8. All composite curves are now provided as Supplementary data.**

**In-line comments:**

19-20: I suggest you remove the "in the one hand" and "on the other hand" they are just not necessary to the point of the sentence which is already clear and concise.

**Done.**

24-25: Perhaps add "with additional impacts from human activity" to the end of the sentence which starts with "Temperature ..."

**Done.**

26: "Biomass growth" this should be "tree growth"

**Done, but we opt for "arboreal growth".**

46: "process" should be "processes"

**Done**

50: Glacial/interglacial cycles were occurring (although much more muted) in the Neogene. I suggest you change this to something like "onset of pronounced glacial/interglacial cycles in the Quaternary"

**Done.**

51: Change "were responsible for" to "played a significant role in"

**Done**

52: Please clarify here if you mean in setting the modern ecosystem distributions or modulating changing ecosystem distributions through time.

**Done.**

56-57: "Weaker fire regime" is not very clear to a general reader. I am okay with the use of weaker and stronger fire regimes throughout the paper, but take a sentence here to explain what characteristics constitute and "weak" versus "strong" fire regime. Additionally, you could maybe also clarify also here what "fire activity" means because you use this general term in the text as well.

Done. We provided some explanation in the method section, to avoid a break in the introduction for explaining the concept. While our approach does not allow us to distinguish specific aspects of the fire regime such as intensity, severity, frequency, seasonality, spatial extent of the burned vegetation, we use the broader terms "fire regime" or "fire activity" to reflect the general response of fire to environmental changes, which is directly interpreted from values of charcoal influx.

"Changes in charcoal records can be linked to past fire activity and used to infer shifts in fire regimes (Power et al., 2008; Daniau et al., 2010; Marlon et al., 2016). While our approach does not allow to resolve specific components of the fire regime (e.g., intensity, severity, frequency, seasonality, spatial extent of burned vegetation), it allows to identify collective changes in fire recorded as changes in charcoal influx within a given region over a long timescale." (lines 221-225).

Also see our Response #5 to Reviewer #3 (Paula A. Rodríguez-Zorro).

81 and throughout: You interchange "mm.yr-1" and "mm yr-1" I recommend you adopt the second in all cases, the added period is unnecessary.

**Done.**

86-87: "...marked by weak seasonality with mean monthly temperatures ranging between 25 and 27 °C and mean annual precipitation of..." would be better as a comma-separated list: ""...marked by weak seasonality, mean monthly temperatures ranging between 25 and 27 °C, and mean annual precipitation of..."

**Done.**

93-94: It is not clear to me which positive feedback loop you are referring to. *E.g.* fire impacts on vegetation structure and knock-on effects on temperature and future fire likelihoods, fires emitting greenhouse gases leading to overall warming and thus more fires, *etc.* Please clarify.

Done. We changed to "Persistent moist conditions of the rainforest naturally inhibit wildfires. However, initial fire events whether triggered by severe droughts (e.g., related to El Niño), ongoing climate changes, and/or human impacts further increase forest flammability through canopy degradation and fuel accumulation. This favors subsequent fire events, thereby fostering a positive feedback loop (Brando et al., 2020; Bush et al., 2008; Cochrane et al., 1999; Nepstad et al., 1999)." (lines 94-97)

99: "area" should be "areas"

**Done.**

103-104: "moist-laden" should be "moisture-laden"

**Done.**

199: I suggest you change "...open grasslands to closed shrublands and woodlands..." to "...open grasslands and savannas to closed shrublands and woodlands..."

**Done.**

121: "...the occurrence South Atlantic..." should be "...the occurrence of the South Atlantic..."

**Done.**

125-145: This is largely a style choice, so up to you, but in all preceding paragraphs you list temperature ranges from low to high (which is convention) but here you list high to low. I see that you might be doing this intentionally as you describe the regions from north to south, but it is a little weird to read temperature ranges from 16 to  $3\,^{\circ}$ C.

Thank you for noting this. The reversed temperature sequence (16 to 3°C) was indeed intentional to match the north-to-south latitudinal progression (32°S to 55°S). We adapted the text to "Mean annual temperature ranges from 3 to 16 °C between the latitudes 55 and 32°S". (line 140)

140: "...southward-displaced..." should be "...southwardly displaced..."

We slightly modified this part to "Precipitation is controlled by the Southern Westerly Winds (SWW), which shift southward during austral summer, [...]" (line 143)

165: For example, here is the only place in the text you mention microcharcoal. See overarching comments for method recommendations.

This comment was addressed in the overarching comments above.

201-203: "For charcoal composites, ..." You need to either provide strong justification that this is a viable method or remove such records from the compilation. See overarching comment.

**Done.**

295: "Despite represented by..." should be "Despite being represented by..."

**Done.**

303: Is this a mistaken paragraph break at the end of this line?

This was intentional as an introduction sentence for the following paragraphs in which we describe the distinct regional patterns. We included a ":" to the end of this line to make it clearer. (line 345)

315: Figure 4 should have y-axis labels. A single label common for all subplots would be fine. The two columns appear unnecessarily squished together horizontally; I think you can add a little separation between the two which should give ample room for the y-axis labels.

Indeed. Thank you for the suggestion. We have now fixed Fig. 4 accordingly.

341: "...high level..." and "...fire regime..." should be "...high levels..." and "...fire regimes..."

**Done.**

349: "p-values" can just be "p"

**Done.**

351: "condition" should be "conditions"

**Done.**

354: drop the "and" before "likely"

**Done.**

359: "...coeval to..." should be "...coeval with..."

**Done.**

357-358: I do not really see a slope break? The decline appears to be part of the larger trend of declining AP % since the EH. So, I would avoid calling it abrupt.

We removed the word "abrupt." Our intention was to convey that the Late Holocene decline in tree cover appears more pronounced than the gradual decrease observed since the Early Holocene, which we interpret as a shift likely amplified by direct human activity.

378: "...featured a reduced..." should be "...featured reduced..."

**Done**

380-382: I thought Amazonia is generally moisture limited, so isn't it surprising that low rainfall leads to low fire activity?

Yes, indeed, multiple interacting forcings contribute to a complex response. For example, the Amazon is a highly heterogeneous and predominantly fire-limited ecosystem, with marked west-east and north-south gradients. During the LGM, eastern Amazonia is interpreted to have been drier (Häggi et al., 2017; Wang et al., 2017) while western Amazonia was wetter, or as wet as modern conditions (Baker et al., 2001; Cheng et al., 2013). In addition, the southern and eastern Amazon rainforests were relatively smaller compared to their pre-Industrial extension (Mayle et al., 2000), possibly as a consequence of lower CO2 (Maksic et al., 2022) and drier conditions (Fontes et al., 2017).

Despite these regional contrasts, charcoal records consistently indicate low fire activity during the LGM and part of Termination 1 across the eastern (Hermanowski et al., 2012), central-northern (Blaus et al., 2024; Bush et al., 2004) and southern (Cordeiro et al., 2014; Fontes et al., 2017)

Amazonia. A different scenario is reported for southwestern Amazonia (Burbridge et al., 2004) with decreasing fire activity towards the Holocene. Over eastern and southern Amazonia, forest vegetation was largely replaced by savannas during the late Pleistocene. This seemingly counterintuitive pattern may be explained by colder temperatures, reduced convective activity (and consequently fewer lightning ignitions). We incorporated this point into the discussion. (lines 433-435)

Also, what do you think happened ca. 19-17 ka? Why are the z-scores for charcoal influx so low? It's hard to tell from the figure, but does the number of records go to zero at points in this interval? It

would be good to comment on this. And perhaps obscure portions of the curve which might have very low confidence due to lack of sufficient data.

This is likely an artifact resulting from the scarcity of data during this period, which makes the composite curve highly unstable and sensitive to specific records. Thus, we are not interpreting such features. The negative value reaches –2.2.

We have now included in the method section the following note of caution:

"In general, caution is warranted when interpreting trends during periods with wide confidence intervals or when the composite curve approaches the upper or lower bounds of the confidence interval, or exhibits outlier shifts. These cases usually relate to periods with few records and indicate greater uncertainty and sensitivity to individual records, thus reflecting local variability of specific sites. (lines 239-242).

517: "Biomass growth" should be "tree growth"

Done.

517-518: "moist-laden" should be "moisture-laden"

Done.

535-536: I would recommend "biomass" in both cases should be "tree cover"

Done for the second case. In the first case we changed to woody biomass.

545: "biomass" should be "tree cover"

Changed to arboreal biomass.

566: "This was likely consequence from significantly..." should be "This was likely the consequence of significantly..."

Done.

568: "tree cover increases along warming..." should be "tree cover increases along with warming..."

Done.

574: "biomass" should be "tree cover"

Changed to woody biomass.

589-590: The human population was already quite high by 7 ka, so why is the influence on fires delayed until 3.5 ka?

Humans were very likely to influence fire during the Mid-Holocene and even earlier. However, their regional impact became more evident (in our results) when both tree cover and humidity increased over the region and, unexpectedly, fire activity also increased. This likely suggests a superimposed impact of human activity on the background natural trend.

604: "biomass" should be "tree biomass"

Done.

606: I think it is worth pointing out to the readers that Haas et al., (2023) is a modeling study and we do not yet fully understand the impacts of low atmospheric  $CO_2$  on global fire regimes.

Done.

Overall, I really enjoyed this well-written and interesting paper, and I am excited to see the manuscript published following these revisions. Nice work!

Thank you very much for the throughout revision and insightful comments.

**Reviewer #2 (Raquel Franco Cassino)**

Review of the manuscript entitled "Vegetation and fire regimes in the Neotropics over the last 21,000 years"

The manuscript presents an excellent and timely synthesis, combining analyses of modern data with an extensive compilation of fossil records and archaeological evidence. The work is well written, clearly organized, and supported by high-quality figures that effectively convey the results.

**Response #1.** We are grateful for the comments from Raquel F. Cassino on our manuscript.**

I would like to offer a few comments and questions that I hope may help to further strengthen the manuscript:

**1-Definition of arboreal pollen**

The authors calculated arboreal pollen (AP) percentages as the sum of woody taxa (trees and palms), excluding mangrove and aquatic taxa, fern spores, and unidentified types. This approach is widely used in paleoecological studies and serves as a valuable proxy for reconstructing past vegetation structure. However, given the floristic complexity of tropical ecosystems, I would like to kindly suggest that the authors provide further clarification regarding the taxonomic criteria used to define AP. Specifically, many plant families in these ecosystems include both arboreal and non-arboreal life forms, which may significantly influence AP percentages depending on how these groups were categorized. It would be helpful to know how the authors distinguished between arboreal and non-arboreal taxa within such families, and whether any standardized criteria were applied in this process.

For instance, I wonder whether the palm *Mauritia flexuosa* was considered part of the AP in the Cerrado records. Given that Mauritia palms are often highly abundant in local swamp environments (veredas), their inclusion could potentially inflate AP percentages without necessarily indicating a broader regional forest expansion. Clarifying this point would be particularly valuable for interpreting AP trends in relation to regional woody cover dynamics. Providing these additional details could enhance the transparency and reproducibility of the study, and also refine the paleoecological interpretations drawn from the AP trends.

**Response #2.** The ecological types are based on Neotoma standardized classification. We included a spreadsheet with all used taxa and corresponding classification (TRSH – trees and shrubs, PALM - palms, UPHE – upland herbs) in the supplementary table.

We modified it to read: "For Neotoma records, arboreal pollen (AP), which serves as an indicator of tree cover relative to herbaceous vegetation, was calculated as the percentage of woody taxa (trees, shrubs, and palms), considering taxa at the genus and family level divided by the total sum of trees and shrubs, palms, and herbs, excluding mangrove and aquatic taxa, fern spores, and unidentified types. We used the Neotoma standardized classification of ecological groups (Supplementary Table 1). For manually extracted records, AP percentages were obtained from published pollen diagrams. Therefore, the criteria used to construct these AP curves may slightly differ from those applied in our calculations based on raw data. For instance, in CEB, Mauritia and Cyperaceae are often excluded from AP calculations, as these taxa are often over-represented due to strong local imprint from palm swamp vegetation (Barberi et al., 2000; Escobar-Torrez et al., 2023; Salgado-Labouriau et al., 1997). To maintain consistency within this specific region, we also excluded Mauritia and Cyperaceae from AP calculations from CEB records. Samples with pollen counts below 100 grains were removed." (lines 205-214)

Particularly for Central-Eastern Brazil (CEB), most records were manually extracted from published pollen diagrams. As such, we had to rely on the available AP curves, which may differ slightly from our calculation methods using datasets for which raw data were accessible (e.g., those obtained from Neotoma or Pangaea).

For example, in the case of Lagoa Feia (Cassino et al., 2020; Escobar-Torrez et al., 2023), we used the AP curve available in Escobar-Torrez et al. (2023), which we assume does not include Mauritia in the AP calculation.

In the Chapada dos Veadeiros record (Ferraz-Vicentini, 1999), *Mauritia* is rare and occurs in low percentages, so its inclusion or exclusion does not meaningfully affect the results. For other records, such as Cromínia (Salgado-Labouriau et al., 1997) and Vereda de Águas Emendadas (Barberi et al., 2000), we calculated AP using the sum of 100% excluding the "vereda" group. This was not the case for Lagoa Bonita (Barberi, 2001), in which we had initially included all AP, but have now corrected the record by excluding the "Pólen de brejo e vereda" group. For Vereda São José (Cassino et al., 2018), available in Neotoma, we removed *Mauritia* to ensure consistency with the manually extracted records from the region.

Nevertheless, the newly calculated AP composite closely resembles the previous one and does not result in any change to our interpretations (Fig. R2.1). Additionally, we included a new precipitation curve based on speleothem records from a site located in the northern parts of CEB (Stríkis et al., 2011, 2018), providing further regional context.

**Fig. R2.1. Updated Fig. 9** – Central-eastern Brazil (CEB) vegetation, fire, climate regimes, and human occupation: (a) Speleothem δ18O from Lapa sem Fim (LSF) and Lapa Grande (LG) caves (Stríkis et al., 2011, 2018). Downcore ln(K/Al) from marine sediment core M125-35-3, which reflects changes in the clay mineral composition and increases with chemical weathering intensity and hence, moisture availability (Meier et al., 2022). (b) Summed density probability of 14C ages from archeological sites in CEB (N = 481). (c) Arboreal pollen (AP) and (d) charcoal influx z-scores composites using 1000-yr (green and red, respectively) and 400 yr (black) smoothing half-window. Charcoal z-score negative anomaly reaching -2.2 is indicated by a circled arrow. Gray areas represent 2.5th and 97.5th confidence intervals. (e) Number (#) of records with available pollen (green) and charcoal (red) data in a 400-yr time bin.

**Fig. R2.2**. Fig. S1. Comparison of charcoal influx data transformed into z-scores with proportional relative scaling (PRS) and proportional relative scaling with a 0.2 – 21 ka base period (PRS.bp). All curves use pre-bin half width of 40 yr and window half width smoothing of 1000 yr.

Fig R2.2. Continued.

These comparisons show that despite changes in the amplitude of variability, all regions hold mostly coherent patterns of changes and a similar temporal structure. However, some key differences also arise. We briefly discuss them below.

Although all composites for NNeo indicate increased fire activity between 18 and 13 ka and during the LGM for CEB, z-scores and PRS.bp suggest a stronger rise, while PRS shows only a mild increase. The use of base period produces very similar trends between both methods, which is particularly important when including records with peak values outside our time of interest. Moreover, variabilities during the Pleistocene are more sensitive to site-specific data, due to the low availability of records.

On the other hand, in Amazonia, although the overall temporal patterns are similar across methods, a key divergence is observed: both PRS and PRS.bp suggest higher fire activity during the LGM compared to the Holocene, whereas z-scores indicate the opposite trend. In this case, z-scores seem more consistent with the known long-term fire history in the region. Several Amazonian records spanning Pleistocene ages suggest higher fire activity during the Holocene (Blaus et al., 2024; Bush et al., 2004; Colinvaux et al., 1997; Fontes et al., 2017; Hermanowski et al., 2012). An exception is southwestern Amazonia, where high charcoal concentrations occur during the Pleistocene, followed by a decline in the late Holocene as rainforest expanded (Burbridge et al., 2004).

Given that Amazonia exhibited important discrepancies between methods and is the region in which PRS is likely to best perform due to its rare frequency of fire events, we tested both z-scores and PRS.bp with multiple settings by applying combinations of sizes for bin half-window (binhw: 1500, 1000, 500, 300, and 40 years) and smoothing half-window (hf: 3000, 1500, 1000, and 400 years) (Fig. R2.3). Results show that PRS.bp is more sensitive to binhw variations, with Holocene base levels increasing systematically with larger bin sizes. Notably, while for binhw of 40 and 300 Pleistocene values are usually higher than those of the Holocene, the opposite is produced by applying binhw of 500, 1000 and 1500 years. In contrast, z-scores remain stable across different settings, consistently showing higher Holocene fire levels and no systematic variation linked to bin size.

**Fig. R2.3** Fig. S2. (a) Charcoal influx composites for Amazonia using z-scores and proportional relative scaling (PRS.bp; base period: 0.2–21 ka), applying different parameters for smoothing window half-width (hf) and pre-bin half width (binhw) lengths in years. (b) average composite values for the Holocene and Pleistocene considering the same parameters used.

Considering these observations, z-scores appear more stable and better suited to capturing long-term trends in fire activity. While PRS seems very promising, particularly for regions such as Amazonia where fire activity is rare, it still requires further testing. For instance, how it performs across different regions and time spans, as well as its sensitivity to variations in base periods and binning parameters. Thus, we see strong arguments to keep z-scores.

This brief discussion on the comparison between these two methods has been included as a Supplementary Material.

Additionally, it is crucial to emphasize that in many fire-prone ecosystems, especially savannas, herbaceous fuels—notably C4 grasses—are the primary drivers of fire regimes, while woody biomass plays a secondary role (e.g., Bond & Keeley, 2005). Therefore, correlations between tree cover and fire activity may not fully capture the dynamics of fuel availability and fire propagation (the authors seem to consider throughout the ms that arboreal cover (interpreted from AP) equals biomass growth and fuel availability).

Overall, the interpretations made by the authors are reasonable within their theoretical framework, but incorporating these additional ecological nuances could further strengthen the discussion and provide a more comprehensive understanding of the complex interplay between vegetation dynamics and fire regimes.

**Response #4**. Following specific comments from Reviewer #1 (Nicholas O'Mara), we fixed this generalization of AP as biomass. We included in the methods that AP is relative to herbaceous vegetation:

"For Neotoma records, arboreal pollen (AP), which serves as an indicator of tree cover relative to herbaceous vegetation, was calculated as the percentage of woody taxa (trees, shrubs, and palms), considering taxa at the genus and family level

divided by the total sum of trees and shrubs, palms, and herbs, excluding mangrove and aquatic taxa, fern spores, and unidentified types." (lines 205-208)

We have also fixed part of the CEB section 5.2.5 in order to account for the importance of herbaceous vegetation to fire dynamics, as follows:

5.2.5 Central-eastern Brazil (CEB): "In CEB, long-term tree cover changes are negatively correlated with fire activity (Fig. 9d,e, Fig. A2d). This pattern points to a feedback mechanism in which herbaceous vegetation facilitates fire activity, while fire, in turn, contributes to the dominance of herbs by limiting tree encroachment. Conversely, moisture-driven development of woody formations leads to the suppression of fire, which further contributes to tree cover expansion (Fig. 9d,e)." (lines 589-591)

Furthermore, we agree with the pointed concerns that fire activity and tree cover relationships are often multidirectional, and attributing clear causality is difficult, especially when both variables may be responding to the same external forcing. Thus, we have adapted the text to provide a more nuanced explanation about the potential forcings and tree cover-fire relationships:

- 5.2.6 Southeastern South America (SESA): "This suggests different relationships at different timescales. In the long-term, either the strengthening of the fire regime was driven by increase in fuel availability from woody biomass, or both tree cover and fire responded to the same external forcing (i.e., deglacial warming). In the short-term, fire events likely acted as a limiting factor for tree cover development." (line 630-634).
- 5.2.7 Extratropical Andes (ExTrAn) (this section before named "Southern Andes" is now renamed after changes related to our Response #3 to Reviewer #3, Paula A. Rodríguez-Zorro): "The positive correlation between woody biomass and fire activity supports a fuel-limited fire regime in the region, and/or suggests that observed increases in tree cover and fire activity were both driven by deglacial warming (Fig. A2f). Nevertheless, peak fire activity in the region is currently achieved at intermediate levels of both woody and herbaceous biomass (Holz et al., 2012), supporting the role of increasing woody vegetation in creating optimal conditions for fire. This is also suggested by the relevant contribution of wood charcoal in lakes from the region (Whitlock et al., 2006). Human populations only started to expand after the ACR and likely also contributed to the intensification of the fire regime (Fig. 11d) (Perez et al., 2016; Salemme and Miotti, 2008)." (lines 653-659)

**4-Potential role of megafauna extinction:**

One additional factor that may have influenced vegetation structure and fire regimes in the Neotropics during the late Quaternary is the extinction of megafauna at the Pleistocene-Holocene transition (or later - e.g. Faria et al., 2025). Large

herbivores are known to play a critical role in shaping vegetation through grazing, browsing, and trampling, thereby modulating fuel loads and fire regimes (Gill et al., 2009; Doughty et al., 2016). The disappearance of these animals in (some parts of) South America may have contributed to changes in woody cover and fuel accumulation, potentially influencing fire dynamics independently or synergistically with climatic and anthropogenic factors. While I understand that this topic may be beyond the primary scope of the manuscript, I wonder whether the authors considered this as a possible additional driver in some regions, or whether any of the available paleoecological records reflect such transitions.

**Response #5**. We appreciate the insightful comment. Although this aspect was not considered in our continental-scale analysis, it raises an important point that we have now acknowledged in the revised text. Quantifying the impact of megafauna remains challenging due to the limited availability of downcore records across Neotropics that directly assess megafaunal population changes, vegetation and fire (e.g., Bush et al., 2022; Raczka et al., 2018, 2019; Rozas-Davila et al., 2016, 2021).

We also consider these ecological changes as secondary to the major climatic shifts associated with Termination 1, as many of the observed trends can be correlated with substantial changes in temperature and precipitation during this period and during the Holocene to increasing human activities. Nevertheless, it is plausible that megafauna played a competing role in limiting fuel accumulation and constraining the encroachment of woody taxa in open savannas and grasslands. We included remarks on these effects in the discussion and include references on the role of megafauna and grazing.

**For instance:**

- 5.2.1 Northern Neotropics (NNeo): "These combined changes importantly affected megafaunal populations in NNeo, yet the implications of their decline and subsequent extinction for fire and vegetation dynamics remain unclear (Dávila et al., 2019; Rozas-Davila et al., 2021)." (lines 396-398)
- 5.2.2 Amazonia: "The impacts of megafaunal extinction also initiated long-term changes in both nutrient distribution and species turnover (Doughty et al., 2013, 2016a). However, its correlation with overall tree cover and fire regime changes remains elusive for the region." (lines 446-448)
- 5.2.5 Central-eastern Brazil (CEB): "Additionally, human activity probably also contributed to tree cover and fire trends during this period, as archaeological records show well-established occupations from ca. 13 ka onwards and expanding population in the EH (Fig. 9c) (Araujo et al., 2025; Strauss et al., 2020). Furthermore, the megafauna functional extinction during this period likely initiated long-term changes in vegetation composition (Raczka et al., 2018) and potential increase of

fuel loads (as in Gill, 2014). This effect, however, seems to have been secondary, as a decrease in tree cover is opposite to the expected response by megafaunal extinction alone (Doughty et al., 2016b; Macias et al., 2014)." (lines 577-583)

5.2.6 Southeastern South America (SESA): "The presence of a diverse megafauna potentially contributed by restricting both woody encroachment and fuel accumulation (Furquim et al., 2024; Macias et al., 2014; Prates and Perez, 2021; da Rosa et al., 2023). The combination of low woody biomass, higher impact of herbivory, cold and moist conditions restricted fire activity during this period (Fig. 10f)." (lines 606-609)

The manuscript is well-structured and comprehensible, offering a significant contribution to the understanding of long-term ecosystem dynamics in the regions studied. Although there is a scarcity of paleo records in some areas, this research is a good example of the importance of databases and open data in enhancing our understanding of Neotropical ecosystems, and it calls for additional research to address the existing knowledge gaps.

**Response #1.** We thank Paula A. Rodríguez-Zorro very much for her evaluation of our manuscript.

**GENERAL COMMENTS**

The title proposed by the authors "Vegetation and fire regimes in the Neotropics over the last 21,000 years" suggest an extensive examination of the entire neotropical region. However, this study is restricted to seven specific subregions, omitting the northern Andes, parts of Bolivia and Chile, and large portions of Brazil and Argentina. This selection was partly due to data availability, yet it resulted in a focus on these seven subregions rather than a comprehensive analysis of the Neotropics.

**Response #2.** We appreciate the observation from Paula A. Rodríguez-Zorro regarding the generality implied by our title. While it is true that our study focuses on seven subregions, we would like to emphasize that these regions collectively represent by far most of the pollen and charcoal records currently available from the Neotropics, covering a substantial portion of the region. Therefore, we believe that the chosen regions still allow us to draw meaningful conclusions about continental-scale vegetation and fire regime dynamics. Consequently, we feel that the title remains appropriate, even if it does not capture every local detail of the entire region.

In the section on vegetation settings, the authors provide a detailed description of the selected subregions, emphasizing the primary climatic drivers, such as the ITCZ, SASM, and ENSO. However, the analysis and discussion neglect the direct climatic influences from the Pacific Ocean (and records from the west flank of the Andes), particularly ENSO. The authors note in section 195: "Datapoints outside the defined subregions (black dots in Fig. 2a) were excluded from subregional analyses because

they were either isolated or located outside subregional definitions, e.g., high montane sites from NNeo >3000 m; low altitudes from CAn

**Figure.** Cyan dots depict records which were included in the analysis for the *Tropical Andes* section.

We have also included discussions on potential ENSO impacts. However, given the long-term scope of our study and the broad spatial and temporal coverage of our dataset, it is difficult to assess short-term variability. Moreover, the simultaneous

intensification of ENSO during the late Holocene, after a damped mid Holocene phase (Carré et al., 2014; Koutavas et al., 2006; Moy et al., 2002) coincides with increasing human influence on the landscape and land-use patterns, making it challenging to disentangle their respective effects (e.g., Nascimento et al., 2020). Nevertheless, we briefly included discussions on how ENSO may have contributed to the observed changes and incorporate the updated ENSO activity curve from Laguna Pallcacocha (Mark et al., 2022) to Fig. 7 from the manuscript (see preliminary version of the updated figure below, Fig. R3.1). Additionally, we have now introduced a new figure in the Appendix (Fig. R3.2), which separates the northern and southern Tropical Andes at 8°S. The discussion in the renamed section "5.2.3 Tropical Andes (TrAn)" was slightly adapted to reflect these changes, as exemplified below.

[revised manuscript text omitted]

In the methodology, it is not entirely clear how the authors have standardized the pollen data. It draws my attention to the part where they have "excluded mangrove and aquatic taxa, fern spores and unidentified types". In the case of fern spores, the authors should evaluate or clarify if they have considered tree ferns in their AP composite. In some regions, such as the Atlantic and Andean Forests, tree ferns, like *Cyathea*, *Lophosoria*, or *Dicksonia* species thrive under specific conditions, with water availability being a common factor. In pollen records, they serve as key indicators of humid conditions and are included in the pollen sum (Kesler et al., 2011, Salazar et al., 2013, de Gasper et al., 2021).

**Response #4.** We addressed the concern regarding the standardization of the methodology in our Response #2 to Reviewer #2 (Raquel F. Cassino).

Regarding tree ferns, while we recognize their value as indicators of humid conditions in regions such as the Andes and Atlantic Forests, we excluded them from the pollen sum due to inconsistent reporting across records. Including them would have limited comparability among sites. To maintain a standardized dataset suitable for regional-scale analyses, we chose to exclude all fern spores from the pollen sum. We acknowledge that this may compromise some site-specific interpretations, but it improves the overall inter-comparability of the dataset.

The following paragraph is now included in the method section:

"Although tree ferns can serve as important indicators of humid conditions in regions such as the Andes and Atlantic Forests, these were excluded from pollen sums due to inconsistent reporting across records, to ensure a standardized dataset suitable for regional-scale analyses." (lines 215-218).

Regarding fire dynamics, it is unclear how the authors determined high or low fire activity. The methodology from the compiled data is not evident. Additionally, the type of data used to reconstruct fire regimes, whether macrocharcoal, microcharcoal, or both is not specified.

Response #5. We use the methodology described in Blarquez et al. (2014). This method has also been used in other compilation studies (Daniau et al., 2012; Marlon et al., 2013; Mooney et al., 2011; Power et al., 2010). The composite z-score curve represents interpolated values using a LOWESS derived from individual site z-scores, which indicate how far charcoal influx at each site deviates from the mean. Positive and negative composite z-scores correspond to periods of above- and below-average charcoal influx, respectively, which we interpret as stronger or weaker fire regimes. While our approach does not allow us to distinguish specific aspects of the fire regime such as intensity, severity, frequency, seasonality, spatial extent of the burned vegetation, we use the broader terms "fire regime" or "fire activity" to reflect the general response of fire to environmental changes, which is directly interpreted from values of charcoal influx. Still, prompted by this comment, we improved a specific passage of the methods by clarifying this aspect of our interpretations as follows:

"Changes in charcoal records can be linked to past fire activity and used to infer shifts in fire regimes (Power et al., 2008; Daniau et al., 2010; Marlon et al., 2016). While our approach does not allow to resolve specific components of the fire regime (e.g., intensity, severity, frequency, seasonality, spatial extent of burned vegetation), it allows to identify collective changes in fire recorded as changes in charcoal influx within a given region over a long timescale." (lines 221-225).

Regarding charcoal descriptions, it was indeed insufficiently explained in the manuscript. Both macroscopic and microscopic charcoal data were treated jointly. We provided further methodological details about charcoal records (see our Response #3 to Reviewer #1 Nicholas O'Mara).

**Community (Kees Nooren)**

The article is well written, and has beautiful figures that nicely supports the results. I very much liked the first part of the article that analysis the relation between modern fire activity and current vegetation and climate. This part of their work could be an article on its own.

**Response#1.** We sincerely thank Kees Nooren for the positive feedback.**

The second part of their work is very much restricted to the lack of enough long records, as the authors also emphasize in the last paragraph of their conclusions. This easily leads to wrong interpretations. The authors for example found for the NNeo a significant negative correlation between charcoal influx, and Arboreal pollen percentage. However, this correlation is heavily influenced by one charcoal record from lake Tulane in Florida, which record is very different from the other long record used (see figure). The Tulane charcoal record is extracted from the Reading Palaeofire database, also occur in Neotoma, but can't be found in the referred article (Grimm et al., 2006). Another charcoal record from lake Tulane (WaΣs and Hansen, 1988), also in the Reading Palaeofire database, is very different (see figure), but hasn't been used. An updated record for lake Tulane is likely to be published soon (Perrot et al., 2023). I would suggest that the authors concentrate their work on the Holocene, or the last 6000 years, with a minimum number of palaeofire records for each subregion of at least ~10. They should define a minimum number of records for each subregion, and adjust the period studied accordingly. The authors should use a pre-binning of 400 or 500 years, instead of 20 years. In their current analysis many datapoints from low resolution records are missing.

**Response#2.** Thank you for the remark on specific records from Florida, USA. However, despite some variability, we still find a significant negative correlation between charcoal influx and arboreal pollen when considering only Holocene pollen and charcoal records, as shown in Fig. R4.1.

**Fig R4.1.** Correlation of charcoal influx (Nneo CHAR) and arboreal pollen (Nneo TRSH) z-scores for the northern Neotropics over (a) the last 6 ka (p < 0.01, r2 = 0.33, r = -0.57), and (b) the last 11.7 ka (p < 0.01, r2 = 0.31, r = -0.56).

We agree that the Pleistocene period is represented by fewer and lower-resolution records, resulting in less robust spatial and temporal coverage, which is acknowledged in the manuscript. Nevertheless, we believe that analyzing the full 21 ka period provides a more valuable contribution to understanding both vegetation and fire dynamics, as well as their uncertainties, than restricting the study to a shorter interval with smaller changes in some of the boundary conditions (e.g., atmospheric CO2 concentration, temperature). That said, we acknowledge that interpretations for periods with a smaller number of records should be treated with greater caution. In response, we further included in the methods a note of caution about the intervals with lower confidence:

"Caution is warranted when interpreting trends during periods with wide confidence intervals or when the composite curve approaches the upper or lower bounds of the confidence interval. These cases usually relate to periods with few records and indicate greater uncertainty and sensitivity to individual records. Strong fluctuations during such periods are likely highly uncertain and may reflect local variability of specific sites."

We applied the two-stage smoothing method with a pre-binning half-window of 20 years (40-year bins) to prevent high-resolution records from disproportionately influencing the composite. This was followed by smoothing using half-windows of 400 and 1000 years. While changing these parameters does not significantly affect long-term trends, it can influence short-term variability. We therefore interpret short-term patterns during the Pleistocene with caution, as sparse data lead to large confidence intervals and unstable composites. Bootstrap resampling further indicates periods of disagreement among records or where individual records may dominate. In general, z-score values are very stable in terms of trends to varying prebinning half-window values, although half-window smoothing can yield more stable curves, while losing details. For a brief discussion on this matter, please see our Response #3 and Fig. R2.3 to Reviewer #2 (Raquel F. Cassino). As such, increasing the pre-binning to 400 or 500 years would require a broader smoothing window, which could diminish the resolution of Holocene features without necessarily improving confidence in the Pleistocene, a period for which we already refrain from drawing further inferences due to the low data reliability.

Regarding the Tulane record, we also included the record from Watts and Hansen (1988) in our analysis. However, we keep the Tulane record available from the Reading Database. This record is also available and referenced in Neotoma (DOI: 10.21233/a8jq-7f35) with the following dataset notes: "Grimm et al (2006) publication describes core, but charcoal dataset not used.". In fact, the data available from the Reading Database were directly provided by Eric Grimm and Jim

Clark for the compilation published by Daniau et al. (2010) and later incorporated into the ACER compilation by Sanchez-Goñi et al. (2017). Thus, although it is not in the 2006 publication, this data was produced by the same authors and has been previously used in other publications. As for the record likely to be published soon (Perrott et al., 2023), it cannot be included in the current synthesis as it is not yet available.

---

## Author Response (AR2)

Dear Editor,

We thank you and the reviewers for the careful evaluation of our manuscript. In this revised version, we implemented three minor changes, outlined below:

- (1) We fixed the labels in Figure 1 to reflect the revised classification, as requested by one of the reviewers.
- (2) Following the same reviewer's suggestion, we incorporated the reference *Pym et al. (2023)* in the Tropical Andes section (lines 489–491) (in blue). This addition required a slight reorganization of the paragraph and the inclusion of the following sentence:
- "Additionally, an overall decline of megafauna in tropical Andes resulted in sensitive ecological consequences associated with vegetation turnover, e.g., the encroachment of both palatable and woody taxa, as well as fuel build-up (Bush et al., 2022; Pym et al., 2023)."
- (3) We added one short sentence in the abstract (lines 29-30) and two short sentences between lines 694–698 (in blue) to complement and refine ideas that are already present in the manuscript.

Lines 29-30: "In warmer tropical regions (NNeo, Amazonia, CEB), moisture availability was likely the main controlling factor of both vegetation and fire, with the effects of low CO2 amplifying these constraints."

Lines 694-698: "A stronger impact of low CO2atm is expected on vegetation in warm tropical regions (e.g., NNeo, Amazonia, CEB, NEB) compared to cold regions (e.g., ExTrAn, SESA, TrAn) (Sage and Coleman, 2001). Reduced CO2atm is also suggested by modelling data to weaken the fire regime by altering availability and properties of biomass (Haas et al., 2023), while it can also increase the severity of fires by slowing post-burn tree recovery (Bond et al., 2003)."

Apart from these stated minor changes, the paper is exactly the same as the previous version.

On behalf of all coauthors,

Thomas Kenji Akabane